# GRADIENT CLIPPING ACCELERATES SADDLE AVOIDANCE IN DISTRIBUTED OPTIMIZATION

## ABSTRACT

A critical challenge in distributed nonconvex optimization is efficiently avoiding saddle points, which is vital for ensuring accurate and fast convergence to a desired equilibrium point. In this work, we demonstrate that gradient clipping, a technique widely used in machine learning to mitigate gradient explosion, can significantly accelerate the escape from saddle points and improve the speed of convergence to second-order stationary points in distributed optimization. More specifically, we propose an algorithm that exploits gradient clipping to achieve faster convergence in distributed nonconvex optimization. The result is significant in that gradient clipping is necessary and widely used anyway to avoid exploding gradients in deep learning, and hence, the extra benefit of faster saddle avoidance is achieved for free. In fact, we prove that our algorithm converges to a desired second-order stationary point faster than all existing saddle avoidance approaches for distributed optimization. Numerical experiments on benchmark datasets validate the effectiveness of the proposed method.

## 1 INTRODUCTION

Distributed optimization has become crucial in applications ranging from training large machine learning models (Dean et al., 2012), coordinating multi-agent systems (Chen et al., 2019), managing sensor networks (Rabbat & Nowak, 2004), optimizing power grids (Mohsenian-Rad et al., 2010), to supporting cloud computing (Chen et al., 2015) and content delivery networks (Apostolopoulos et al., 2002). It enables multiple spatially distributed computing devices, referred to as agents, to work collaboratively in optimizing a global objective function that is only partially observable by individual agents due to the distributed and local acquisition/storage of data. Unlike centralized methods which require all data to be aggregated to a single location, distributed optimization allows each agent to store data locally and communicate only with its immediate neighbors.

In many distributed optimization applications, the problem can be formulated as follows:

$$\min_{\boldsymbol{\theta} \in \mathbb{R}^d} F(\boldsymbol{\theta}) = \frac{1}{N} \sum_{i=1}^{N} f_i(\boldsymbol{\theta}), \tag{1}$$

where $f_i(\cdot) : \mathbb{R}^d \to \mathbb{R}$ represents the local objective function of agent $i$ and is private to agent $i$. $F(\cdot) : \mathbb{R}^d \to \mathbb{R}$ is the average of the local objective functions. $N$ is the number of agents. Many methods have been proposed to solve problem (1), based on techniques such as gradient methods (see, e.g., Nedić & Ozdaglar (2009); Lian et al. (2017); Tang et al. (2018); Shi et al. (2015); Xu et al. (2017); Qu & Li (2017); Xin & Khan (2018)) , distributed alternating direction method of multipliers (see, e.g., Shi et al. (2014)), and distributed Newton methods (see, e.g., Zhang et al. (2018)). Most of the results consider convex objectives. Although there has also been progress on decentralized nonconvex optimization, the majority of these works provide only first-order stationarity guarantees (see, e.g., Sun et al. (2020); Pan et al. (2020); Xin et al. (2021)). In contrast, distributed algorithms that achieve second-order guarantees remain relatively limited(Xian & Huang, 2023). Escaping saddle points is a central challenge in nonconvex learning, as the primary difficulty often lies in flat saddle regions rather than the presence of many local minima (Ge et al., 2015).

In nonconvex optimization, one of the main challenges is escaping saddle points (stationary points that are not local extrema). In fact, avoiding saddle points is commonly regarded as one of the most

important tasks in deep learning since the main bottleneck in parameter optimization is not due to the existence of multiple local minima but the existence of many saddle points that trap gradient update (Ge et al., 2015).

Saddle-point avoidance is an active area of research even for centralized optimization, with new results emerging frequently. Existing results can be broadly divided into second-order (Hessian-based) and first-order (gradient-based) approaches. Hessian-based algorithms utilize the Hessian matrix to differentiate between first- and second-order stationary points. These methods either compute the full Hessian matrix (e.g., Curtis et al. (2017); Nesterov & Polyak (2006)) or rely on Hessian-vector products (e.g., Agarwal et al. (2017); Carmon et al. (2018); Carmon & Duchi (2019)) at each iteration. However, the computation and storage costs associated with these approaches grow linearly or even quadratically with the dimension of the optimization vector. This scalability issue is especially acute in modern deep learning applications, where the variable dimension can reach hundreds of millions (Tang et al., 2020). First-order algorithms have recently gained increased attention due to their computational efficiency. Variants of gradient descent (GD) incorporating random perturbations have been shown to escape saddle points effectively (see, e.g., Ge et al. (2015); Du et al. (2017)). In addition, random initialization has also been proven to ensure convergence to a second-order stationary point almost surely (Lee et al., 2016). However, despite its theoretical guarantees, it has been demonstrated that this approach can take exponential time to escape saddle points under certain conditions, making it impractical in many cases (Du et al., 2017).

Compared with the centralized case, distributed optimization brings additional challenges in saddle-point avoidance. In fact, in distributed optimization, the saddle points of individual local objective functions generally differ from those of the global objective function. Combined with the fact that the local objective functions are private to individual agents, a participating agent does not have access to necessary global information, such as the gradient or Hessian information of $F(\cdot)$, to detect or strategically avoid saddle points of the global objective function in its local iterations.

## 1.1 OUR CONTRIBUTIONS

- We achieve, for the first time, gradient clipping in gradient-tracking based distributed optimization with provable convergence. Although gradient-tracking based distributed optimization algorithms outperform distributed gradient methods in both convergence accuracy and speed (see, e.g., Yang et al. (2019); Nedić et al. (2018)), their integration with gradient clipping is elusive since gradient-clipping induced nonlinearity will significantly complicate convergence analysis. In fact, the inability to retain provable convergence under gradient clipping is one of the main obstacles in applying gradient-tracking based algorithms in machine learning where the gradient may grow unbounded.

- Based on the proposed gradient-tracking algorithm that retains provable convergence after incorporating gradient clipping, we systematically show, for the first time, that gradient clipping can be exploited to accelerate escape from saddle points. Note that although Levy (2016) uses a similar concept of normalization to evade saddle points, it has fundamental differences from our gradient clipping based approach: the normalization in Levy (2016) persistently divides the gradient by its norm even when the gradient is close to zero. This leads to constant oscillations around stationary points, making it impossible to stay precisely at a desired equilibrium point; To the contrary, we use gradient clipping which divides the gradient by its norm only when the gradient value is over a certain threshold. This fundamental difference makes our approach ensure much better and stable convergence. Moreover, Levy (2016) considers the centralized setting and assumes strong convexity in the vicinity of stationary points, which is not needed here.

- More interestingly, we show that this gradient-clipping based approach enables us to achieve a saddle-evading speed that is faster than all existing counterparts in a fully distributed setting. Note that by equipping each agent with a counter to keep track of the state of each agent, Xian & Huang (2023) achieves the same saddle-point avoidance speed as ours. However, its counter based strategy relies on global statistical information of local counters, which makes it difficult to implement in a fully distributed setting.

- We performed numerical evaluations using standard nonconvex optimization experiments (logistic regression and robust Principal Component Analysis based video processing) with benchmark datasets. The results confirm that our algorithm indeed has faster and more accurate convergence than existing counterparts.

## 1.2 RELATED WORK

**Distributed Optimization**    Distributed algorithms for problem (1) have been extensively studied (see, e.g., Nedić & Ozdaglar (2009); Di Lorenzo & Scutari (2016); Jakovetić et al. (2014); Kia et al. (2015); Lobel et al. (2011)). Recently, distributed first-order algorithms have garnered significant attention, particularly in deep learning, due to their high computational efficiency. These algorithms can be classified as gradient-descent-based algorithms and gradient-tracking-based algorithms.

Distributed gradient descent algorithms are among the most studied for solving (1) (Nedić & Ozdaglar, 2009; Yang et al., 2019; Nedić et al., 2018; Nedić & Liu, 2018; Nedić, 2020; Notarstefano et al., 2019; Nedić et al., 2010). However, due to inherent discrepancies between global and local objective functions, these algorithms converge to a consensual exact solution only when a diminishing stepsize is used, which can significantly slow down convergence. To cope with the speed-accuracy dilemma of gradient-descent based algorithms, gradient-tracking based methods have gained intensive attention (see, e.g., Xu et al. (2017); Qu & Li (2017); Xin & Khan (2018); Di Lorenzo & Scutari (2016); Nedić et al. (2017); Xin et al. (2019); Wang & Nedić (2023); Song et al. (2024); Lu et al. (2019); Wang et al. (2024); Pu et al. (2020); Pu (2020); Scutari & Sun (2019); Sun et al. (2016; 2019); Xu et al. (2015); Pu & Nedić (2021); Wang & Başar (2022)) , which can converge to an exact consensual solution under a constant stepsize. Gradient-tracking algorithms are agnostic to data heterogeneity, making them suitable to deal with non-IID data in machine learning (Koloskova et al., 2021; Nguyen et al., 2024).

**Saddle-Point Avoidance**    Results have emerged for saddle avoidance in the fully distributed setting (see, e.g., Vlaski & Sayed (2021); Bo & Wang (2024)). Most of them are based on first-order methods. To make the consensus error small, they use very small constant stepsizes (see, e.g., Vlaski & Sayed (2021)) or diminishing stespizes (see, e.g., Bo & Wang (2024)), which slow down convergence. Results have also been proposed for distributed optimization in a semi-distributed setting where some global information is assumed to be available for saddle avoidance (see, e.g., Xian & Huang (2023); Tziotis et al. (2020); Avdiukhin & Yaroslavtsev (2021)). Notably, the approach in Xian & Huang (2023) leverages a counter mechanism to track local phases. However, it requires global statistics of all counters, which limits its applicability in a fully distributed setting.

**Gradient Normalization and Clipping**    Gradient normalization has been used in both discrete-time optimization (see, e.g., Levy (2016)) and continuous-time optimization (see, e.g., Murray et al. (2019)). The basic idea is to normalize a gradient $\nabla f(\boldsymbol{x})$ as $\frac{\nabla f(\boldsymbol{x})}{\|\nabla f(\boldsymbol{x})\|}$ before feeding it to iteration. It is worth noting that Levy (2016) has proposed to use normalization to evade saddle point. However, since this approach divides the gradient by its norm even when the gradient is close to zero, it may cause oscillations around the equilibrium.

Gradient clipping approaches clip a gradient $\nabla f(\boldsymbol{x}^k)$ as $c_0 \min\left\{\frac{1}{c_0}, \frac{1}{\|\nabla f(\boldsymbol{x}^k)\|}\right\} \nabla f(\boldsymbol{x}^k)$ where $c_0$ is the clipping threshold: gradients larger than $c_0$ are scaled down to have a norm of $c_0$, while gradients smaller than $c_0$ remain unchanged. This adaptive mechanism avoids the issue of oscillations encountered in gradient normalization. Gradient clipping has recently been used in gradient-descent based distributed optimization (Liu et al., 2022). However, when the optima of local objectives are different from the global optimum, directly applying clipping on local gradients will amply optimization error since the local gradients are non-zero when evaluated at a global optimum.

## 2 PROBLEM FORMULATION AND PRELIMINARIES

### 2.1 NOTATIONS

We use bold lower-case and upper-case letters such as $\boldsymbol{x}$ and $\boldsymbol{W}$ to denote vectors and matrices. We use $\|\cdot\|$ to represent the $\ell_2$ norm of vectors and the Frobenius norm of matrices. For a function $F(\cdot) : \mathbb{R}^d \to \mathbb{R}$, we use $\nabla F(\cdot)$ and $\nabla^2 F(\cdot)$ to denote its gradient and Hessian, respectively. We use $\mathcal{O}(\cdot)$ to hide absolute constants, and $\tilde{\mathcal{O}}(\cdot)$ to hide poly-logarithmic factors that do not depend on any problem-specific parameter. We use $[N]$ to represent the set $\{1, 2, \cdots, N\}$ and $\lambda_{\min}(\cdot)$ to represent the minimal eigenvalue of a matrix.

## 2.2 DISTRIBUTED OPTIMIZATION

In problem (1), we assume that $f_i(\cdot)$ is private to agent $i$, and no single agent has access to the global objective function. To cooperatively optimize a global objective, gradient-tracking methods require agents to exchange both intermediate optimization variables and local estimates of the global gradient with their neighbors at each iteration. We consider a directed communication graph, where the exchange of the two variables uses two (possibly different) mixing matrices $\mathbf{R} = [R_{ij}]_{N \times N}$ and $\mathbf{C} = [C_{ij}]_{N \times N}$. An entry $R_{ij} > 0$ (or $C_{ij} > 0$) indicates that agent $i$ can receive information from agent $j$. As in most existing results on distributed optimization over directed graphs (Xin & Khan, 2018; Pu et al., 2020; Pu & Nedić, 2021), we impose the following assumption on the mixing matrices:

**Assumption 2.1.** $\mathbf{R} \in \mathbb{R}^{N \times N}$ *is nonnegative and row-stochastic* ($\mathbf{R1} = \mathbf{1}$)*, and* $\mathbf{C} \in \mathbb{R}^{N \times N}$ *is nonnegative and column-stochastic* ($\mathbf{1}^\top \mathbf{C} = \mathbf{1}^\top$)*. Both have positive diagonal entries. The graph* $\mathcal{G}_{\mathbf{R}}$ *contains at least one spanning tree, and* $\mathcal{G}_{\mathbf{C}^\top}$ *is strongly connected.*

**Lemma 2.2.** *Under Assumption 2.1, the matrix* $\mathbf{R}$ *has a nonnegative left eigenvector* $\mathbf{u}^\top$ *(w.r.t. eigenvalue 1) with* $\mathbf{u}^\top \mathbf{1} = N$*. Similarly, the matrix* $\mathbf{C}$ *has a strictly positive right eigenvector* $\mathbf{v}$ *(w.r.t. eigenvalue 1) with* $\mathbf{1}^\top \mathbf{v} = N$.

Problem (1) can be reformulated as the following multi-agent optimization problem:

$$\min_{\boldsymbol{x} \in \mathbb{R}^{N \times d}} f(\boldsymbol{x}) = \frac{1}{N} \sum_{i=1}^N f_i(\boldsymbol{x}_i), \ \ \text{s.t.} \ \boldsymbol{x}_1 = \boldsymbol{x}_2 = \cdots = \boldsymbol{x}_N, \tag{2}$$

where $\boldsymbol{x} = [\boldsymbol{x}_1^\top; \boldsymbol{x}_2^\top; \ldots; \boldsymbol{x}_N^\top] \in \mathbb{R}^{N \times d}$. In this paper, the local objective function $f_i(\boldsymbol{x}_i)$ and global objective function $f(\boldsymbol{x})$ can be nonconvex. They are assumed to satisfy the following conditions:

**Assumption 2.3.** *The function* $f$ *is lower bounded, i.e., there exists some constant* $\bar{f}$ *such that* $\bar{f} := \inf_{\boldsymbol{x} \in \mathbb{R}^d} f(\boldsymbol{x}) > -\infty$.

**Assumption 2.4.** *Every* $f_i(\cdot)$ *is differentiable and* $L_i$*-Lipschitz as well as* $\rho_i$*-Hessian Lipschitz. Specifically, for any* $\boldsymbol{x}_1, \boldsymbol{x}_2 \in \mathbb{R}^d$, $\|\nabla f_i(\boldsymbol{x}_1) - \nabla f_i(\boldsymbol{x}_2)\| \leqslant L_i \|\boldsymbol{x}_1 - \boldsymbol{x}_2\|$ *and* $\|\nabla^2 f_i(\boldsymbol{x}_1) - \nabla^2 f_i(\boldsymbol{x}_2)\| \leqslant \rho_i \|\boldsymbol{x}_1 - \boldsymbol{x}_2\|$.

*One can verify that the global gradient* $\nabla F(\boldsymbol{\theta}) = \frac{1}{N} \sum_{i=1}^N \nabla f_i(\boldsymbol{\theta})$ *and Hessian* $\nabla^2 F(\boldsymbol{\theta}) = \frac{1}{N} \sum_{i=1}^N \nabla^2 f_i(\boldsymbol{\theta})$ *are L-Lipschitz and* $\rho$*-Hessian Lipschitz, with* $L = \frac{1}{N} \sum_i L_i$ *and* $\rho = \frac{1}{N} \sum_i \rho_i$.

## 2.3 SADDLE POINTS

**Definition 2.5.** *For a twice differentiable objective function* $F(\cdot)$*, we call* $\boldsymbol{\theta}^\star \in \mathbb{R}^d$ *a first-order (respt. second-order) stationary point if* $\nabla F(\boldsymbol{\theta}^\star) = \mathbf{0}$ *(respt.* $\nabla F(\boldsymbol{\theta}^\star) = \mathbf{0}$ *and* $\lambda_{\min}(\nabla^2 F(\boldsymbol{\theta}^\star)) \geqslant 0$*) holds. Moreover, a first-order stationary point* $\boldsymbol{\theta}^\star$ *can be viewed as belonging to one of the following three categories: 1, 2)* local minimum *(respt.* local maximum*): there exists a scalar* $\gamma > 0$ *such that* $F(\boldsymbol{\theta}^\star) \leqslant F(\boldsymbol{\theta})$ *(respt.* $F(\boldsymbol{\theta}^\star) \geqslant F(\boldsymbol{\theta})$*) holds for any* $\boldsymbol{\theta}$ *satisfying* $\|\boldsymbol{\theta}^\star - \boldsymbol{\theta}\| \leqslant \gamma$*; 3)* saddle point*: neither of the above two cases is true, i.e., for any* $\gamma > 0$*, there exist* $\boldsymbol{\theta}_1$ *and* $\boldsymbol{\theta}_2$ *satisfying* $\|\boldsymbol{\theta}_1 - \boldsymbol{\theta}^\star\| \leqslant \gamma$ *and* $\|\boldsymbol{\theta}_2 - \boldsymbol{\theta}^\star\| \leqslant \gamma$ *such that* $F(\boldsymbol{\theta}_1) < F(\boldsymbol{\theta}^\star) < F(\boldsymbol{\theta}_2)$ *holds.*

Since distinguishing saddle points from local minima for smooth functions is NP-hard in general (Nesterov, 2000), we focus on a subclass of saddle points, i.e., $(\epsilon, \sqrt{\epsilon})-$strict saddle points:

**Definition 2.6.** *For a twice-differentiable function* $F(\cdot)$*, we say that* $\boldsymbol{\theta}^\star \in \mathbb{R}^d$ *is an* $\epsilon-$***strict saddle point*** *if 1)* $\boldsymbol{\theta}^\star$ *is an* $\epsilon-$*first-order stationary point, i.e.,* $\|\nabla F(\boldsymbol{\theta}^\star)\| \leqslant \epsilon$*; and 2)* $\lambda_{\min}(\nabla^2 F(\boldsymbol{\theta}^\star)) \leqslant -\sqrt{\rho\epsilon}$*, where* $\rho$ *is the Hessian Lipschitz parameter in Assumption 2.4. Similarly,* $\boldsymbol{\theta}^\star \in \mathbb{R}^d$ *is an* $\epsilon-$***second-order stationary point*** *if 1)* $\boldsymbol{\theta}^\star$ *is an* $\epsilon-$*first-order stationary point, i.e.,* $\|\nabla F(\boldsymbol{\theta}^\star)\| \leqslant \epsilon$ *and 2)* $\lambda_{\min}(\nabla^2 F(\boldsymbol{\theta}^\star)) > -\sqrt{\rho\epsilon}$.

For a smooth nonconvex function, a generic saddle point satisfies $\lambda_{\min}(\nabla^2 F(\boldsymbol{\theta}^\star)) \leqslant 0$. By focusing specifically on "strict" saddle points, we exclude cases where $\lambda_{\min}(\nabla^2 F(\boldsymbol{\theta}^\star)) = 0$. In fact, recent machine learning studies have shown that, in many commonly used models, all saddle points are strict. Some typical examples include tensor decomposition (Ge et al., 2015), dictionary learning (Sun et al., 2017), smooth semidefinite programming (Boumal et al., 2016), and robust principal component analysis (Ge et al., 2017).

---

**Algorithm 1** Clipped-Gradient Tracking Algorithm over Directed Graphs

---

**Initialization:** Each agent $i$ chooses $\boldsymbol{x}_i^0 \in \mathbb{R}^d$ and sets $\boldsymbol{y}_i^0 = \nabla f_i(\boldsymbol{x}_i^0)$.
**Parameters:** Stepsize $\alpha$, clipping threshold $c_0$, row-stochastic matrix $\mathbf{R}$, column-stochastic matrix $\mathbf{C}$, noise amplitude $\theta$, and noise interval $\mathcal{K}_0$.
**for** $k = 0, 1, 2, \ldots$ **do**
   **for** all $i \in [N]$ **do**
     1. Agent $i$ sends $\boldsymbol{x}_i^k$ and $\boldsymbol{y}_i^k$ to its out-neighbors and receives $\boldsymbol{x}_j^k$, $\boldsymbol{y}_j^k$ from its in-neighbors.
     2. Receive $\boldsymbol{x}_j^k$ and $\boldsymbol{y}_j^k$ from neighbors and update the local optimization variable $\boldsymbol{x}_i^{k+1}$:

$$\boldsymbol{x}_i^{k+1} = \begin{cases} \sum_{j=1}^N R_{ij}\boldsymbol{x}_j^k + \theta\boldsymbol{\xi}_i^k - \alpha \cdot \min\{1, \frac{c_0}{\|\boldsymbol{y}_i^k\|}\}\boldsymbol{y}_i^k, & k > 0 \ \& \ k \bmod \mathcal{K}_0 = 0, \\ \sum_{j=1}^N R_{ij}\boldsymbol{x}_j^k - \alpha \cdot \min\{1, \frac{c_0}{\|\boldsymbol{y}_i^k\|}\}\boldsymbol{y}_i^k, & \text{otherwise.} \end{cases} \quad (3)$$

     3. Update the gradient estimate $\boldsymbol{y}_i^{k+1}$ as:

$$\boldsymbol{y}_i^{k+1} = \sum_{j=1}^N C_{ij}\boldsymbol{y}_j^k + \nabla f_i(\boldsymbol{x}_i^{k+1}) - \nabla f_i(\boldsymbol{x}_i^k) \quad (4)$$

---

## 3 PROPOSED ALGORITHM

In this section, we propose an algorithm that incorporates gradient clipping in gradient-tracking based distributed optimization. To our knowledge, no such results have been reported in the literature since the nonlinearity induced by gradient clipping will greatly exacerbate the already complicated convergence analysis for gradient-tracking based algorithms. By employing carefully constructed inequalities to encapsulate the effects of gradient clipping operations, we successfully prove that convergence can still be ensured in our proposed algorithm, even with gradient clipping (Section 4.1). More importantly, we prove that the gradient clipping technique can greatly accelerate the escape from saddle points (Section 4.2), which, to our knowledge, has not been reported before. It is worth noting that although gradient normalization has already been used in Levy (2016) to avoid saddle points in centralized optimization, it is fundamentally different from our gradient-clipping based technique: the normalization based approach in Levy (2016) persistently divides the gradient by its norm even when the gradient is close to zero, which makes it impossible to stay precisely at a desired equilibrium point (as even when the actual gradient is close to zero, it becomes very big after normalization and hence drives the state away from a desired stationary point); On the contrary, we use gradient clipping which divides the gradient by its norm only when the gradient value is over a certain threshold value, which makes it possible for the state to stay precisely at a desired equilibrium point. In addition, Levy (2016) assumes that the objective function is strongly convex in the vicinity of desired stationary points, which restricts its applications.

The algorithm is summarized in Algorithm 1. It can be seen that a gradient-tracking structure is employed. Namely, besides the state variable $\boldsymbol{x}_i$, each agent employs an additional variable $\boldsymbol{y}_i$ to estimate the global gradient. As indicated in Karimireddy et al. (2020), such an additional variable is necessary to ensure accurate convergence when the data on different agents are non-IID, even in the presence of a parameter server.

In the second step of each iteration, agents mix their local values with information received from neighbors. During this step, Gaussian noise $\boldsymbol{\xi}_i$ is periodically injected into the optimization process to facilitate escaping from saddle points (Jin et al., 2021). We assume that the noise satisfies Assumption 3.1.

**Assumption 3.1.** *The noise $\{\boldsymbol{\xi}_i\}$ follows zero-mean Gaussian distribution with covariance $(1/d)\boldsymbol{I}$, ensuring $\mathbb{E}\left[\|\boldsymbol{\xi}_i\|^2\right] = 1$.*

In Eq. (3) of Algorithm 1, $\boldsymbol{y}_i^k$ is clipped. When the magnitude of $\boldsymbol{y}_i^k$ is small but greater than the threshold value $c_0$ (e.g., when the iteration is in the vicinity of a saddle point), this operation will increase the effective gradient that is actually fed into the algorithm, and hence, will facilitate escape from a saddle point in the presence of injected noise $\boldsymbol{\xi}_i$. When the value of $\|\boldsymbol{y}_i^k\|$ is below the threshold value, clipping will not take place, and hence, our mechanism allows the state to

stay precisely at a desired equilibrium point. The threshold value $c_0$ is set according to the desired accuracy $\epsilon$, as detailed later in Section 4.2.

It is worth noting that instead of applying gradient clipping on the local gradient directly, we apply clipping on individual agents' local estimates $\boldsymbol{y}_i^k$. This is because in a general distributed optimization problem, the optimal values of local objective functions are different from the global optimal solution, and hence, the local gradients evaluated at a desired global equilibrium could be far away from zero and different from each other. This is the main reason why the distributed gradient methods (see, e.g., Song et al. (2024)) are subject to optimization errors even under strongly convex objective functions. Applying clipping will exacerbate the differences in local gradients, and hence, could amplify this error. To the contrary, in our proposed gradient tracking algorithm, all local estimates $\boldsymbol{y}_i^k$ will be zero on an optimal solution of the global objective function, and hence, applying clipping on them will not lead to errors.

Next, we show that Algorithm 1 escapes saddle points more quickly than existing approaches and therefore converges faster to a $\epsilon-$second-order stationary point. We leave all proofs in the Appendix.

# 4 THEORETICAL ANALYSIS

## 4.1 CONSENSUAL CONVERGENCE TO A FIRST-ORDER STATIONARY POINT

We first prove that the proposed algorithm can ensure all agents to reach consensus both on their optimization variables $\boldsymbol{x}_i$ and global gradient estimates $\boldsymbol{y}_i^k$. To this end, we express the iterative dynamics in a more compact form.

By defining $\boldsymbol{x}^k = [(\boldsymbol{x}_1^k)^\top; \cdots; (\boldsymbol{x}_N^k)^\top] \in \mathbb{R}^{N \times d}$, $\boldsymbol{\xi}^k = [(\boldsymbol{\xi}_1^k)^\top; \cdots; (\boldsymbol{\xi}_N^k)^\top] \in \mathbb{R}^{N \times d}$, $\boldsymbol{y}^k = [(\boldsymbol{y}_1^k)^\top; \cdots; (\boldsymbol{y}_N^k)^\top] \in \mathbb{R}^{N \times d}$, $\nabla f(\boldsymbol{x}^k) = [\nabla f_1^\top(\boldsymbol{x}_1^k); \cdots; \nabla f_N^\top(\boldsymbol{x}_N^k)] \in \mathbb{R}^{N \times d}$, and $\boldsymbol{\alpha}_k \boldsymbol{y}^k = [(\alpha_1^k \boldsymbol{y}_1^k)^\top; \cdots; (\alpha_N^k \boldsymbol{y}_N^k)^\top] \in \mathbb{R}^{N \times d}$ where $\alpha_i^k = \alpha \min\{1, \frac{c_0}{\|\boldsymbol{y}_i^k\|}\} \in \mathbb{R}$, we can recast the update in (3) and (4) into the following more compact form:

$$\boldsymbol{x}^{k+1} = \boldsymbol{R}\boldsymbol{x}^k + \theta_k \boldsymbol{\xi}^k - \boldsymbol{\alpha}_k \boldsymbol{y}^k, \quad \boldsymbol{y}^{k+1} = \boldsymbol{C}\boldsymbol{y}^k + \nabla f\left(\boldsymbol{x}^{k+1}\right) - \nabla f\left(\boldsymbol{x}^k\right), \tag{5}$$

where $\theta_k = \theta$ when $k > 0$ and $k \bmod \mathcal{K}_0 = 0$; otherwise, $\theta_k = 0$.

We define the network-average variables as $\bar{\boldsymbol{x}}^k = \frac{1}{N}\boldsymbol{u}^\top \boldsymbol{x}^k$ and $\bar{\boldsymbol{y}}^k = \frac{1}{N}\mathbf{1}^\top \boldsymbol{y}^k$, where $\boldsymbol{u}$ is the left eigenvector of $\boldsymbol{R}$ corresponding to eigenvalue 1. Their dynamics follow:

$$\bar{\boldsymbol{x}}^{k+1} = \bar{\boldsymbol{x}}^k - \frac{1}{N}\boldsymbol{u}^\top \boldsymbol{\alpha}_k \boldsymbol{y}^k, \quad \bar{\boldsymbol{y}}^{k+1} = \bar{\boldsymbol{y}}^k + \frac{1}{N}\mathbf{1}^\top\left(\nabla f(\boldsymbol{x}^{k+1}) - \nabla f(\boldsymbol{x}^k)\right). \tag{6}$$

Define the error between individual agents' local optimization variables and the average optimization variable $\bar{\boldsymbol{x}}^k$ as $\boldsymbol{e}_{x,k} := \boldsymbol{x}^k - \mathbf{1}(\bar{\boldsymbol{x}}^k)^\top$. Similarly, denote the gradient tracking error as $\boldsymbol{e}_{y,k} := \boldsymbol{y}^k - \boldsymbol{v}(\bar{\boldsymbol{y}}^k)^\top$. Based on (3) and (4), along with $\boldsymbol{R}\mathbf{1} = \mathbf{1}$ and $\mathbf{1}^\top \boldsymbol{C} = \mathbf{1}^\top$, we can obtain the dynamics for $\boldsymbol{e}_{x,k}$ and $\boldsymbol{e}_{y,k}$:

$$\boldsymbol{e}_{x,k+1} = \left(\boldsymbol{R} - \frac{\mathbf{1}\boldsymbol{u}^\top}{N}\right)\boldsymbol{e}_{x,k} + \theta_k\left(\boldsymbol{I} - \frac{\mathbf{1}\boldsymbol{u}^\top}{N}\right)\boldsymbol{\xi}^k - \left(\boldsymbol{I} - \frac{\mathbf{1}\boldsymbol{u}^\top}{N}\right)\boldsymbol{\alpha}_k \boldsymbol{y}^k, \tag{7}$$

$$\boldsymbol{e}_{y,k+1} = \left(\boldsymbol{C} - \frac{\boldsymbol{v}\mathbf{1}^\top}{N}\right)\boldsymbol{e}_{y,k} + \left(\boldsymbol{I} - \frac{\boldsymbol{v}\mathbf{1}^\top}{N}\right)\left(\nabla f(\boldsymbol{x}^{k+1}) - \nabla f(\boldsymbol{x}^k)\right). \tag{8}$$

We can prove that $\|\boldsymbol{e}_{x,k}\|^2$ converges to zero (with an error determined by the noise amplitude):

**Theorem 4.1** (Consensual Convergence to a First-order Stationary Point). *Let Assumption 2.1, Assumption 2.3, Assumption 2.4, and Assumption 3.1 hold and the stepsize $\alpha$ satisfies $0 < \alpha < \min\{C_1, \ldots, C_4\}$ [1]. Then, $F\left(\bar{\boldsymbol{x}}^k\right)$ converges and the convergence rate is $\frac{1}{k}\sum_{s=1}^k \bar{\alpha}_s \mathbb{E}[\|\nabla F\left(\bar{\boldsymbol{x}}^s\right)\|^2] \leqslant \mathcal{O}(N\theta^2) + \mathcal{O}(\frac{\sqrt{N}}{k(1-\sigma_C^2)})$, where $\bar{\alpha}_s = \alpha \min\{1, \frac{c_0}{\|\boldsymbol{v}\|\|\nabla F(\bar{\boldsymbol{x}}^s)\|}\}$. We also have the consensus rate of $\{\boldsymbol{x}^k\}$ and $\{\boldsymbol{y}^k\}$ as $\frac{1}{k}\sum_{s=1}^k \mathbb{E}[\|\boldsymbol{e}_{x,s}\|^2] \leqslant \mathcal{O}(\frac{N^{3/2}}{k(1-\sigma_R^2)}) + \mathcal{O}(\frac{N^{3/2}\theta^2}{1-\sigma_R^2})$ and $\frac{1}{k}\sum_{s=1}^k \mathbb{E}[\|\boldsymbol{e}_{y,s}\|^2] \leqslant \mathcal{O}(\frac{N}{k(1-\sigma_R^2)(1-\sigma_C^2)}) + \mathcal{O}(\frac{N\theta^2}{1-\sigma_C^2})$.*

---

[1] $C_1 = N^{\frac{1}{2}}(1 - \sigma_R^2)^{\frac{1}{2}}/2\sqrt{3}L\|\boldsymbol{v}\|\|\boldsymbol{I} - \frac{\mathbf{1}\boldsymbol{u}^\top}{N}\|_{\boldsymbol{R}}$, $C_2 = (1 - \sigma_C^2)^{\frac{1}{2}}/12\|\boldsymbol{I} - \frac{\boldsymbol{v}\mathbf{1}^\top}{N}\|_C$, $C_3 = (1 - \sigma_R^2)^{\frac{1}{2}}(1 - \sigma_C^2)^{\frac{1}{2}}/2\sqrt{2}\mathcal{C}_{x,1}^{\frac{1}{2}}\mathcal{C}_{y,1}^{\frac{1}{2}}$, $C_4 = (\boldsymbol{u}^\top \boldsymbol{v})^{\frac{1}{2}}(1 - \sigma_R^2)^{\frac{1}{2}}/4L\|\boldsymbol{v}\|\sqrt{3\mathcal{C}_{x,2}}(3L\kappa_{uv}\|\boldsymbol{v}\| + \frac{2\kappa_{uv}^2\|\boldsymbol{v}\|^2}{\boldsymbol{u}^\top \boldsymbol{v}})^{\frac{1}{2}}$, $\mathcal{C}_{x,1} = \frac{12(1+\sigma_R^2)\|\boldsymbol{I} - \frac{\mathbf{1}\boldsymbol{u}^\top}{N}\|_{\boldsymbol{R}}^2 \delta_{R,2}^2}{1-\sigma_R^2}$, $\mathcal{C}_{x,2} = \frac{3N(1+\sigma_R^2)\|\boldsymbol{I} - \frac{\mathbf{1}\boldsymbol{u}^\top}{N}\|_{\boldsymbol{R}}^2 \delta_{R,2}^2\|\boldsymbol{v}\|^2}{1-\sigma_R^2}$, and $\mathcal{C}_{y,1} = \frac{2(1+\sigma_C^2)(1+2\sigma_R^2)\|\boldsymbol{I} - \frac{\boldsymbol{v}\mathbf{1}^\top}{N}\|_C^2 L^2}{1-\sigma_C^2}$

Detailed analytical expressions for the consensus rate are provided in Eq. (49) in the Appendix.

Theorem 4.1 explicitly shows how the convergence of both gradient norms and consensus errors depend on the number of agents $N$ and the spectral gaps $1 - \sigma_R^2$ and $1 - \sigma_C^2$ of the mixing matrices. In particular, weaker connectivity (i.e., $\sigma_R$ or $\sigma_C$ closer to one) or a larger network size slows the decay of both the consensus error and the gradient-tracking error. The results also imply that the proposed algorithm converges to a neighborhood of a first-order stationary point, with the radius determined jointly by the network topology and the noise level $\theta^2$.

The result in Theorem 4.1 can be extended to the stochastic gradient scenario under Assumption 4.2 below (detailed consensus rate expressions can be found in Appendix Eq. (59)):

**Assumption 4.2.** *For agent $i$ at iteration $k$, the stochastic gradient satisfies* $\mathbb{E}[\nabla f_i(\boldsymbol{x}_i^k, \boldsymbol{v}_i^k)] = \nabla f_i(\boldsymbol{x}_i^k)$ *and* $\mathbb{E}[\|\nabla f_i(\boldsymbol{x}_i^k, \boldsymbol{v}_i^k) - \nabla f_i(\boldsymbol{x}_i^k)\|^2] = \sigma_{\zeta}^2$.

**Corollary 4.3.** *Under the stochastic gradient setting in Assumption 4.2, and assuming similar conditions and stepsize selection as in Theorem 4.1, the function value $F(\bar{\boldsymbol{x}}^k)$ converges. The convergence rate is given by* $\frac{1}{k} \sum_{s=1}^{k} \bar{\alpha}_s \mathbb{E}[\|\nabla F(\bar{\boldsymbol{x}}^s)\|^2] \leqslant \mathcal{O}(\frac{\sqrt{N}}{k(1-\sigma_C^2)}) + \mathcal{O}(N\theta^2) + \mathcal{O}(\frac{N^{1/2}}{(1-\sigma_R^2)^{1/2}(1-\sigma_C^2)}\sigma_{\zeta}^2)$. *We also have the consensus rate of $\{\boldsymbol{x}^k\}$ and $\{\boldsymbol{y}^k\}$ as* $\frac{1}{k} \sum_{s=1}^{k} \mathbb{E}[\|\boldsymbol{e}_{x,s}\|^2] \leqslant \mathcal{O}(\frac{N^{3/2}}{k(1-\sigma_R^2)}) + \mathcal{O}(\frac{N^{3/2}\theta^2}{(1-\sigma_R^2)}) + \mathcal{O}(\frac{\sigma_{\zeta}^2}{1-\sigma_R^2})$ *and* $\frac{1}{k} \sum_{s=1}^{k} \mathbb{E}[\|\boldsymbol{e}_{y,s}\|^2] \leqslant \mathcal{O}(\frac{N}{k(1-\sigma_R^2)(1-\sigma_C^2)}) + \mathcal{O}(\frac{N\theta^2}{1-\sigma_C^2}) + \mathcal{O}(\frac{\sigma_{\zeta}^2}{1-\sigma_C^2})$.

### 4.2 SADDLE AVOIDANCE AND CONVERGENCE TO A SECOND-ORDER STATIONARY POINT

We focus on saddle avoidance when all agents are in consensus based on the following two observations: 1) From Theorem 4.1, the local optimization variables $\boldsymbol{x}_i^k$ of individual agents are guaranteed to achieve consensus, and 2) before consensus is reached, the inter-agent interaction introduces an additional force that keeps individual states $\boldsymbol{x}_i^k$ evolving and prevents them from being trapped at any fixed value. Thus, escaping saddle-point is the most difficult when all agents' optimization variables are in consensus. In this case, the gradient tracking variables $\boldsymbol{y}_i^k$ are small. Therefore, dividing the gradient by its norm effectively amplifies the magnitude of the effective gradient, and hence, accelerates the escape from saddle point. This is proven formally as follows:

**Theorem 4.4** (Escaping Saddle Points). *Let Assumption 2.1, Assumption 2.3, Assumption 2.4, and Assumption 3.1 hold. Algorithm 1 ensures that each noise period ($\mathcal{K}_0$ iterations) reduces the objective function by a substantial amount in expectation. More specifically, in no more than $\mathcal{K}_0 = \mathcal{O}(\frac{1}{\epsilon})$ iterations after one noise injection, we have $\mathbb{E}[F(\bar{\boldsymbol{x}}^k)] - F(\tilde{\boldsymbol{x}}) \leqslant -Q$, where $\tilde{\boldsymbol{x}}$ is an $\epsilon-$strict saddle point and $Q$ is a constant satisfying $Q = \mathcal{O}(\epsilon^2)$.*

**Remark 4.5.** *The proof of Theorem 4.4 shows that the stepsizes, noise amplitude/period and clipping threshold should satisfy $\alpha = \mathcal{O}(\epsilon)$, $\theta = \mathcal{O}(\epsilon)$, $\mathcal{K}_0 = \mathcal{O}(\frac{1}{\alpha})$, and $c_0 = \mathcal{O}(\epsilon)$ to escape $(\epsilon, \sqrt{\epsilon})-$strict saddle points while ensuring agreement of all agents' states. By setting $\alpha c_0 = \epsilon^2$, our method aligns with the stepsize choice in Levy (2016) for normalized gradient methods but generalizes it to the clipped gradient scenario. Moreover, the stepsize $\alpha$ and noise amplitude $\theta$ jointly determine the number of iterations required to escape saddle points, as formalized in Definition C.2 of Section C.1.*

In fact, besides evading a saddle point, Theorem 4.4 establishes that in each $\mathcal{K}_0 = \mathcal{O}(\frac{1}{\epsilon})$ iterations following one addition of noise, the algorithm is guaranteed to decrease in the function value by a significant amount. Therefore, by using some lower bound on the optimal function value $\bar{f}$, we can repeatedly add noise sufficient times to ensure avoidance of all potentially encountered saddle points, and hence, to ensure convergence to a second-order stationary point. [2]

We can prove that within a finite number of iterations, Algorithm 1 will encounter an $\epsilon-$second-order stationary point at least once:

---

[2]It is worth noting that such a bound can be obtained easily in many settings. For example, in the matrix factorization problem where a low-rank matrix $\boldsymbol{U} \in \mathbb{R}^{d \times r}$ is used to approximate a high-dimension matrix $\boldsymbol{M}^\star \in \mathbb{R}^{d \times d}$, the objective function is $f(\boldsymbol{U}) = \frac{1}{2}\|\boldsymbol{U}\boldsymbol{U}^\top - \boldsymbol{M}^\star\|_F^2$ and we can use $\bar{f} = 0$ as the lower bound (Jin et al., 2017).

Table 1: Results for finding $\epsilon$-second-order stationary point.

| SETTING | ALGORITHM(S) | COMPLEXITY DETERMINISTIC GRADIENT | COMPLEXITY STOCHASTIC GRADIENT |
|---|---|---|---|
| Centralized | Jin et al. (2021) | $\tilde{\mathcal{O}}(\epsilon^{-2})$ | $\mathcal{O}(\epsilon^{-4})$ |
| | Daneshmand et al. (2018) | $\mathcal{O}(\epsilon^{-2.5})$ | $\mathcal{O}(\epsilon^{-5})$ |
| | Chen et al. (2022), Levy (2016) | - | $\tilde{\mathcal{O}}(\epsilon^{-3})$ |
| | Allen-Zhu (2018b) | - | $\tilde{\mathcal{O}}(\epsilon^{-3.25})$ |
| | Allen-Zhu (2018a), Xu et al. (2018) | - | $\tilde{\mathcal{O}}(\epsilon^{-3.5})$ |
| | Li (2019) | $\tilde{\mathcal{O}}(\epsilon^{-2})$ | $\tilde{\mathcal{O}}(\epsilon^{-3.5})$ |
| | Ge et al. (2015), | - | $\mathcal{O}(\epsilon^{-4})$ |
| | Zhang & Li (2021) | $\tilde{\mathcal{O}}(\epsilon^{-1.75})$ | $\tilde{\mathcal{O}}(\epsilon^{-4})$ |
| | Jin et al. (2018) | $\tilde{\mathcal{O}}(\epsilon^{-1.75})$ | - |
| | Fang et al. (2018) | - | $\tilde{\mathcal{O}}(\epsilon^{-3})$ |
| | Daneshmand et al. (2018) | - | $\tilde{\mathcal{O}}(\epsilon^{-5})$ |
| Semi-distributed | Xian & Huang (2023) | - | $\tilde{\mathcal{O}}(\epsilon^{-3})$ |
| | Tziotis et al. (2020) | $\mathcal{O}(\epsilon^{-4})$ | - |
| | Avdiukhin & Yaroslavtsev (2021) | - | $\tilde{\mathcal{O}}(\epsilon^{-4})$ |
| Distributed | Vlaski & Sayed (2021) | - | $\mathcal{O}(\epsilon^{-6.5})$ |
| | This work | $\mathcal{O}(\epsilon^{-3})$ | $\mathcal{O}(\epsilon^{-3})$ |

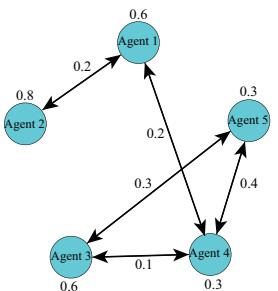

Figure 1: The interaction graph of five agents.

**Theorem 4.6** (Converging to an $\epsilon-$second-order Stationary Point). *Let Assumption 2.1, Assumption 2.3, Assumption 2.4, and Assumption 3.1 hold. For any $\epsilon > 0$, Algorithm 1 will visit an $\epsilon-$second-order stationary point at least once in $K = \mathcal{O}(\frac{1}{\epsilon^3})$ iterations.*

The result can be extended to the stochastic gradient setting under Assumption 4.2:

**Corollary 4.7.** *Under the stochastic gradient setting in Assumption 4.2, the iteration complexity required to escape $(\epsilon, \sqrt{\epsilon})-$saddle points and converge to an $\epsilon$-second-order stationary point remains similar to that in the deterministic gradient setting. Specifically, the number of iterations needed to escape an $(\epsilon, \sqrt{\epsilon})-$saddle point is $\mathcal{K}_0 = \mathcal{O}(\frac{1}{\epsilon})$, while the total iteration complexity to attain an $\epsilon$-second-order stationary point is $K = \mathcal{O}(\frac{1}{\epsilon^3})$.*

**Remark 4.8.** *From Theorem 4.6, Algorithm 1 reaches an $\epsilon$-second-order stationary point within $K = \mathcal{O}(\frac{1}{\epsilon^3})$ iterations. This implies that noise—whose purpose is to escape saddle regions—is injected only a finite number of times during the first $\mathcal{O}(\epsilon^{-3})$ iterations. After this stage, the algorithm can proceed without additional noise. Therefore, in the infinite-time time horizon, the convergence bound in Theorem 4.1 can be rewritten as $\frac{1}{k}\sum_{s=1}^{k}\bar{\alpha}_s\mathbb{E}[\|\nabla F(\bar{x}^s)\|^2] \leqslant \mathcal{O}(\frac{N\theta^2}{k}) + \mathcal{O}(\frac{\sqrt{N}}{k(1-\sigma_C^2)})$, and Algorithm 1 achieves exact convergence eventually.*

Theorem 4.6 (Corollary 4.7) demonstrates that our algorithm converges to an $\epsilon-$second-order stationary point in $\mathcal{O}(\frac{1}{\epsilon^3})$ iterations. This is much faster than the $\mathcal{O}(\frac{1}{\epsilon^{6.5}})$ saddle-point evading speed in Vlaski & Sayed (2021) where only gradient noise but no clipping is used. To our knowledge, our $\mathcal{O}(\frac{1}{\epsilon^3})$ iteration complexity in reaching a second-order stationary point is faster than all existing results for saddle-point avoidance in distributed optimization. In fact, it matches the state-of-the-art convergence rates in the centralized setting or semi-distributed setting [3] in finding second-order stationary points (see, e.g., Xian & Huang (2023); Chen et al. (2022)). The detailed comparison with existing results is summarized in Table 1.

## 5 NUMERICAL EXPERIMENTS

Here, we present experimental results for logistic regression, robust Principal Component Analysis and convolutional neural network (CNN). The first two experiments were performed under $R = C$ (which reduces to an undirected graph depicted in Figure 1) whereas the third experiment used distinct $R = C$ corresponding to a directed graph as described in Appendix D.1. Additional experimental results for binary classification (where the saddle point can be analytically obtained) and neural network training are provided in Appendix D.

---

[3]We refer to methods for "semi-distributed" setting as methods that are partially decentralized but still require global information or network-wide aggregation at certain steps. For example, in Xian and Huang (2023), the termination mechanism depends on knowing exactly how many worker nodes satisfy a stopping condition, which requires an additional global aggregation process to collect Boolean indicators from all agents. In contrast, our algorithm is fully distributed by design: all operations are based solely on local states and neighbor-to-neighbor communication, without any global statistics, global counters, or centralized processes.

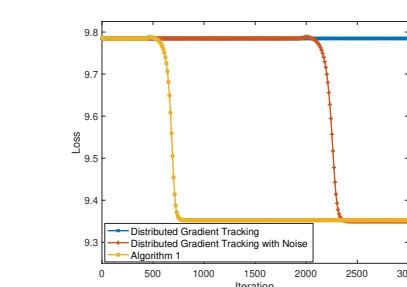
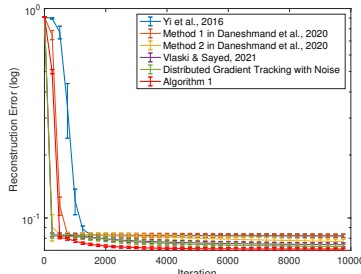

Figure 2: Regularized Logistic Regression on Gisette dataset.

Figure 3: Reconstruction error in Robust PCA background subtraction.

### 5.1 REGULARIZED LOGISTIC REGRESSION

In this experiment, we employ non-convex regularized logistic regression to solve a binary classification problem using the Gisette dataset (Guyon et al., 2004). The feature vectors of the training samples are denoted as $\boldsymbol{h} \in \mathbb{R}^d$ with $d = 5000$ for each sample in the dataset, and the labels of the binary classes are $y \in \{-1, 1\}$. The loss function is $\ell(\boldsymbol{x}; \{\boldsymbol{h}, y\}) = -y \log(\frac{1}{1+\exp(\boldsymbol{x}^\top \boldsymbol{h})}) + (1 - y) \log(\frac{\exp(\boldsymbol{x}^\top \boldsymbol{h})}{1+\exp(\boldsymbol{x}^\top \boldsymbol{h})}) + \sum_{i=1}^{d} \frac{x_i^2}{1+x_i^2}$, where $\{\boldsymbol{h}, y\}$ represents a training tuple.

To visualize the performance of our algorithm around the saddle point, we explicitly compute the saddle points of the loss function using 10 data samples and initialize all agents on a saddle point. For Algorithm 1, the noise amplitude for each dimension was set to $1 \times 10^{-7}$, and the clipping threshold and stepsize were set to $c_0 = 0.4$ and $\alpha = 0.06$, respectively. The evolution of the global loss was shown by the yellow line in Figure 2, which shows that Algorithm 1 can evade the saddle point. To show that clipping can expedite saddle avoidance, we also plotted the global-loss trajectory of our algorithm with the same amount of noise but without clipping (termed as "Distributed Gradient Tracking with Noise" and represented by the red curve in Figure 2). The stepsize was set to $\alpha = 0.039$, which was found in our experiments to lead to the fastest escape from the saddle point while not causing divergence. Figure 2 shows that the escape speed from the saddle point is reduced significantly, which confirms that clipping indeed expedites saddle avoidance. In addition, we also found that further removing noise (termed as "Distributed Gradient Tracking" and represented by the blue curve) will lose the capability of saddle avoidance. We tested Algorithm 1 with various choices of parameters $c_0$ and $\theta$ (see additional experimental results in Appendix D.4), and Algorithm 1 consistently demonstrated satisfactory saddle-point evading performance.

### 5.2 ROBUST PRINCIPAL COMPONENT ANALYSIS (PCA)

In this experiment, we address the background subtraction problem in computer vision using robust PCA. For a video represented as a sequence of images, robust PCA is utilized to distinguish dynamic objects from the static background. Specifically, given a sequence of images represented by a data matrix $\boldsymbol{M} \in \mathbb{R}^{m \times n}$, robust PCA decomposes $\boldsymbol{M}$ into a low-rank component $\boldsymbol{U}\boldsymbol{V}^\top$ capturing the background, and a sparse component $\boldsymbol{S}$ representing moving objects. Here, $\boldsymbol{U} \in \mathbb{R}^{m \times r}$, $\boldsymbol{V} \in \mathbb{R}^{n \times r}$, $\boldsymbol{S} \in \mathbb{R}^{m \times n}$, and $r \ll \min\{m, n\}$. The decomposition can be formulated as the optimization problem $\min_{\boldsymbol{U}, \boldsymbol{V}} \ f(\boldsymbol{U}, \boldsymbol{V}) + \mu_2 \|\boldsymbol{U}^\top \boldsymbol{U} - \boldsymbol{V}^\top \boldsymbol{V}\|_F^2$, where $f(\boldsymbol{U}, \boldsymbol{V}) = \min_{\boldsymbol{S} \in \mathcal{S}_{\bar{\alpha}}} \frac{1}{2} \|\boldsymbol{U}\boldsymbol{V}^\top + \boldsymbol{S} - \boldsymbol{M}\|_F^2$, $\mu_2$ is a constant and $\mathcal{S}_{\bar{\alpha}}$ represents the set of matrices with at most $\bar{\alpha}$−fraction of nonzero entries in every column and every row (Ma & Aybat, 2018). It has been proven in Ge et al. (2015) that all saddle points in robust PCA are strict saddle points.

The "WallFlower" datasets from Microsoft (RPC, 2017) were employed in evaluation. We randomly assign 200 image frames with $56 \times 56$ pixels to each agent, creating the data matrix $\boldsymbol{M}_i$ of agent $i$ with dimensions $m = 9408$ and $n = 200$. The optimization parameters were chosen as $\mu_2 = 0.01$, $\bar{\alpha} = 0.2$, and $r = 30$. For comparison, we also implemented the saddle avoidance approaches in Daneshmand et al. (2020), Vlaski & Sayed (2021) and Yi et al. (2016). The stepsizes for these algorithms were carefully chosen such that doubling the stepsize will lead to divergence. We also implemented the conventional gradient-tracking algorithm without clipping. For our algorithm, the noise amplitude was set to $\theta = 0.01$ and the clipping threshold was set to $c_0 = 0.2$.

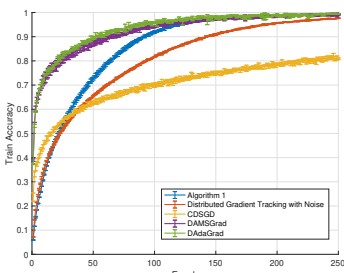

Figure 4: CNN experiment on the CIFAR-100 dataset: Training accuracy.

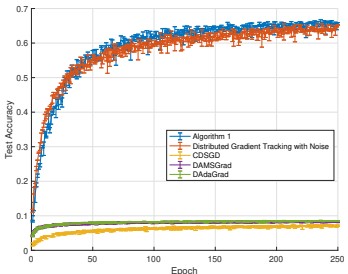

Figure 5: CNN experiment on the CIFAR-100 dataset: Test accuracy.

Figure 3 shows the evolutions of the reconstruction error $\mathcal{E} = \sum_{i=1}^{N} \|M_i - U_i V_i^\top - S_i\| / \|M_i\|_F^2$ for all the algorithms. Our algorithm has a much faster and more accurate convergence. This indicates that our algorithm effectively navigates towards better stationary points by successfully evading strict saddle points.

### 5.3 CONVOLUTIONAL NEURAL NETWORK (CNN)

For this experiment, we train a CNN on the CIFAR-100 dataset, which contains 50,000 training images spanning 100 classes. Training on CIFAR-100 is a highly nonconvex problem with a considerably more complex loss landscape due to the large number of classes. The dataset is evenly partitioned across the five agents, and the batch size is set to 32. The CNN architecture is a deep network identical to that used in Jiang et al. (2017).

We compare the proposed Algorithm 1 with the decentralized stochastic gradient method CDSGD (Jiang et al., 2017), as well as the decentralized adaptive methods DAMSGrad (Chen et al., 2023) and DAdaGrad (Chen et al., 2023), which adapt the well-known adaptive methods AMSGrad (Reddi et al.) and AdaGrad (Duchi et al., 2011) to the decentralized setting. For Algorithm 1, the noise amplitude, noise-injection interval, and clipping threshold are set to $\theta = 1 \times 10^{-4}$, $\mathcal{K}_0 = 100$, and $c_0 = 10$, respectively. The stepsize $\alpha$ is chosen as $0.05$ for Algorithm 1. For CDSGD, the largest stepsizes that ensure convergence were adopted to provide a fair comparison.

The training and test accuracies, averaged over 10 runs, are shown in Figure 4 and Figure 5, respectively. Both CDSGD and the decentralized adaptive algorithms exhibit poor generalization (test accuracy). In these algorithms, consensus among agents is insufficient, as each agent follows its own gradient direction, with adaptive methods amplifying this effect through local-geometry-based step sizes. This leads to overfitting on local data (reflected by high training accuracy on local data), which is particularly problematic in the highly heterogeneous CIFAR-100 setting with 100 classes. Consequently, these methods achieve high training accuracy but substantially lower test accuracy compared to our approach. It is worth noting that removing clipping degrades performance (see the variant termed "Distributed Gradient Tracking with Noise" which is Algorithm 1 with all components except gradient clipping), highlighting that both noise injection and gradient clipping are essential for effective saddle avoidance in our algorithm.

## 6 CONCLUSIONS

In this work, we propose a distributed optimization algorithm that can evade saddle points faster than all existing results for distributed nonconvex optimization. Our basic idea is to integrate gradient tracking and gradient clipping, which is of interest itself given that gradient-tracking algorithms have been shown to converge faster and more accurately than conventional distributed gradient descent algorithms when local objective functions are different or local data follow non-IID distributions. We prove that gradient clipping also provides an effective means for saddle-point avoidance. Given that gradient clipping is already widely used in deep learning to address exploding gradients, our result paves the way for attaining the additional benefit of fast saddle-point avoidance in distributed optimization without incurring additional costs.

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

## APPENDIX CONTENTS

# A PRELIMINARY RESULTS

**Lemma A.1.** *For any agent $i \in [N]$ in Algorithm 1, define*

$$\alpha_i^k = \alpha \min\left\{1, \frac{c_0}{\|\boldsymbol{y}_i^k\|}\right\}, \quad \bar{\alpha}_i^k = \alpha \min\left\{1, \frac{c_0}{v_i\|\nabla F(\bar{\boldsymbol{x}}^k)\|}\right\}.$$

*Then, the following inequality holds under Assumption 2.1, Assumption 2.4, and Assumption 2.3:*

$$\left|\alpha_i^k - \bar{\alpha}_i^k\right| \|\boldsymbol{y}_i^k\| \leqslant \bar{\alpha}_i^k \|\boldsymbol{y}_i^k - v_i \nabla F(\bar{\boldsymbol{x}}^k)\|. \tag{9}$$

*Furthermore, if we define $\bar{\alpha}_k = \alpha \min\left\{1, \frac{c_0}{\|\boldsymbol{v}\|\|\nabla F(\bar{\boldsymbol{x}}^k)\|}\right\}$, then we have $\bar{\alpha}_k \leqslant \bar{\alpha}_i^k \leqslant \frac{\|\boldsymbol{v}\|}{v_i}\bar{\alpha}_k$. Thus, we have*

$$\left|\alpha_i^k - \bar{\alpha}_i^k\right| \|\boldsymbol{y}_i^k\| \leqslant \frac{\|\boldsymbol{v}\|}{v_i}\bar{\alpha}_k \|\boldsymbol{y}_i^k - v_i \nabla F(\bar{\boldsymbol{x}}^k)\|. \tag{10}$$

*Proof.* According to the definitions of $\alpha_i^k$ and $\bar{\alpha}_i^k$, we establish relationship in Equation (9) on a case-by-case basis as follows:

**Case 1:** $v_i\|\nabla F(\bar{\boldsymbol{x}}^k)\| \leqslant c_0, \|\boldsymbol{y}_i^k\| \leqslant c_0$:
In this case, we have $\bar{\alpha}_i^k = \alpha_i^k = \alpha$, which implies

$$\left|\alpha_i^k - \bar{\alpha}_i^k\right| \|\boldsymbol{y}_i^k\| = 0.$$

**Case 2:** $v_i\|\nabla F(\bar{\boldsymbol{x}}^k)\| \leqslant c_0, \|\boldsymbol{y}_i^k\| > c_0$:
In this case, we have $\alpha_i^k < \alpha = \bar{\alpha}_i^k$, which implies

$$\begin{aligned}
\left|\alpha_i^k - \bar{\alpha}_i^k\right| \|\boldsymbol{y}_i^k\| &= \alpha c_0 \left|\frac{1}{\|\boldsymbol{y}_i^k\|} - \frac{1}{c_0}\right| \|\boldsymbol{y}_i^k\| \\
&= \alpha c_0 \left(\frac{1}{c_0} - \frac{1}{\|\boldsymbol{y}_i^k\|}\right) \|\boldsymbol{y}_i^k\| \\
&= \alpha\left(1 - \frac{c_0}{\|\boldsymbol{y}_i^k\|}\right) \|\boldsymbol{y}_i^k\|.
\end{aligned} \tag{11}$$

In the second equality above, we have used the condition $\|\boldsymbol{y}_i^k\| > c_0$ in this case.

By substituting $\alpha$ with $\bar{\alpha}_i^k$ and using the bound $v_i\|\nabla F(\bar{\boldsymbol{x}}^k)\| \leqslant c_0$, we obtain

$$\begin{aligned}
\left|\alpha_i^k - \bar{\alpha}_i^k\right| \|\boldsymbol{y}_i^k\| &\leqslant \bar{\alpha}_i^k\left(\|\boldsymbol{y}_i^k\| - \|v_i\nabla F(\bar{\boldsymbol{x}}^k)\|\right) \\
&\leqslant \bar{\alpha}_i^k\|\boldsymbol{y}_i^k - v_i\nabla F(\bar{\boldsymbol{x}}^k)\|,
\end{aligned}$$

where the last inequality follows from $\big|\|a\| - \|b\|\big| \leqslant \|a - b\|$.

**Case 3:** $v_i\|\nabla F(\bar{\boldsymbol{x}}^k)\| > c_0, \|\boldsymbol{y}_i^k\| \leqslant c_0$:
In this case, we have $\alpha_i^k = \alpha > \bar{\alpha}_i^k$, which implies

$$\begin{aligned}
\left|\alpha_i^k - \bar{\alpha}_i^k\right| \|\boldsymbol{y}_i^k\| &= \alpha c_0 \left|\frac{1}{c_0} - \frac{1}{v_i\|\nabla F(\bar{\boldsymbol{x}}^k)\|}\right| \|\boldsymbol{y}_i^k\| \\
&= \alpha\left(1 - \frac{c_0}{v_i\|\nabla F(\bar{\boldsymbol{x}}^k)\|}\right) \|\boldsymbol{y}_i^k\| \\
&= \frac{\alpha\|\boldsymbol{y}_i^k\|}{v_i\|\nabla F(\bar{\boldsymbol{x}}^k)\|}\left(\|v_i\nabla F(\bar{\boldsymbol{x}}^k)\| - c_0\right),
\end{aligned} \tag{12}$$

where in the second equality we have used the condition $v_i\|\nabla F(\bar{\boldsymbol{x}}^k)\| > c_0$ in this case. Using the definition of $\bar{\alpha}_i^k$ and the triangle inequality $|\|a\| - \|b\|| \leqslant \|a - b\|$, we have the following inequality:

$$\left|\alpha_i^k - \bar{\alpha}_i^k\right| \left\|\boldsymbol{y}_i^k\right\| \leqslant \frac{\alpha c_0}{v_i \left\|\nabla F\left(\bar{\boldsymbol{x}}^k\right)\right\|} \left(v_i \left\|\nabla F\left(\bar{\boldsymbol{x}}^k\right)\right\| - \left\|\boldsymbol{y}_i^k\right\|\right)$$
$$\leqslant \bar{\alpha}_i^k \left\|\boldsymbol{y}_i^k - v_i \nabla F\left(\bar{\boldsymbol{x}}^k\right)\right\|. \tag{13}$$

**Case 4:** $v_i\|\nabla F(\bar{\boldsymbol{x}}^k)\| > c_0, \|\boldsymbol{y}_i^k\| > c_0$:

In this case, we have $\alpha_i^k < \alpha$, $\bar{\alpha}_i^k < \alpha$, which leads to

$$\left|\alpha_i^k - \bar{\alpha}_i^k\right| \|\boldsymbol{y}_i^k\| = \alpha c_0 \left|\frac{1}{\|\boldsymbol{y}_i^k\|} - \frac{1}{v_i\|\nabla F(\bar{\boldsymbol{x}}^k)\|}\right| \|\boldsymbol{y}_i^k\|$$
$$= \alpha c_0 \frac{\left|v_i\|\nabla F(\bar{\boldsymbol{x}}^k)\| - \|\boldsymbol{y}_i^k\|\right|}{v_i\|\nabla F(\bar{\boldsymbol{x}}^k)\|}.$$

Using $\bar{\alpha}_i^k = \frac{\alpha c_0}{v_i\|\nabla F(\bar{\boldsymbol{x}}^k)\|}$ and the triangle inequality $|\|a\| - \|b\|| \leqslant \|a - b\|$, we obtain

$$\left|\alpha_i^k - \bar{\alpha}_i^k\right| \|\boldsymbol{y}_i^k\| \leqslant \bar{\alpha}_i^k \|\boldsymbol{y}_i^k - v_i \nabla F\left(\bar{\boldsymbol{x}}^k\right)\|.$$

Therefore, the relationship in equation (9) is true in all cases, which completes the proof. $\qquad\square$

**Lemma A.2.** *Under Assumption 2.1 and Assumption 2.4, the iterations of Algorithm 1 satisfy the following inequalities:*

$$\left\|\boldsymbol{\alpha}_k \boldsymbol{y}^k\right\|^2 \leqslant 6\bar{\alpha}_k^2 \left\|\boldsymbol{y}^k - \boldsymbol{v}\nabla F(\bar{\boldsymbol{x}}^k)\right\|^2 + 3N\bar{\alpha}_k^2\|\boldsymbol{v}\|^2 \left\|\nabla F(\bar{\boldsymbol{x}}^k)\right\|^2, \tag{14}$$

$$\left\|\boldsymbol{y}^k - \boldsymbol{v}\nabla F(\bar{\boldsymbol{x}}^k)\right\|^2 \leqslant 2\|\boldsymbol{e}_{y,k}\|^2 + \frac{2\|\boldsymbol{v}\|^2 L^2}{N}\|\boldsymbol{e}_{x,k}\|^2, \tag{15}$$

*Proof.* We first analyze the term $\boldsymbol{\alpha}^k \boldsymbol{y}^k$.

Recalling the notation $\boldsymbol{\alpha}^k \boldsymbol{y}^k = \left[(\alpha_1^k \boldsymbol{y}_1^k)^\top; \cdots; (\alpha_N^k \boldsymbol{y}_N^k)^\top\right] \in \mathbb{R}^{N \times d}$, we obtain

$$\left\|\boldsymbol{\alpha}_k \boldsymbol{y}^k\right\|^2 \leqslant \sum_{i=1}^{N} \left\|\alpha_i^k \boldsymbol{y}_i^k\right\|^2.$$

To further decompose each block, we add and subtract the same reference quantities $\bar{\alpha}_k \boldsymbol{y}_i^k$ and $\bar{\alpha}_k \boldsymbol{v}\nabla F(\bar{\boldsymbol{x}}^k)$ as follows:

$$\alpha_i^k \boldsymbol{y}_i^k = (\alpha_i^k - \bar{\alpha}_k)\boldsymbol{y}_i^k + \bar{\alpha}_k \left(\boldsymbol{y}_i^k - \boldsymbol{v}\nabla F(\bar{\boldsymbol{x}}^k)\right) + \bar{\alpha}_k \boldsymbol{v}\nabla F(\bar{\boldsymbol{x}}^k).$$

Applying the inequality $\|a + b + c\|^2 \leqslant 3\|a\|^2 + 3\|b\|^2 + 3\|c\|^2$ yields

$$\left\|\boldsymbol{\alpha}_k \boldsymbol{y}^k\right\|^2 \leqslant 3\sum_{i=1}^{N} \left\|(\alpha_i^k - \bar{\alpha}_k)\boldsymbol{y}_i^k\right\|^2 + 3\sum_{i=1}^{N} \left\|\bar{\alpha}_k(\boldsymbol{y}_i^k - \boldsymbol{v}\nabla F(\bar{\boldsymbol{x}}^k))\right\|^2$$
$$+ 3\sum_{i=1}^{N} \left\|\bar{\alpha}_k \boldsymbol{v}\nabla F(\bar{\boldsymbol{x}}^k)\right\|^2.$$

Using Lemma A.1 to bound the first two summations, we obtain

$$\left\|\boldsymbol{\alpha}_k \boldsymbol{y}^k\right\|^2 \leqslant 6\bar{\alpha}_k^2 \sum_{i=1}^{N} \left\|\boldsymbol{y}_i^k - \boldsymbol{v}\nabla F(\bar{\boldsymbol{x}}^k)\right\|^2 + 3N\bar{\alpha}_k^2\|\boldsymbol{v}\|^2\|\nabla F(\bar{\boldsymbol{x}}^k)\|^2,$$

which establishes inequality (14).

We now turn to the second inequality (15). We rewrite the term by adding and subtracting $\boldsymbol{v}\bar{\boldsymbol{y}}$:

$$\left\|\boldsymbol{y}^k - \boldsymbol{v}\nabla F(\bar{\boldsymbol{x}}^k)\right\|^2 \leqslant 2\left\|\boldsymbol{y}^k - \boldsymbol{v}\bar{\boldsymbol{y}}^k\right\|^2 + 2\|\boldsymbol{v}\|^2\left\|\bar{\boldsymbol{y}}^k - \nabla F(\bar{\boldsymbol{x}}^k)\right\|^2.$$

Using the definitions

$$\bar{\boldsymbol{y}}^k = \frac{1}{N}\sum_{i=1}^N \nabla f_i(\boldsymbol{x}_i^k), \qquad \nabla F(\bar{\boldsymbol{x}}^k) = \frac{1}{N}\sum_{i=1}^N \nabla f_i(\bar{\boldsymbol{x}}^k),$$

we obtain

$$\bar{\boldsymbol{y}}^k - \nabla F(\bar{\boldsymbol{x}}^k) = \frac{1}{N}\sum_{i=1}^N \left(\nabla f_i(\boldsymbol{x}_i^k) - \nabla f_i(\bar{\boldsymbol{x}}^k)\right).$$

By Jensen's inequality,

$$\left\|\frac{1}{N}\sum_{i=1}^N a_i\right\|^2 \leqslant \frac{1}{N}\sum_{i=1}^N \|a_i\|^2,$$

we have

$$\left\|\bar{\boldsymbol{y}}^k - \nabla F(\bar{\boldsymbol{x}}^k)\right\|^2 \leqslant \frac{1}{N}\sum_{i=1}^N \left\|\nabla f_i(\boldsymbol{x}_i^k) - \nabla f_i(\bar{\boldsymbol{x}}^k)\right\|^2.$$

By the Lipschitz gradient condition in Assumption 2.4, we further obtain

$$\left\|\bar{\boldsymbol{y}}^k - \nabla F(\bar{\boldsymbol{x}}^k)\right\|^2 \leqslant \frac{L^2}{N}\sum_{i=1}^N \|\boldsymbol{x}_i^k - \bar{\boldsymbol{x}}^k\|^2 = \frac{L^2}{N}\|\boldsymbol{e}_{x,k}\|^2.$$

Combining the above results yields

$$\left\|\boldsymbol{y}^k - \boldsymbol{v}\nabla F(\bar{\boldsymbol{x}}^k)\right\|^2 \leqslant 2\|\boldsymbol{e}_{y,k}\|^2 + \frac{2\|\boldsymbol{v}\|^2 L^2}{N}\|\boldsymbol{e}_{x,k}\|^2,$$

which establish inequality (15). This completes the proof. $\square$

## B  PROOF FOR CONSENSUAL CONVERGENCE TO A FIRST-ORDER STATIONARY POINT

In this section, we establish the first-order convergence of the proposed algorithm. The proof proceeds in four main steps:

- **Bounding Consensus Errors**: We begin by deriving iterative relations for the optimization variable errors and the gradient tracking errors, specifically $\mathbb{E}[\|\boldsymbol{e}_{x,k+1}\|^2]$ and $\mathbb{E}[\|\boldsymbol{e}_{y,k+1}\|^2]$. The detailed expressions can be found in Section B.1.

- **Function Descent Inequality**: We derive a descent inequality that bounds $\mathbb{E}[\|\nabla F(\bar{\boldsymbol{x}}^k)\|^2]$ using the function value difference and the previously derived error terms. This step is detailed in Section B.2.

- **Convergence Rate Characterization (proof for Theorem 4.1)**: We combine the results from the first two steps into a linear time-invariant (LTI) system that characterizes the convergence behavior. This establishes the algorithm's convergence to a first-order stationary point, as shown in Section B.3.

- **Stochastic Gradient Scenario (proof for Corollary 4.3)**: Finally, we extend the analysis to the stochastic setting, where the gradient tracking update is modified as $\boldsymbol{y}_i^{k+1} = \sum_{j\in\mathcal{N}_i} w_{ij}\boldsymbol{y}_j^k + \nabla f_i(\boldsymbol{x}_i^{k+1}, \boldsymbol{\zeta}_i^{k+1}) - \nabla f_i(\boldsymbol{x}_i^k, \boldsymbol{\zeta}_i^k)$, with $\boldsymbol{\zeta}_i^k$ representing random data samples. This modification introduces slight changes in the analysis, as detailed in Section B.4.

### B.1 Bounding Consensus Errors

We define the optimization error between each agent's local optimization variable and the global average variable $\bar{\boldsymbol{x}}^k$ as

$$\boldsymbol{e}_{x,k} := \boldsymbol{x}^k - \mathbf{1}(\bar{\boldsymbol{x}}^k)^\top. \tag{16}$$

Similarly, we define the gradient tracking error as

$$\boldsymbol{e}_{y,k} := \boldsymbol{y}^k - \boldsymbol{v}(\bar{\boldsymbol{y}}^k)^\top. \tag{17}$$

By construction, the $i$-th row of $\boldsymbol{e}_{x,k}$ and $\boldsymbol{e}_{y,k}$, denoted $\boldsymbol{e}_{x,k,i}$ and $\boldsymbol{e}_{y,k,i}$, satisfy

$$\boldsymbol{e}_{x,k,i} = (\boldsymbol{x}_i^k)^\top - (\bar{\boldsymbol{x}}^k)^\top, \qquad \boldsymbol{e}_{y,k,i} = (\boldsymbol{y}_i^k)^\top - (v_i \bar{\boldsymbol{y}}^k)^\top. \tag{18}$$

Using the update rules in Eq. (3) and Eq. (4), together with the properties $\boldsymbol{R}\mathbf{1} = \mathbf{1}$ and $\mathbf{1}^\top \boldsymbol{C} = \mathbf{1}^\top$, the consensus error dynamics can be expressed as

$$\boldsymbol{e}_{x,k+1} = \left(\boldsymbol{R} - \frac{\mathbf{1}\boldsymbol{u}^\top}{N}\right)\boldsymbol{e}_{x,k} + \theta_k\left(\boldsymbol{I} - \frac{\mathbf{1}\boldsymbol{u}^\top}{N}\right)\boldsymbol{\xi}^k - \left(\boldsymbol{I} - \frac{\mathbf{1}\boldsymbol{u}^\top}{N}\right)\boldsymbol{\alpha}_k \boldsymbol{y}^k, \tag{19}$$

and

$$\boldsymbol{e}_{y,k+1} = \left(\boldsymbol{C} - \frac{\boldsymbol{v}\mathbf{1}^\top}{N}\right)\boldsymbol{e}_{y,k} + \left(\boldsymbol{I} - \frac{\boldsymbol{v}\mathbf{1}^\top}{N}\right)\left(\nabla f(\boldsymbol{x}^{k+1}) - \nabla f(\boldsymbol{x}^k)\right). \tag{20}$$

Based on the dynamics of $\boldsymbol{e}_{x,k}$ and $\boldsymbol{e}_{y,k}$, we can prove that the consensus error $\|\boldsymbol{e}_{x,k}\|^2$ converges to zero (with an error determined by the noise amplitude), and hence, all $\boldsymbol{x}_i^k$ will converge to each other.

**Lemma B.1.** *Under the definitions of the optimization error $\boldsymbol{e}_{x,k}$ in equation (16) and the gradient tracking error $\boldsymbol{e}_{y,k}$ in equation (17), when the stepsize $\alpha$ satisfies*

$$0 < \alpha \leqslant \min\left\{\frac{\sqrt{N(1-\sigma_{\boldsymbol{R}}^2)}}{2\sqrt{3}L\|\boldsymbol{v}\|\|\boldsymbol{I} - \frac{\mathbf{1}\boldsymbol{u}^\top}{N}\|_{\boldsymbol{R}}}, \frac{\sqrt{1-\sigma_C^2}}{12\|\boldsymbol{I} - \frac{\boldsymbol{v}\mathbf{1}^\top}{N}\|_C}\right\},$$

*the following relation holds for all $k \geqslant 0$:*

$$\mathbb{E}\left[\|\boldsymbol{e}_{x,k+1}\|_{\boldsymbol{R}}^2\right] \leqslant \frac{1+\sigma_{\boldsymbol{R}}^2}{2}\mathbb{E}\left[\|\boldsymbol{e}_{x,k}\|_{\boldsymbol{R}}^2\right] + \alpha^2 \mathcal{C}_{x,1}\mathbb{E}\left[\|\boldsymbol{e}_{y,k}\|_C^2\right] + \alpha\, \mathcal{C}_{x,2}\,\bar{\alpha}_k \mathbb{E}\left[\|\nabla F(\bar{\boldsymbol{x}}^k)\|^2\right]$$
$$+ \|\boldsymbol{I} - \frac{\mathbf{1}\boldsymbol{u}^\top}{N}\|_{\boldsymbol{R}}^2\, \theta^2,$$
$$\mathbb{E}\left[\|\boldsymbol{e}_{y,k+1}\|_C^2\right] \leqslant \mathcal{C}_{y,1}\mathbb{E}\left[\|\boldsymbol{e}_{x,k}\|_{\boldsymbol{R}}^2\right] + \frac{1+\sigma_C^2}{2}\mathbb{E}\left[\|\boldsymbol{e}_{y,k}\|_C^2\right] + \alpha\,\mathcal{C}_{y,2}\bar{\alpha}_k \mathbb{E}\left[\|\nabla F(\bar{\boldsymbol{x}}^k)\|^2\right] \tag{21}$$
$$+ \mathcal{C}_{y,3}\theta^2,$$

*where*

$$\mathcal{C}_{x,1} = \frac{12(1+\sigma_{\boldsymbol{R}}^2)\|\boldsymbol{I} - \frac{\mathbf{1}\boldsymbol{u}^\top}{N}\|_{\boldsymbol{R}}^2 \delta_{R,2}^2}{1-\sigma_{\boldsymbol{R}}^2},$$

$$\mathcal{C}_{x,2} = \frac{3N(1+\sigma_{\boldsymbol{R}}^2)\|\boldsymbol{I} - \frac{\mathbf{1}\boldsymbol{u}^\top}{N}\|_{\boldsymbol{R}}^2 \delta_{R,2}^2 \|\boldsymbol{v}\|^2}{1-\sigma_{\boldsymbol{R}}^2},$$

$$\mathcal{C}_{y,1} = \frac{2(1+\sigma_C^2)(1+2\sigma_{\boldsymbol{R}}^2)\|\boldsymbol{I} - \frac{\boldsymbol{v}\mathbf{1}^\top}{N}\|_C^2 L^2}{1-\sigma_C^2}, \tag{22}$$

$$\mathcal{C}_{y,2} = \frac{12N(1+\sigma_C^2)\|\boldsymbol{I} - \frac{\boldsymbol{v}\mathbf{1}^\top}{N}\|_C^2 \|\boldsymbol{v}\|^2 L^2 \bar{\alpha}_k}{1-\sigma_C^2},$$

$$\mathcal{C}_{y,3} = \frac{2L^2(1+\sigma_C^2)}{1-\sigma_C^2}\|\boldsymbol{I} - \frac{\boldsymbol{v}\mathbf{1}^\top}{N}\|_C^2.$$

*Proof.* For the dynamics of the optimization error $\boldsymbol{e}_{x,k}$ in (19), taking normalization and expectation on the both sides yields

$$\mathbb{E}\left[\|\boldsymbol{e}_{x,k+1}\|_{\boldsymbol{R}}^2\right] = \mathbb{E}\left[\|\left(\boldsymbol{R} - \frac{\mathbf{1}\boldsymbol{u}^\top}{N}\right)\boldsymbol{e}_{x,k} - \left(\boldsymbol{I} - \frac{\mathbf{1}\boldsymbol{u}^\top}{N}\right)\boldsymbol{\alpha}_k \boldsymbol{y}^k\|_{\boldsymbol{R}}^2\right] + \mathbb{E}\left[\|\theta_k\left(\boldsymbol{I} - \frac{\mathbf{1}\boldsymbol{u}^\top}{N}\right)\boldsymbol{\xi}^k\|_{\boldsymbol{R}}^2\right]$$
$$\leqslant (1+\eta)\sigma_{\boldsymbol{R}}^2 \mathbb{E}\left[\|\boldsymbol{e}_{x,k}\|_{\boldsymbol{R}}^2\right] + \left(1+\frac{1}{\eta}\right)\|\boldsymbol{I} - \frac{\mathbf{1}\boldsymbol{u}^\top}{N}\|_{\boldsymbol{R}}^2 \delta_{R,2}^2 \|\boldsymbol{\alpha}_k \boldsymbol{y}^k\|^2 + \theta^2\|\boldsymbol{I} - \frac{\mathbf{1}\boldsymbol{u}^\top}{N}\|_{\boldsymbol{R}}^2. \tag{23}$$

Using Lemma A.2, under a step size $\alpha \leqslant \frac{\sqrt{N(1-\sigma_{\boldsymbol{R}}^2)}}{2\sqrt{3}\|\boldsymbol{v}\|\|\boldsymbol{I}-\frac{1\boldsymbol{u}^\top}{N}\|_{\boldsymbol{R}}L}$, setting $\eta = \frac{1-\sigma_{\boldsymbol{R}}^2}{3\sigma_{\boldsymbol{R}}^2}$ in (23) leads to

$$\mathbb{E}\big[\|\boldsymbol{e}_{x,k+1}\|_{\boldsymbol{R}}^2\big]$$

$$\leqslant \frac{1+\sigma_{\boldsymbol{R}}^2}{2}\mathbb{E}\big[\|\boldsymbol{e}_{x,k}\|_{\boldsymbol{R}}^2\big] + \frac{12(1+\sigma_{\boldsymbol{R}}^2)\|\boldsymbol{I}-\frac{1\boldsymbol{u}^\top}{N}\|_{\boldsymbol{R}}^2\delta_{R,2}^2}{1-\sigma_{\boldsymbol{R}}^2}\,\bar{\alpha}_k^2\mathbb{E}\big[\|\boldsymbol{e}_{y,k}\|_{\boldsymbol{C}}^2\big]$$

$$+ \frac{3N(1+\sigma_{\boldsymbol{R}}^2)\|\boldsymbol{I}-\frac{1\boldsymbol{u}^\top}{N}\|_{\boldsymbol{R}}^2\delta_{R,2}^2\|\boldsymbol{v}\|^2}{1-\sigma_{\boldsymbol{R}}^2}\,\bar{\alpha}_k^2\mathbb{E}\big[\|\nabla F(\bar{\boldsymbol{x}}^k)\|^2\big] + \theta^2\|\boldsymbol{I}-\frac{1\boldsymbol{u}^\top}{N}\|_{\boldsymbol{R}}^2. \tag{24}$$

For notational convenience, the inequality (24) can be written compactly as

$$\mathbb{E}\big[\|\boldsymbol{e}_{x,k+1}\|_{\boldsymbol{R}}^2\big] \leqslant \frac{1+\sigma_{\boldsymbol{R}}^2}{2}\mathbb{E}\big[\|\boldsymbol{e}_{x,k}\|_{\boldsymbol{R}}^2\big] + \alpha^2\mathcal{C}_{x,1}\mathbb{E}\big[\|\boldsymbol{e}_{y,k}\|_{\boldsymbol{C}}^2\big] + \alpha\,\mathcal{C}_{x,2}\bar{\alpha}_k\mathbb{E}\big[\|\nabla F(\bar{\boldsymbol{x}}^k)\|^2\big]$$

$$+ \theta^2\|\boldsymbol{I}-\frac{1\boldsymbol{u}^\top}{N}\|_{\boldsymbol{R}}^2, \tag{25}$$

where $\mathcal{C}_{x,1} = \frac{12(1+\sigma_{\boldsymbol{R}}^2)\|\boldsymbol{I}-\frac{1\boldsymbol{u}^\top}{N}\|_{\boldsymbol{R}}^2\delta_{R,2}^2}{1-\sigma_{\boldsymbol{R}}^2}$ and $\mathcal{C}_{x,2} = \frac{3N(1+\sigma_{\boldsymbol{R}}^2)\|\boldsymbol{I}-\frac{1\boldsymbol{u}^\top}{N}\|_{\boldsymbol{R}}^2\delta_{R,2}^2\|\boldsymbol{v}\|^2}{1-\sigma_{\boldsymbol{R}}^2}$.

Next, we analyze the evolution of $\|\boldsymbol{e}_{y,k}\|_{\boldsymbol{C}}^2$. From (20), we have

$$\mathbb{E}\big[\|\boldsymbol{e}_{y,k+1}\|_{\boldsymbol{C}}^2\big] \leqslant (1+\eta)\sigma_C^2\mathbb{E}\big[\|\boldsymbol{e}_{y,k}\|_{\boldsymbol{C}}^2\big] + (1+\tfrac{1}{\eta})\Big\|\boldsymbol{I}-\frac{\boldsymbol{v}1^\top}{N}\Big\|_{\boldsymbol{C}}^2\mathbb{E}\big[\|\nabla f(\boldsymbol{x}^{k+1}) - \nabla f(\boldsymbol{x}^k)\|_{\boldsymbol{C}}^2\big]. \tag{26}$$

For the gradient difference term, using Assumption (2.4), we have

$$\mathbb{E}\big[\|\nabla f(\boldsymbol{x}^{k+1}) - \nabla f(\boldsymbol{x}^k)\|^2\big] \leqslant 2L^2\big(\mathbb{E}\big[\|\boldsymbol{x}^{k+1} - 1\bar{\boldsymbol{x}}^k\|^2\big] + \mathbb{E}\big[\|\boldsymbol{x}^k - 1\bar{\boldsymbol{x}}^k\|^2\big]\big). \tag{27}$$

By Lemma A.2, we can further bound the first term on the right hand side of the preceding inequality as

$$\mathbb{E}\big[\|\boldsymbol{x}^{k+1} - 1\bar{\boldsymbol{x}}^k\|^2\big]$$

$$\leqslant 2\Big\|\boldsymbol{R}-\frac{1\boldsymbol{u}^\top}{N}\Big\|_{\boldsymbol{R}}^2\mathbb{E}\big[\|\boldsymbol{x}^k - 1\bar{\boldsymbol{x}}^k\|_{\boldsymbol{R}}^2\big] + 2\mathbb{E}\big[\|\boldsymbol{\alpha}_k\boldsymbol{y}^k\|^2\big] + \theta^2$$

$$\leqslant \Big(2\sigma_R^2 + \frac{24\bar{\alpha}_k^2\|\boldsymbol{v}\|^2L^2}{N}\Big)\mathbb{E}\big[\|\boldsymbol{e}_{x,k}\|_{\boldsymbol{R}}^2\big] + 24\bar{\alpha}_k^2\mathbb{E}\big[\|\boldsymbol{e}_{y,k}\|_{\boldsymbol{C}}^2\big] + 6N\|\boldsymbol{v}\|^2\bar{\alpha}_k^2\mathbb{E}\big[\|\nabla F(\bar{\boldsymbol{x}}^k)\|^2\big] + \theta^2. \tag{28}$$

Substituting (27) and (28) into (26), and applying $\alpha \leqslant \frac{\sqrt{1-\sigma_C^2}}{12\|\boldsymbol{I}-\frac{\boldsymbol{v}1^\top}{N}\|_C}$, we obtain

$$\mathbb{E}\big[\|\boldsymbol{e}_{y,k+1}\|_{\boldsymbol{C}}^2\big] \leqslant \frac{1+\sigma_C^2}{2}\mathbb{E}\big[\|\boldsymbol{e}_{y,k}\|_{\boldsymbol{C}}^2\big] + \mathcal{C}_{y,1}\mathbb{E}\big[\|\boldsymbol{e}_{x,k}\|_{\boldsymbol{R}}^2\big] + \alpha\,\mathcal{C}_{y,2}\bar{\alpha}_k\mathbb{E}\big[\|\nabla F(\bar{\boldsymbol{x}}^k)\|^2\big], \tag{29}$$

where the constants are given by

$$\mathcal{C}_{y,1} = \frac{2(1+\sigma_C^2)(1+2\sigma_{\boldsymbol{R}}^2)\|\boldsymbol{I}-\frac{\boldsymbol{v}1^\top}{N}\|_C^2L^2}{1-\sigma_C^2}, \tag{30}$$

and

$$\mathcal{C}_{y,2} = \frac{12N(1+\sigma_C^2)\big\|\boldsymbol{I}-\frac{\boldsymbol{v}1^\top}{N}\big\|_C^2\|\boldsymbol{v}\|^2L^2\bar{\alpha}_k}{1-\sigma_C^2}. \tag{31}$$

□

## B.2 FUNCTION DESCENT INEQUALITY

**Lemma B.2.** *Under Assumption 2.1 and Assumption 2.4, the objective function $F(\bar{\boldsymbol{x}}^k)$ in Algorithm 1 satisfies the following inequality:*

$$\frac{\boldsymbol{u}^\top\boldsymbol{v}}{3N}\bar{\alpha}_k\mathbb{E}\big[\|\nabla F(\bar{\boldsymbol{x}}^k)\|^2\big] \leqslant \mathbb{E}\big[F(\bar{\boldsymbol{x}}^k)\big] - \mathbb{E}\big[F(\bar{\boldsymbol{x}}^{k+1})\big]$$

$$+ \Big(\frac{6L\kappa_{uv}\|\boldsymbol{v}\|}{N} + \frac{4\kappa_{uv}^2\|\boldsymbol{v}\|^2}{N\boldsymbol{u}^\top\boldsymbol{v}}\Big)\bar{\alpha}_k\mathbb{E}\big[\|\boldsymbol{e}_{y,k}\|^2\big]$$

$$+ \Big(\frac{6L\kappa_{uv}\|\boldsymbol{v}\|^3}{N} + \frac{4\kappa_{uv}^2\|\boldsymbol{v}\|^4}{N\boldsymbol{u}^\top\boldsymbol{v}}\Big)L^2\bar{\alpha}_k\mathbb{E}\big[\|\boldsymbol{e}_{x,k}\|^2\big] + \frac{L}{2}\theta^2. \tag{32}$$

*Proof.* According to the dynamics of $\bar{\boldsymbol{x}}^k$ in (6), we can expend the function $F(\bar{\boldsymbol{x}}^{k+1})$ as follows:

$$
\begin{aligned}
F(\bar{\boldsymbol{x}}^{k+1}) &\leqslant F(\bar{\boldsymbol{x}}^k) - \left\langle \nabla F(\bar{\boldsymbol{x}}^k), \bar{\boldsymbol{x}}^{k+1} - \bar{\boldsymbol{x}}^k \right\rangle + \frac{L}{2}\|\bar{\boldsymbol{x}}^{k+1} - \bar{\boldsymbol{x}}^k\|^2 \\
&\leqslant F(\bar{\boldsymbol{x}}^k) + \left\langle \nabla F(\bar{\boldsymbol{x}}^k), \theta\bar{\boldsymbol{\xi}}^k - \frac{1}{N}\boldsymbol{u}^\top \boldsymbol{\alpha}_k \boldsymbol{y}^k \right\rangle + \frac{L}{2}\|\frac{1}{N}\boldsymbol{u}^\top \boldsymbol{\alpha}_k \boldsymbol{y}^k\|^2 + \frac{L}{2}\theta^2.
\end{aligned}
\tag{33}
$$

For the inner product term, we have

$$
\begin{aligned}
&\left\langle \nabla F(\bar{\boldsymbol{x}}^k), \frac{1}{N}\boldsymbol{u}^\top \boldsymbol{\alpha}_k \boldsymbol{y}^k \right\rangle \\
&= \left\langle \nabla F(\bar{\boldsymbol{x}}^k), \frac{1}{N}\boldsymbol{u}^\top \left( \boldsymbol{\alpha}_k \boldsymbol{y}^k - \tilde{\boldsymbol{\alpha}}_k \boldsymbol{v}\nabla F(\bar{\boldsymbol{x}}^k) \right) \right\rangle + \left\langle \nabla F(\bar{\boldsymbol{x}}^k), \frac{1}{N}\boldsymbol{u}^\top \tilde{\boldsymbol{\alpha}}_k \boldsymbol{v}\nabla F(\bar{\boldsymbol{x}}^k) \right\rangle,
\end{aligned}
\tag{34}
$$

where $\tilde{\boldsymbol{\alpha}}_k \boldsymbol{v}\nabla F(\bar{\boldsymbol{x}}^k) = [(\bar{\alpha}_1^k v_1 \nabla F(\bar{\boldsymbol{x}}^k))^\top; \cdots ; (\bar{\alpha}_N^k v_N \nabla F(\bar{\boldsymbol{x}}^k))^\top]$. We first analyze the first term $\left\langle \nabla F(\bar{\boldsymbol{x}}^k), \frac{1}{N}\boldsymbol{u}^\top \left( \boldsymbol{\alpha}_k \boldsymbol{y}^k - \tilde{\boldsymbol{\alpha}}_k \boldsymbol{v}\nabla F(\bar{\boldsymbol{x}}^k) \right) \right\rangle$.

$$
\begin{aligned}
&\left\langle \nabla F(\bar{\boldsymbol{x}}^k), \frac{1}{N}\boldsymbol{u}^\top \left( \boldsymbol{\alpha}_k \boldsymbol{y}^k - \tilde{\boldsymbol{\alpha}}_k \boldsymbol{v}\nabla F(\bar{\boldsymbol{x}}^k) \right) \right\rangle \\
&= \left\langle \nabla F(\bar{\boldsymbol{x}}^k), \frac{1}{N}\boldsymbol{u}^\top \left( \boldsymbol{\alpha}_k \boldsymbol{y}^k - \tilde{\boldsymbol{\alpha}}_k \boldsymbol{y}^k \right) \right\rangle + \left\langle \nabla F(\bar{\boldsymbol{x}}^k), \frac{1}{N}\boldsymbol{u}^\top \left( \tilde{\boldsymbol{\alpha}}_k \boldsymbol{y}^k - \tilde{\boldsymbol{\alpha}}_k \boldsymbol{v}\nabla F(\bar{\boldsymbol{x}}^k) \right) \right\rangle.
\end{aligned}
\tag{35}
$$

For the first term in (35), using Lemma A.1, we have

$$
\begin{aligned}
&\left| \left\langle \nabla F(\bar{\boldsymbol{x}}^k), \frac{1}{N}\boldsymbol{u}^\top \left( \boldsymbol{\alpha}_k \boldsymbol{y}^k - \tilde{\boldsymbol{\alpha}}_k \boldsymbol{y}^k \right) \right\rangle \right| \\
&\leqslant \frac{1}{N}\|\nabla F(\bar{\boldsymbol{x}}^k)\| \sum_{i=1}^{N} u_i \left| \alpha_i^k - \bar{\alpha}_i^k \right| \|\boldsymbol{y}_i^k\| \\
&\leqslant \frac{1}{N}\|\nabla F(\bar{\boldsymbol{x}}^k)\| \sum_{i=1}^{N} u_i \bar{\alpha}_i^k \|\boldsymbol{y}_i^k - v_i \nabla F(\bar{\boldsymbol{x}}^k)\| \\
&\leqslant \frac{\|\boldsymbol{v}\|}{N}\bar{\alpha}_k \|\nabla F(\bar{\boldsymbol{x}}^k)\| \sum_{i=1}^{N} \frac{u_i}{v_i} \|\boldsymbol{y}_i^k - v_i \nabla F(\bar{\boldsymbol{x}}^k)\|,
\end{aligned}
\tag{36}
$$

where in the last inequality we have used the relation $\bar{\alpha}_k \leqslant \bar{\alpha}_i^k \leqslant \frac{\|\boldsymbol{v}\|}{v_i}\bar{\alpha}_k$. Denoting $\kappa_{uv} := \sup_i \frac{u_i}{v_i}$, the inequality in (36) can be represented as follows:

$$
\begin{aligned}
&\left| \left\langle \nabla F(\bar{\boldsymbol{x}}^k), \frac{1}{N}\boldsymbol{u}^\top \left( \boldsymbol{\alpha}_k \boldsymbol{y}^k - \tilde{\boldsymbol{\alpha}}_k \boldsymbol{y}^k \right) \right\rangle \right| \\
&\leqslant \frac{\kappa_{uv}\|\boldsymbol{v}\|}{N}\bar{\alpha}_k \|\nabla F(\bar{\boldsymbol{x}}^k)\| \sum_{i=1}^{N} \|\boldsymbol{y}_i^k - v_i \nabla F(\bar{\boldsymbol{x}}^k)\|.
\end{aligned}
\tag{37}
$$

Similarly, for the second term in (35), we have

$$
\begin{aligned}
&\left| \left\langle \nabla F(\bar{\boldsymbol{x}}^k), \frac{1}{N}\boldsymbol{u}^\top \left( \tilde{\boldsymbol{\alpha}}_k \boldsymbol{y}^k - \tilde{\boldsymbol{\alpha}}_k \boldsymbol{v}\nabla F(\bar{\boldsymbol{x}}^k) \right) \right\rangle \right| \\
&= \frac{1}{N}\|\nabla F(\bar{\boldsymbol{x}}^k)\| \sum_{i=1}^{N} u_i \bar{\alpha}_i^k \|\boldsymbol{y}_i^k - v_i \nabla F(\bar{\boldsymbol{x}}^k)\| \\
&\leqslant \frac{\kappa_{uv}\|\boldsymbol{v}\|}{N}\bar{\alpha}_k \|\nabla F(\bar{\boldsymbol{x}}^k)\| \sum_{i=1}^{N} \|\boldsymbol{y}_i^k - v_i \nabla F(\bar{\boldsymbol{x}}^k)\|
\end{aligned}
\tag{38}
$$

Combining (37) and (38), we can bound the inner product term in (33) as follows:

$$
\begin{aligned}
&-\left\langle \nabla F(\bar{\boldsymbol{x}}^k), \frac{1}{N}\boldsymbol{u}^\top \boldsymbol{\alpha}_k \boldsymbol{y}^k \right\rangle \\
&\leqslant -\frac{\boldsymbol{u}^\top \boldsymbol{v}}{2N}\bar{\alpha}_k \|\nabla F(\bar{\boldsymbol{x}}^k)\|^2 + \frac{2\kappa_{uv}^2\|\boldsymbol{v}\|^2}{N\boldsymbol{u}^\top \boldsymbol{v}}\bar{\alpha}_k \|\boldsymbol{y}^k - \boldsymbol{v}\nabla F(\bar{\boldsymbol{x}}^k)\|^2.
\end{aligned}
\tag{39}
$$

We have already known that

$$
\begin{aligned}
\mathbb{E}\left[ \|\boldsymbol{y}^k - \boldsymbol{v}\nabla F(\bar{\boldsymbol{x}}^k)\|^2 \right] &= \mathbb{E}\left[ \|\boldsymbol{y}^k - \boldsymbol{v}\bar{\boldsymbol{y}}^k + \boldsymbol{v}\bar{\boldsymbol{y}}^k - \boldsymbol{v}\nabla F(\bar{\boldsymbol{x}}^k)\|^2 \right] \\
&\leqslant 2\mathbb{E}\left[ \|\boldsymbol{y}^k - \boldsymbol{v}\bar{\boldsymbol{y}}^k\|^2 \right] + \mathbb{E}\left[ \|\boldsymbol{v}\bar{\boldsymbol{y}}^k - \boldsymbol{v}\nabla F(\bar{\boldsymbol{x}}^k)\|^2 \right] \\
&\leqslant 2\mathbb{E}\left[ \|\boldsymbol{e}_{y,k}\|^2 \right] + 2L^2\|\boldsymbol{v}\|^2 \mathbb{E}\left[ \|\boldsymbol{e}_{x,k}\|^2 \right].
\end{aligned}
\tag{40}
$$

For the $\frac{L}{2N^2}\|\boldsymbol{u}^\top\boldsymbol{\alpha}_k\boldsymbol{y}^k\|^2$ term, we can bound it as follows:

$$
\frac{L}{2N^2}\|\boldsymbol{u}^\top\boldsymbol{\alpha}_k\boldsymbol{y}^k\|^2 \leqslant \frac{L}{2N}\sum_{i=1}^N \left\|u_i\alpha_i^k\boldsymbol{y}_i^k\right\|^2
$$

$$
\leqslant \frac{3L}{2N}\Big(\sum_{i=1}^N u_i\left\|\alpha_i^k\boldsymbol{y}_i^k - \bar\alpha_i^k\boldsymbol{y}_i^k\right\|^2 + \sum_{i=1}^N u_i\bar\alpha_i^k\left\|\boldsymbol{y}_i^k - v_i\nabla F(\bar{\boldsymbol{x}}^k)\right\|^2
$$

$$
+ \sum_{i=1}^N \left\|v_i\bar\alpha_i^k\nabla F(\bar{\boldsymbol{x}}^k)\right\|^2\Big)
$$

$$
\leqslant \frac{3L}{2N}\Big(2\bar\alpha_k\sum_{i=1}^N \frac{u_i}{v_i}\|\boldsymbol{v}\|\left\|\boldsymbol{y}_i^k - v_i\nabla F(\bar{\boldsymbol{x}}^k)\right\|^2 + \|\boldsymbol{v}\|\,(\bar\alpha_k)^2\,N\left\|\nabla F(\bar{\boldsymbol{x}}^k)\right\|^2\Big)
$$

$$
\leqslant \frac{3L}{2}\|\boldsymbol{v}\|\,(\bar\alpha_k)^2\left\|\nabla F(\bar{\boldsymbol{x}}^k)\right\|^2 + \frac{6L\|\boldsymbol{v}\|\,\kappa_{uv}}{2N}\bar\alpha_k\left\|\boldsymbol{y}^k - \boldsymbol{v}\nabla F(\bar{\boldsymbol{x}}^k)\right\|^2.
\tag{41}
$$

Combining (39), (40) and (41), we obtain

$$
\frac{\boldsymbol{u}^\top\boldsymbol{v}}{3N}\bar\alpha_k\mathbb{E}\big[\|\nabla F(\bar{\boldsymbol{x}}^k)\|^2\big]
$$

$$
\leqslant \mathbb{E}\big[F(\bar{\boldsymbol{x}}^k)\big] - \mathbb{E}\big[F(\bar{\boldsymbol{x}}^{k+1})\big] + \left(\frac{6L\kappa_{uv}\|\boldsymbol{v}\|}{2N} + \frac{2\kappa_{uv}^2\|\boldsymbol{v}\|^2}{N\boldsymbol{u}^\top\boldsymbol{v}}\right)\bar\alpha_k\mathbb{E}\big[\|\boldsymbol{y}^k - \boldsymbol{v}\nabla F(\bar{\boldsymbol{x}}^k)\|^2\big] + \frac{L}{2}\theta^2.
\tag{42}
$$

Substituting (40) into (42) yields

$$
\frac{\boldsymbol{u}^\top\boldsymbol{v}}{3N}\bar\alpha_k\mathbb{E}\big[\|\nabla F(\bar{\boldsymbol{x}}^k)\|^2\big] \leqslant \mathbb{E}\big[F(\bar{\boldsymbol{x}}^k)\big] - \mathbb{E}\big[F(\bar{\boldsymbol{x}}^{k+1})\big]
$$

$$
+ \left(\frac{6L\kappa_{uv}\|\boldsymbol{v}\|}{N} + \frac{4\kappa_{uv}^2\|\boldsymbol{v}\|^2}{N\boldsymbol{u}^\top\boldsymbol{v}}\right)\bar\alpha_k\mathbb{E}\big[\|\boldsymbol{e}_{y,k}\|^2\big]
$$

$$
+ \left(\frac{6L\kappa_{uv}\|\boldsymbol{v}\|^3}{N} + \frac{4\kappa_{uv}^2\|\boldsymbol{v}\|^4}{N\boldsymbol{u}^\top\boldsymbol{v}}\right)L^2\bar\alpha_k\mathbb{E}\big[\|\boldsymbol{e}_{x,k}\|^2\big] + \frac{L}{2}\theta^2.
\tag{43}
$$

$\square$

### B.3  Proof for Theorem 4.1

In this subsection, we establish the convergence rate of Algorithm 1 by writing the contraction inequalities in (21) and (32) as a linear-time-invariant system.

**Lemma B.3.** *When the stepsize $\alpha$ satisfies*

$$
0 < \alpha \leqslant \min\left\{\frac{\sqrt{N(1-\sigma_R^2)}}{2\sqrt{3}L\|\boldsymbol{v}\|\|\boldsymbol{I}-\frac{\mathbf{1}\boldsymbol{u}^\top}{N}\|_R},\ \frac{\sqrt{1-\sigma_C^2}}{12\|\boldsymbol{I}-\frac{\boldsymbol{v}\mathbf{1}^\top}{N}\|_C},\ \sqrt{\frac{(1-\sigma_R^2)(1-\sigma_C^2)}{8\mathcal{C}_{x,1}\mathcal{C}_{y,1}}},\ \sqrt{\frac{\boldsymbol{u}^\top\boldsymbol{v}(1-\sigma_R^2)}{(36L\kappa_{uv}\|\boldsymbol{v}\|^3 + \frac{24\kappa_{uv}^2\|\boldsymbol{v}\|^4}{\boldsymbol{u}^\top\boldsymbol{v}})4\,\mathcal{C}_{x,2}L^2}}\right\},
$$

*the following relation holds for all $k \geqslant 0$:*

$$
\frac{1}{k}\sum_{s=1}^k \mathbb{E}[\|\boldsymbol{e}_{y,s}\|^2] \leqslant \mathcal{O}\big(\tfrac{N}{k(1-\sigma_R^2)(1-\sigma_C^2)}\big) + \mathcal{O}\big(\tfrac{N\theta^2}{k(1-\sigma_C^2)}\big),
$$

$$
\frac{1}{k}\sum_{s=1}^k \mathbb{E}[\|\boldsymbol{e}_{x,s}\|^2] \leqslant \mathcal{O}\big(\tfrac{N^{3/2}}{k(1-\sigma_R^2)}\big) + \mathcal{O}\big(\tfrac{N^{3/2}\theta^2}{k(1-\sigma_R^2)}\big),
\tag{44}
$$

$$
\frac{1}{k}\sum_{s=1}^k \bar\alpha_s\mathbb{E}[\|\nabla F(\bar{\boldsymbol{x}}^s)\|^2] \leqslant \mathcal{O}\big(\tfrac{N\theta^2}{k}\big) + \mathcal{O}\big(\tfrac{1}{k}\big).
$$

*Proof.* By the relations of $\|e_{x,k+1}\|_R^2$ and $\|e_{y,k+1}\|_C^2$ in Lemma B.1 and Lemma B.2, we have

$$
\begin{bmatrix} \|e_{x,k+1}\|_R^2 \\ \|e_{y,k+1}\|_C^2 \end{bmatrix} \leqslant \begin{bmatrix} \frac{1+\sigma_R^2}{2} & \alpha^2 \mathcal{C}_{x,1} \\ \mathcal{C}_{y,1} & \frac{1+\sigma_C^2}{2} \end{bmatrix} \begin{bmatrix} \|e_{x,k}\|_R^2 \\ \|e_{y,k}\|_C^2 \end{bmatrix} + \begin{bmatrix} \alpha\,\mathcal{C}_{x,2} \\ \alpha\,\mathcal{C}_{y,2} \end{bmatrix} \bar{\alpha}_k \|\nabla F(\bar{\boldsymbol{x}}^k)\|^2
$$
$$
+ \begin{bmatrix} \|\boldsymbol{I} - \frac{\boldsymbol{1}\boldsymbol{u}^\top}{N}\|_R^2 \\ \mathcal{C}_{y,3} \end{bmatrix} \theta^2.
\tag{45}
$$

From Lemma B.1, the evolution of the consensus error satisfies

$$
\boldsymbol{u}_{k+1} \leqslant \boldsymbol{G}\boldsymbol{u}_k + \boldsymbol{b}_k,
\tag{46}
$$

where the vector $\boldsymbol{u}_k$, $\boldsymbol{b}_k$, and the matrix $\boldsymbol{G}$ are given by

$$
\boldsymbol{u}_k = \begin{bmatrix} \|e_{x,k+1}\|_R^2 \\ \|e_{y,k+1}\|_C^2 \end{bmatrix}, \quad \boldsymbol{G} = \begin{bmatrix} \frac{1+\sigma_R^2}{2} & \alpha^2 \mathcal{C}_{x,1} \\ \mathcal{C}_{y,1} & \frac{1+\sigma_C^2}{2} \end{bmatrix}
$$

and

$$
\boldsymbol{b}_k = \begin{bmatrix} \alpha\,\mathcal{C}_{x,2} \\ \alpha\,\mathcal{C}_{y,2} \end{bmatrix} \bar{\alpha}_k \|\nabla F(\bar{\boldsymbol{x}}^k)\|^2 + \begin{bmatrix} \|\boldsymbol{I} - \frac{\boldsymbol{1}\boldsymbol{u}^\top}{N}\|_R^2 \\ \mathcal{C}_{y,3} \end{bmatrix} \theta^2
$$

Under the condition $\alpha \leqslant \sqrt{\frac{\left(1-\sigma_R^2\right)\left(1-\sigma_C^2\right)}{8\mathcal{C}_{x,1}\mathcal{C}_{y,1}}}$, we have $|I_2 - G| \geqslant \frac{\left(1-\sigma_R^2\right)\left(1-\sigma_C^2\right)}{8}$, and hence

$$
(I_2 - G)^{-1} \leqslant \begin{bmatrix} \frac{4}{1-\sigma_R^2} & \frac{8\alpha^2 \mathcal{C}_{x,1}}{\left(1-\sigma_R^2\right)\left(1-\sigma_C^2\right)} \\ \frac{8\mathcal{C}_{y,1}}{\left(1-\sigma_R^2\right)\left(1-\sigma_C^2\right)} & \frac{4}{1-\sigma_C^2} \end{bmatrix}
$$

Recursively applying equation (46) from $k = 0$ to $K$, we obtain

$$
\sum_{k=0}^{K} \boldsymbol{u}_k \leqslant (I_2 - G)^{-1} \boldsymbol{u}_0 + (I_2 - G)^{-1} \sum_{k=0}^{K-1} \boldsymbol{b}_k.
\tag{47}
$$

In light of equation (47), we further compute an entry-wise upper bound on the consensus error and obtain the following result

$$
\sum_{k=0}^{K} \|e_{x,k}\|_R^2 \leqslant \frac{8\alpha^2 \mathcal{C}_{x,1}}{\left(1-\sigma_R^2\right)\left(1-\sigma_C^2\right)} \|e_{y,0}\|_C^2 + \frac{4\alpha\,\mathcal{C}_{x,2}}{1-\sigma_R^2} \sum_{k=0}^{K} \bar{\alpha}_k \|\nabla F(\bar{\boldsymbol{x}}^k)\|^2
$$
$$
+ \frac{4\|\boldsymbol{I} - \frac{\boldsymbol{1}\boldsymbol{u}^\top}{N}\|_R^2}{1-\sigma_R^2} \theta^2,
$$
$$
\tag{48}
$$
$$
\sum_{k=0}^{K} \|e_{y,k}\|_C^2 \leqslant \frac{4}{1-\sigma_C^2} \|e_{y,0}\|_C^2 + \left[\frac{8\alpha\,\mathcal{C}_{x,2}\mathcal{C}_{y,1}}{\left(1-\sigma_R^2\right)\left(1-\sigma_C^2\right)} + \frac{4\alpha\,\mathcal{C}_{y,2}}{1-\sigma_C^2}\right] \sum_{k=0}^{K} \bar{\alpha}_k \|\nabla F(\bar{\boldsymbol{x}}^k)\|^2
$$
$$
+ \left[\frac{8\|\boldsymbol{I} - \frac{\boldsymbol{1}\boldsymbol{u}^\top}{N}\|_R^2 \mathcal{C}_{y,1}}{\left(1-\sigma_R^2\right)\left(1-\sigma_C^2\right)} + \frac{4\mathcal{C}_{y,3}}{1-\sigma_C^2}\right] \theta^2.
$$

Finally, combining equation (32) and equation (48), we obtain the following results under $\alpha \leqslant$

$$\sqrt{\frac{\boldsymbol{u}^\top \boldsymbol{v}\left(1-\sigma_R^2\right)}{(36L\kappa_{uv}\|\boldsymbol{v}\|^3 + \frac{24\kappa_{uv}^2\|\boldsymbol{v}\|^4}{\boldsymbol{u}^\top \boldsymbol{v}})4\,\mathcal{C}_{x,2}L^2}} = \mathcal{O}\left(\sqrt{\frac{\left(1-\sigma_R^2\right)\left(1-\sigma_R^2\right)}{N\left(1+\sigma_R^2\right)}}\right):$$

$$\sum_{k=0}^{K} \mathbb{E}\left[\|\boldsymbol{e}_{x,k}\|_R^2\right] \leqslant \mathcal{R}_{x,1}\left(F(\bar{\boldsymbol{x}}^0) - \bar{f}\right) + \mathcal{R}_{x,2}\mathbb{E}\left[\|\boldsymbol{e}_{y,0}\|_C^2\right] + \mathcal{R}_{x,3}\theta^2,$$

$$\sum_{k=0}^{K} \mathbb{E}\left[\|\boldsymbol{e}_{y,k}\|_C^2\right] \leqslant \mathcal{R}_{y,1}\left(F(\bar{\boldsymbol{x}}^0) - \bar{f}\right) + \mathcal{R}_{y,2}\mathbb{E}\left[\|\boldsymbol{e}_{y,0}\|_C^2\right] + \mathcal{R}_{y,3}\theta^2, \qquad (49)$$

$$\sum_{k=0}^{K-1} \bar{\alpha}_k \mathbb{E}\left[\|\nabla F(\bar{\boldsymbol{x}}^k)\|^2\right] \leqslant \frac{6N}{\boldsymbol{u}^\top \boldsymbol{v}}\left(F(\bar{\boldsymbol{x}}^0) - \bar{f}\right) + \mathcal{R}_{\nabla,1}\mathbb{E}\left[\|\boldsymbol{e}_{y,0}\|_C^2\right] + \mathcal{R}_{\nabla,2}\theta^2.$$

where the constants are given by

$$\mathcal{R}_{\nabla,1} = \frac{6N}{\boldsymbol{u}^\top \boldsymbol{v}}\left[\left(\frac{6L\kappa_{uv}\|\boldsymbol{v}\|}{N} + \frac{4\kappa_{uv}^2\|\boldsymbol{v}\|^2}{N\boldsymbol{u}^\top \boldsymbol{v}}\right)\frac{4\alpha}{1-\sigma_C^2} + \left(\frac{6L\kappa_{uv}\|\boldsymbol{v}\|^3}{N} + \frac{4\kappa_{uv}^2\|\boldsymbol{v}\|^4}{N\boldsymbol{u}^\top \boldsymbol{v}}\right)\frac{8\alpha^5\mathcal{C}_{x,1}L^2}{(1-\sigma_R^2)(1-\sigma_C^2)}\right] = \mathcal{O}\left(\frac{\sqrt{N}}{1-\sigma_C^2}\right)$$

$$\mathcal{R}_{\nabla,2} = \frac{6N}{\boldsymbol{u}^\top \boldsymbol{v}}\left[\left(\frac{6L\kappa_{uv}\|\boldsymbol{v}\|}{N} + \frac{4\kappa_{uv}^2\|\boldsymbol{v}\|^2}{N\boldsymbol{u}^\top \boldsymbol{v}}\right)\left(\frac{8\|\boldsymbol{I}-\frac{1\boldsymbol{u}^\top}{N}\|_R^2\mathcal{C}_{y,1}}{(1-\sigma_R^2)(1-\sigma_C^2)} + \frac{4\mathcal{C}_{y,3}}{1-\sigma_C^2}\right)\alpha \right.$$

$$\left. + \left(\frac{6L\kappa_{uv}\|\boldsymbol{v}\|^3}{N} + \frac{4\kappa_{uv}^2\|\boldsymbol{v}\|^4}{N\boldsymbol{u}^\top \boldsymbol{v}}\right)\frac{4\|\boldsymbol{I}-\frac{1\boldsymbol{u}^\top}{N}\|_R^2L^2}{1-\sigma_R^2}\alpha + \frac{L}{2}\right] = \mathcal{O}\left(\frac{1-\sigma_R^2}{1-\sigma_C^2} + N\right)$$

$$\mathcal{R}_{x,1} = \frac{24N\mathcal{C}_{x,2}}{\boldsymbol{u}^\top \boldsymbol{v}(1-\sigma_R^2)}\alpha = \mathcal{O}\left(\frac{N^{3/2}}{1-\sigma_R^2}\right)$$

$$\mathcal{R}_{x,2} = \frac{4\mathcal{R}_{\nabla,1}\mathcal{C}_{x,2}}{1-\sigma_R^2}\alpha + \frac{8\mathcal{C}_{x,1}}{(1-\sigma_R^2)(1-\sigma_C^2)}\alpha^2 = \mathcal{O}\left(\frac{N}{1-\sigma_R^2}\right)$$

$$\mathcal{R}_{x,3} = \frac{4\mathcal{R}_{\nabla,2}\mathcal{C}_{x,2}}{1-\sigma_R^2}\alpha + \frac{4\|\boldsymbol{I}-\frac{1\boldsymbol{u}^\top}{N}\|_R^2}{1-\sigma_R^2} = \mathcal{O}\left(\frac{\sqrt{N}}{1-\sigma_C^2} + \frac{N^{3/2}}{1-\sigma_R^2}\right)$$

$$\mathcal{R}_{y,1} = \frac{6N}{\boldsymbol{u}^\top \boldsymbol{v}}\left(\frac{8\mathcal{C}_{x,2}\mathcal{C}_{y,1}}{(1-\sigma_R^2)(1-\sigma_C^2)} + \frac{4\mathcal{C}_{y,2}}{1-\sigma_C^2}\right)\alpha = \mathcal{O}\left(\frac{\sqrt{N}}{(1-\sigma_R^2)(1-\sigma_C^2)}\right)$$

$$\mathcal{R}_{y,2} = \mathcal{R}_{\nabla,1}\left(\frac{8\mathcal{C}_{x,2}\mathcal{C}_{y,1}}{(1-\sigma_R^2)(1-\sigma_C^2)} + \frac{4\mathcal{C}_{y,2}}{1-\sigma_C^2}\right)\alpha + \frac{4}{1-\sigma_C^2} = \mathcal{O}\left(\frac{1}{1-\sigma_C^2}\right)$$

$$\mathcal{R}_{y,3} = \mathcal{R}_{\nabla,2}\left(\frac{8\mathcal{C}_{x,2}\mathcal{C}_{y,1}}{(1-\sigma_R^2)(1-\sigma_C^2)} + \frac{4\mathcal{C}_{y,2}}{1-\sigma_C^2}\right)\alpha$$

$$+ \left(\frac{8\|\boldsymbol{I}-\frac{1\boldsymbol{u}^\top}{N}\|_R^2\mathcal{C}_{y,1}}{(1-\sigma_R^2)(1-\sigma_C^2)} + \frac{4\mathcal{C}_{y,3}}{1-\sigma_C^2}\right) = \mathcal{O}\left(\frac{N}{1-\sigma_C^2} + \frac{1}{(1-\sigma_R^2)(1-\sigma_C^2)^2}\right)$$

$\square$

### B.4 PROOF FOR COROLLARY 4.3

*Proof.* Under Assumption 4.2, we can recast the update in Eq. (5) as follows:

$$\boldsymbol{x}^{k+1} = \boldsymbol{R}\boldsymbol{x}^k + \theta_k\boldsymbol{\xi}^k - \boldsymbol{\alpha}_k\boldsymbol{y}^k, \qquad (50)$$

$$\boldsymbol{y}^{k+1} = \boldsymbol{C}\boldsymbol{y}^k + \nabla f\left(\boldsymbol{x}^{k+1}, \boldsymbol{v}^{k+1}\right) - \nabla f\left(\boldsymbol{x}^k, \boldsymbol{v}^k\right), \qquad (51)$$

where

$$\nabla f(\boldsymbol{x}^k, \boldsymbol{v}^k) = \begin{bmatrix} -\nabla f_1^\top(\boldsymbol{x}_1^k, \boldsymbol{v}_1^k)- \\ -\nabla f_2^\top(\boldsymbol{x}_2^k, \boldsymbol{v}_2^k)- \\ \vdots \\ -\nabla f_N^\top(\boldsymbol{x}_N^k, \boldsymbol{v}_N^k)- \end{bmatrix} \in \mathbb{R}^{N \times d}.$$

We will express $\nabla f(\boldsymbol{x}^k, \boldsymbol{v}^k) = \nabla f(\boldsymbol{x}^k) + \boldsymbol{\zeta}^k$. Parallel to the deterministic analysis, we now analyze the consensus error and descent behavior of the proposed algorithm in the stochastic setting.

**Consensus error for $\boldsymbol{x}^k$.** Corresponding to the deterministic gradient case (see Eq. (23)), the consensus error associated with the optimization variable $\boldsymbol{x}^k$ can be bounded as

$$\mathbb{E}\big[\|\boldsymbol{e}_{x,k+1}\|^2\big] \leqslant (1+\eta)\,\sigma_W^2\,\mathbb{E}\big[\|\boldsymbol{e}_{x,k}\|^2\big] + \Big(1+\tfrac{1}{\eta}\Big)\,\mathbb{E}\big[\|\boldsymbol{\alpha}_k\boldsymbol{y}^k - \mathbf{1}\,\overline{\boldsymbol{\alpha}_k\boldsymbol{y}^k}\|^2\big] + \theta_k^2. \tag{52}$$

For the term $\|\boldsymbol{\alpha}_k\boldsymbol{y}^k - \mathbf{1}\,\overline{\boldsymbol{\alpha}_k\boldsymbol{y}^k}\|^2$, using Lemma A.2 and $\bar{\boldsymbol{y}}^k = \nabla f(\boldsymbol{x}^k) + \boldsymbol{\zeta}^k$, we obtain

$$\begin{aligned}
\mathbb{E}\Big[\|\boldsymbol{\alpha}_k\boldsymbol{y}^k - \mathbf{1}\,\overline{\boldsymbol{\alpha}_k\boldsymbol{y}^k}\|^2\Big] &= 6\bar{\alpha}_k^2\,\mathbb{E}\Big[\|\boldsymbol{y}^k - \mathbf{1}\,(\bar{\boldsymbol{y}}^k)^\top\|^2\Big] \\
&\quad + \tfrac{4L^2}{N}\bar{\alpha}_k^2\,\mathbb{E}\Big[\|\boldsymbol{x}^k - \mathbf{1}\,(\bar{\boldsymbol{x}}^k)^\top\|^2\Big] + 4\bar{\alpha}_k^2\,\mathbb{E}\big[\|\boldsymbol{\zeta}^k\|^2\big].
\end{aligned} \tag{53}$$

By combining Eq. (52) and Eq. (53), and choosing $\eta = \frac{1-\sigma_R^2}{3\sigma_R^2}$, we arrive at

$$\begin{aligned}
\mathbb{E}\big[\|\boldsymbol{e}_{x,k+1}\|_{\boldsymbol{R}}^2\big] &\leqslant \tfrac{1+\sigma_R^2}{2}\mathbb{E}\big[\|\boldsymbol{e}_{x,k}\|_{\boldsymbol{R}}^2\big] + \alpha^2\mathcal{C}_{x,1}\mathbb{E}\big[\|\boldsymbol{e}_{y,k}\|_{\boldsymbol{C}}^2\big] + \alpha\,\mathcal{C}_{x,2}\bar{\alpha}_k\mathbb{E}\big[\|\nabla F(\bar{\boldsymbol{x}}^k)\|^2\big] \\
&\quad + \theta^2\|\boldsymbol{I} - \tfrac{\mathbf{1}\boldsymbol{u}^\top}{N}\|_{\boldsymbol{R}}^2 + \alpha^2\mathcal{C}_{x,3}\sigma_\zeta^2,
\end{aligned} \tag{54}$$

where $\mathcal{C}_{x,1} = \frac{12(1+\sigma_R^2)\|\boldsymbol{I}-\frac{\mathbf{1}\boldsymbol{u}^\top}{N}\|_{\boldsymbol{R}}^2\delta_{R,2}^2}{1-\sigma_R^2}$, $\mathcal{C}_{x,2} = \frac{3N(1+\sigma_R^2)\|\boldsymbol{I}-\frac{\mathbf{1}\boldsymbol{u}^\top}{N}\|_{\boldsymbol{R}}^2\delta_{R,2}^2\|\boldsymbol{v}\|^2}{1-\sigma_R^2}$ and $\mathcal{C}_{x,3} = \frac{6(1+2\sigma_R^2)\|\boldsymbol{I}-\frac{\mathbf{1}\boldsymbol{u}^\top}{N}\|_{\boldsymbol{R}}^2\delta_{R,2}^2}{1-\sigma_R^2}$.

**Consensus error for $\boldsymbol{y}^k$.** Similarly, following the derivation of Eq. (26), the consensus error for the gradient-tracking variable $\boldsymbol{y}^k$ in the stochastic gradient scenario satisfies:

$$\mathbb{E}\big[\|\boldsymbol{e}_{y,k+1}\|_{\boldsymbol{C}}^2\big] \leqslant (1+\eta)\sigma_C^2\mathbb{E}\big[\|\boldsymbol{e}_{y,k}\|_{\boldsymbol{C}}^2\big] + (1+\tfrac{1}{\eta})\Big\|\boldsymbol{I} - \tfrac{\boldsymbol{v}\mathbf{1}^\top}{N}\Big\|_{\boldsymbol{C}}^2\mathbb{E}\big[\|\nabla f(\boldsymbol{x}^{k+1}) - \nabla f(\boldsymbol{x}^k)\|_{\boldsymbol{C}}^2\big] \tag{55}$$

By replacing $\nabla f(\boldsymbol{x}^k, \boldsymbol{v}^k)$ with $\nabla f(\boldsymbol{x}^k) + \boldsymbol{\zeta}^k$ and utilizing the gradient noise assumption in Assumption 4.2, we have

$$\mathbb{E}\big[\|\boldsymbol{e}_{y,k+1}\|_{\boldsymbol{C}}^2\big] \leqslant \tfrac{1+\sigma_C^2}{2}\mathbb{E}\big[\|\boldsymbol{e}_{y,k}\|_{\boldsymbol{C}}^2\big] + \mathcal{C}_{y,1}\mathbb{E}\big[\|\boldsymbol{e}_{x,k}\|_{\boldsymbol{R}}^2\big] + \alpha\,\mathcal{C}_{y,2}\bar{\alpha}_k\mathbb{E}\big[\|\nabla F(\bar{\boldsymbol{x}}^k)\|^2\big] + \mathcal{C}_{y,3}\sigma_\zeta^2, \tag{56}$$

where the constants are defined as

$$\mathcal{C}_{y,1} = \frac{2(1+\sigma_C^2)(1+2\sigma_R^2)\|\boldsymbol{I}-\frac{\boldsymbol{v}\mathbf{1}^\top}{N}\|_{\boldsymbol{C}}^2 L^2}{1-\sigma_C^2},$$

$$\mathcal{C}_{y,2} = \frac{12N(1+\sigma_C^2)\|\boldsymbol{I}-\frac{\boldsymbol{v}\mathbf{1}^\top}{N}\|_{\boldsymbol{C}}^2\|\boldsymbol{v}\|^2 L^2\bar{\alpha}_k}{1-\sigma_C^2}, \tag{57}$$

$$\mathcal{C}_{y,3} = \frac{2(1+2\sigma_C^2)}{1-\sigma_C^2}\|\boldsymbol{I} - \tfrac{\boldsymbol{v}\mathbf{1}^\top}{N}\|_{\boldsymbol{C}}^2.$$

**Descent inequality under stochastic gradients.** Corresponding to the deterministic gradient case (see Lemma B.2), the descent inequality in the stochastic case can be derived as follows:

$$\begin{aligned}
\frac{\boldsymbol{u}^\top\boldsymbol{v}}{3N}\bar{\alpha}_k\mathbb{E}\big[\|\nabla F(\bar{\boldsymbol{x}}^k)\|^2\big] &\leqslant \mathbb{E}\big[F(\bar{\boldsymbol{x}}^k)\big] - \mathbb{E}\big[F(\bar{\boldsymbol{x}}^{k+1})\big] \\
&\quad + \left(\frac{6L\kappa_{uv}\|\boldsymbol{v}\|}{N} + \frac{4\kappa_{uv}^2\|\boldsymbol{v}\|^2}{N\boldsymbol{u}^\top\boldsymbol{v}}\right)\bar{\alpha}_k\mathbb{E}\big[\|\boldsymbol{e}_{y,k}\|^2\big] \\
&\quad + \left(\frac{6L\kappa_{uv}\|\boldsymbol{v}\|^3}{N} + \frac{4\kappa_{uv}^2\|\boldsymbol{v}\|^4}{N\boldsymbol{u}^\top\boldsymbol{v}}\right)L^2\bar{\alpha}_k\mathbb{E}\big[\|\boldsymbol{e}_{x,k}\|^2\big] + \frac{L}{2}\theta^2 \\
&\quad + \left(\frac{6L\kappa_{uv}\|\boldsymbol{v}\|^3}{N} + \frac{4\kappa_{uv}^2\|\boldsymbol{v}\|^4}{N\boldsymbol{u}^\top\boldsymbol{v}}\right)\bar{\alpha}_k\sigma_\zeta^2
\end{aligned} \tag{58}$$

**Final bound.** Using an argument similar to that in Theorem 4.1 (see Section B.3), we obtain the result stated in Corollary 4.3. Specifically, under

$$0 < \alpha \leqslant \min\left\{ \frac{\sqrt{N(1-\sigma_R^2)}}{2\sqrt{3}L\|\boldsymbol{v}\|\|\boldsymbol{I}-\frac{1\boldsymbol{u}^\top}{N}\|_R}, \frac{\sqrt{1-\sigma_C^2}}{12\|\boldsymbol{I}-\frac{\boldsymbol{v}\boldsymbol{1}^\top}{N}\|_C}, \sqrt{\frac{(1-\sigma_R^2)(1-\sigma_C^2)}{8\mathcal{C}_{x,1}\mathcal{C}_{y,1}}}, \sqrt{\frac{\boldsymbol{u}^\top\boldsymbol{v}(1-\sigma_R^2)}{(36L\kappa_{uv}\|\boldsymbol{v}\|^3+\frac{24\kappa_{uv}^2\|\boldsymbol{v}\|^4}{\boldsymbol{u}^\top\boldsymbol{v}})4\mathcal{C}_{x,2}L^2}} \right\},$$

we have

$$\frac{1}{k}\sum_{s=1}^{k}\mathbb{E}[\|\boldsymbol{e}_{y,s}\|^2] \leqslant \mathcal{O}\left(\frac{N}{k(1-\sigma_R^2)(1-\sigma_C^2)}\right) + \mathcal{O}\left(\frac{N\theta^2}{k(1-\sigma_C^2)}\right) + \mathcal{O}\left(\frac{1}{1-\sigma_C^2}\sigma_\zeta^2\right),$$

$$\frac{1}{k}\sum_{s=1}^{k}\mathbb{E}[\|\boldsymbol{e}_{x,s}\|^2] \leqslant \mathcal{O}\left(\frac{N^{3/2}}{k(1-\sigma_R^2)}\right) + \mathcal{O}\left(\frac{N^{3/2}\theta^2}{k(1-\sigma_R^2)}\right) + \mathcal{O}\left(\frac{1}{1-\sigma_R^2}\sigma_\zeta^2\right), \tag{59}$$

$$\frac{1}{k}\sum_{s=1}^{k}\bar{\alpha}_s\mathbb{E}[\|\nabla F(\bar{\boldsymbol{x}}^s)\|^2] \leqslant \mathcal{O}\left(\frac{1}{k}\right) + \mathcal{O}\left(\frac{N\theta^2}{k}\right) + \mathcal{O}\left(\sqrt{\frac{N}{(1-\sigma_R^2)(1-\sigma_C^2)^2}}\sigma_\zeta^2\right).$$

$\square$

## C ESCAPING SADDLE POINTS AND CONVERGING TO A SECOND-ORDER STATIONARY POINT

We directly discuss the stochastic gradient scenario under Assumption 4.2. Without loss of generality, denote the stochastic gradient $\nabla f(\boldsymbol{x}^k, \boldsymbol{v}^k)$ as $\nabla f(\boldsymbol{x}^k) + \boldsymbol{\zeta}^k$, where $\boldsymbol{\zeta}^k$ denotes noise in gradient estimation. For the deterministic results in Theorem 4.6, we have $\sigma_{\boldsymbol{\zeta}}^2 = 0$.

Our proof follows the framework of Jin et al. (2021). We first establish Lemma C.3, which shows that if the objective value does not sufficiently decrease within a bounded number of steps, then the iterates remain localized near their initial point. Next, we introduce the coupling sequence technique: by constructing two coupled trajectories and applying Lemma C.3, we show that if both sequences fail to achieve notable function decrease, then within one noise injection cycle they must remain trapped around the saddle region (Lemma C.5). Finally, by analyzing the collective dynamics of all agents near such a saddle point (Lemma C.6), we conclude that without significant descent, the optimization variables of all agents inevitably stay confined around the saddle point during a noise cycle. Finally, by analyzing the collective dynamics of all agents near a saddle point, we conclude that the objective function value will, in expectation, experience a significant decrease within one noise cycle.

### C.1 PROOF FOR ESCAPING SADDLE POINTS

**Theorem C.1** (Escaping Saddle Points). *Let Assumption 2.1, Assumption 2.4, Assumption 3.1, and Assumption 4.2 hold. If $\alpha$, $c_0$, $\theta$, and $Q$ are chosen as specified in Definition C.2, and $\tilde{\boldsymbol{x}}$ is an $\epsilon$-strict saddle point, Algorithm 1 ensures that each noise period ($\mathcal{K}_0$ iterations) reduces the objective function by a substantial amount in expectation.*

*More specifically, in no more than $\mathcal{K}_0 = \mathcal{O}\left(\frac{1}{\epsilon}\right)$ iterations after one noise injection, Algorithm 1 ensures that the objective function has the following significant decrease in expectation:*

$$\mathbb{E}[F(\bar{\boldsymbol{x}}^k)] - F(\tilde{\boldsymbol{x}}) \leqslant -Q \tag{60}$$

*where $Q$ is a constant satisfying $Q = \mathcal{O}\left(\epsilon^2\right)$.*

**Definition C.2** (Parameter Choices). *For Algorithm 1, the parameters are set as*

$$\alpha = c_\alpha\epsilon, \quad \theta = c_\theta\epsilon, \quad c_0 = \frac{1}{c_\alpha}\epsilon, \quad \varphi = c_\varphi\epsilon, \quad \mathcal{K}_0 = \frac{c_{\mathcal{K}_0}}{\alpha}, \quad Q = c_Q\epsilon^2, \tag{61}$$

*where $c_\alpha$, $c_\theta$, $c_\varphi$, $c_{\mathcal{K}_0}$ and $c_Q$ are constants satisfying*

$$\frac{4}{3}c_Q \leqslant c_{\mathcal{K}_0} \leqslant \frac{1}{10L}, \qquad c_\theta \geqslant \frac{4\sqrt{\pi}}{3}c_\varphi, \qquad 50c_{\mathcal{K}_0}c_Q \leqslant c_\varphi^2,$$

$$c_\theta c_\alpha \geqslant \frac{20\sqrt{\pi}\left(1+\frac{1}{2}\sigma_\zeta\right)}{3}c_{\mathcal{K}_0}, \qquad c_\alpha \leqslant \min\left\{ \frac{Nc_{\mathcal{K}_0}}{6\Delta F} + \frac{Nc_{\mathcal{K}_0}^2}{2(9L^2+3)}, \frac{c_\varphi\sqrt{N(1-\sigma_W^2)}}{4\sigma_\zeta\sqrt{6c_{\mathcal{K}_0}b_{23}}}, \frac{1}{16\sigma_\zeta^2} \right\}.$$

To prove Theorem C.1, we analyze the iterative dynamics over $\mathcal{K}_0$ steps following a single noise injection. Without loss of generality, we consider the iterations from $m\mathcal{K}_0$ to $(m+1)\mathcal{K}_0 - 1$ as a representative case in the derivation. For clarity, we enumerate these $\mathcal{K}_0$ iterations as follows:

$$
\begin{cases}
\underline{\boldsymbol{x}}^{m\mathcal{K}_0} = R\boldsymbol{x}^{m\mathcal{K}_0} + \theta\boldsymbol{\xi}^{m\mathcal{K}_0} \\
\boldsymbol{x}^{m\mathcal{K}_0+1} = \underline{\boldsymbol{x}}^{m\mathcal{K}_0} - \boldsymbol{\alpha}_{m\mathcal{K}_0}\boldsymbol{y}^{m\mathcal{K}_0} \\
\boldsymbol{y}^{m\mathcal{K}_0+1} = C\boldsymbol{y}^{m\mathcal{K}_0} + \nabla f\left(\boldsymbol{x}^{m\mathcal{K}_0+1}, \boldsymbol{v}^{m\mathcal{K}_0+1}\right) - \nabla f\left(\boldsymbol{x}^{m\mathcal{K}_0}, \boldsymbol{v}^{m\mathcal{K}_0}\right) \\
\vdots \\
\boldsymbol{x}^{(m+1)\mathcal{K}_0-1} = R\boldsymbol{x}^{(m+1)\mathcal{K}_0-2} - \boldsymbol{\alpha}_{(m+1)\mathcal{K}_0-1}\boldsymbol{y}^{(m+1)\mathcal{K}_0-2} \\
\boldsymbol{y}^{(m+1)\mathcal{K}_0-1} = C\boldsymbol{y}^{(m+1)\mathcal{K}_0-2} + \nabla f\left(\boldsymbol{x}^{(m+1)\mathcal{K}_0-1}, \boldsymbol{v}^{(m+1)\mathcal{K}_0-1}\right) \\
\qquad\qquad - \nabla f\left(\boldsymbol{x}^{(m+1)\mathcal{K}_0-2}, \boldsymbol{v}^{(m+1)\mathcal{K}_0-2}\right),
\end{cases}
\tag{62}
$$

where $m \in \mathbb{Z}^+$.

For the sake of notation simplicity, we use $\bar{\boldsymbol{x}}^0$, $\underline{\bar{\boldsymbol{x}}}^0$, and $\bar{\boldsymbol{x}}^k$ to denote $\bar{\boldsymbol{x}}^{m\mathcal{K}_0}$, $\underline{\bar{\boldsymbol{x}}}^{m\mathcal{K}_0}$, and $\bar{\boldsymbol{x}}^{m\mathcal{K}_0+k}$, where $0 \leqslant k \leqslant \mathcal{K}_0 - 1$. To simplify the derivation, we assume the noise variance $r = 1$. We denote the parameter representing the clipped stepsize for the average of all agents' variables $\bar{\boldsymbol{x}}^s$ as $\bar{\alpha}_s = \alpha c_0 \min\left\{\frac{1}{c_0}, \frac{1}{\|\boldsymbol{v}\|\|\nabla F(\bar{\boldsymbol{x}}^s)\|}\right\} \triangleq \alpha_0 \min\left\{1, \frac{c_0}{\|\boldsymbol{v}\|\|\nabla F(\bar{\boldsymbol{x}}^s)\|}\right\} \in \mathbb{R}$.

In the next lemma, we show that if the objective value does not sufficiently decrease within a bounded number of steps, then the iterates remain localized near their starting point.

**Lemma C.3** (Improve or Localize). *Under Assumption 2.1, Assumption 2.3, Assumption 2.4, Assumption 3.1, and Assumption 4.2, for any fixed $0 \leqslant k \leqslant \mathcal{K}_0 - 1$ gradient tracking iterations in Eq. (62), the proposed algorithm guarantees the following results:*

$$
\begin{aligned}
\frac{\mathbb{E}\left[\left\|\bar{\boldsymbol{x}}^\tau - \bar{\boldsymbol{x}}^0\right\|^2\right]}{\mathcal{K}_0\alpha} \leqslant {}& \mathcal{O}(1) + \mathcal{O}\left(\frac{6\alpha}{\left(1-\sigma_R^2\right)\left(1-\sigma_C^2\right)}\right) + \mathcal{O}\left(\frac{\alpha}{N^{1/2}\left(1-\sigma_R^2\right)}\right) \\
& + \left(\mathcal{O}(N) + \mathcal{O}\left(\frac{\alpha}{\left(1-\sigma_C^2\right)}\right) + \mathcal{O}\left(\frac{N^{1/2}\alpha}{\left(1-\sigma_R^2\right)}\right)\right)\theta^2 \\
& + \left(\mathcal{O}\left(\sqrt{\frac{N}{\left(1-\sigma_R^2\right)\left(1-\sigma_C^2\right)^2}}\right) + \mathcal{O}\left(\frac{\alpha}{N\left(1-\sigma_C^2\right)}\right) + \mathcal{O}\left(\frac{\alpha}{N^2\left(1-\sigma_R^2\right)}\right)\right)\sigma_\zeta^2.
\end{aligned}
\tag{63}
$$

*Proof.* From the dynamics of $\bar{\boldsymbol{x}}$ in Eq. (6) and Lemma A.2, for any $\tau \leqslant k$, we have

$$
\left\|\bar{\boldsymbol{x}}^\tau - \bar{\boldsymbol{x}}^0\right\|^2 = \left\|\sum_{l=1}^{\tau-1}\overline{\boldsymbol{\alpha}_l\boldsymbol{y}^l}\right\|^2 \leqslant k\sum_{l=1}^{\tau-1}\left\|\overline{\boldsymbol{\alpha}_l\boldsymbol{y}^l}\right\|^2.
\tag{64}
$$

In the stochastic gradient case, $\bar{\boldsymbol{y}}^k = \nabla f(\boldsymbol{x}^k) + \boldsymbol{\zeta}^k$. Following an argument similar to that in Lemma A.2, we have

$$
\begin{aligned}
\left\|\overline{\boldsymbol{\alpha}_k\boldsymbol{y}^k}\right\|^2 \leqslant {}& \frac{6\bar{\alpha}_k^2}{N}\sum_{i=1}^N\left\|\boldsymbol{y}_i^k - v_i\nabla F(\bar{\boldsymbol{x}}^k)\right\|^2 + 3\bar{\alpha}_k^2\|\boldsymbol{v}\|^2\left\|\nabla F(\bar{\boldsymbol{x}}^k)\right\|^2 \\
= {}& \frac{6\bar{\alpha}_k^2}{N}\sum_{i=1}^N\left\|\boldsymbol{y}_i^k - v_i\bar{\boldsymbol{y}}^k + v_i\bar{\boldsymbol{y}}^k - v_i\nabla F(\bar{\boldsymbol{x}}^k)\right\|^2 + 3\bar{\alpha}_k^2\|\boldsymbol{v}\|^2\left\|\nabla F(\bar{\boldsymbol{x}}^k)\right\|^2 \\
= {}& \frac{6\bar{\alpha}_k^2}{N}\sum_{i=1}^N\left(\left\|\boldsymbol{y}_i^k - v_i\bar{\boldsymbol{y}}^k\right\|^2 + \left\|v_i\bar{\boldsymbol{y}}^k - v_i\nabla F(\bar{\boldsymbol{x}}^k)\right\|^2\right) + 3\bar{\alpha}_k^2\|\boldsymbol{v}\|^2\left\|\nabla F(\bar{\boldsymbol{x}}^k)\right\|^2 \\
\leqslant {}& \frac{6\bar{\alpha}_k^2}{N}\sum_{i=1}^N\left(\left\|\boldsymbol{y}_i^k - v_i\bar{\boldsymbol{y}}^k\right\|^2 + v_i^2\left\|\nabla f(\boldsymbol{x}^k) - \nabla f(1\bar{\boldsymbol{x}}^k) + \boldsymbol{\zeta}^k\right\|^2\right) \\
& + 3\bar{\alpha}_k^2\|\boldsymbol{v}\|^2\left\|\nabla F(\bar{\boldsymbol{x}}^k)\right\|^2.
\end{aligned}
\tag{65}
$$

Taking expectations on both sides of Eq. (65) and using Theorem 4.2, we obtain

$$
\mathbb{E}\!\left[\|\overline{\boldsymbol{\alpha}_k \boldsymbol{y}^k}\|^2\right] \leqslant 3\bar{\alpha}_k^2 \|\boldsymbol{v}\|^2 \mathbb{E}\!\left[\|\nabla F(\bar{\boldsymbol{x}}^k)\|^2\right] + \tfrac{6}{N}\bar{\alpha}_k^2 \mathbb{E}\!\left[\|\boldsymbol{y}^k - \boldsymbol{v}\bar{\boldsymbol{y}}^k\|^2\right]
$$
$$
+ \tfrac{6L^2\|\boldsymbol{v}\|^2}{N}\bar{\alpha}_k^2 \mathbb{E}\!\left[\|\boldsymbol{x}^k - 1(\bar{\boldsymbol{x}}^k)^\top\|^2\right] + 6\bar{\alpha}_k^2 \|\boldsymbol{v}\|^2 \mathbb{E}\!\left[\|\boldsymbol{\zeta}^k\|^2\right]. \tag{66}
$$

By taking expectations on both sides of Eq. (64), substituting Eq. (66) into it, and applying the result in Eq. (59), we have the following inequality after rearranging like terms:

$$
\mathbb{E}\!\left[\|\bar{\boldsymbol{x}}^\tau - \bar{\boldsymbol{x}}^0\|^2\right] \leqslant 3k \sum_{l=1}^{\tau-1} \bar{\alpha}_l^2 \, \mathbb{E}\!\left[\|\nabla F(\bar{\boldsymbol{x}}^l)\|^2\right] + \frac{6k}{N} \sum_{l=1}^{\tau-1} \bar{\alpha}_l^2 \, \mathbb{E}\!\left[\|\boldsymbol{e}_{y,l}\|^2\right]
$$
$$
+ \frac{6kL^2}{N^2} \sum_{l=1}^{\tau-1} \bar{\alpha}_l^2 \, \mathbb{E}\!\left[\|\boldsymbol{e}_{x,l}\|^2\right] + 6k \sum_{l=1}^{\tau-1} \bar{\alpha}_l^2 \, \mathbb{E}\!\left[\|\bar{\boldsymbol{\zeta}}^k\|^2\right]
$$
$$
\leqslant 3k\alpha\Big(\mathcal{O}(\tfrac{1}{k}) + \mathcal{O}(\tfrac{N\theta^2}{k}) + \mathcal{O}\big(\sqrt{\tfrac{N}{(1-\sigma_R^2)(1-\sigma_C^2)^2}}\sigma_\zeta^2\big)\Big) \tag{67}
$$
$$
+ \frac{6k\alpha^2}{N}\Big(\mathcal{O}(\tfrac{N}{k(1-\sigma_R^2)(1-\sigma_C^2)}) + \mathcal{O}(\tfrac{N\theta^2}{k(1-\sigma_C^2)}) + \mathcal{O}(\tfrac{1}{1-\sigma_C^2}\sigma_\zeta^2)\Big)
$$
$$
+ \frac{6kL^2\alpha^2}{N^2}\Big(\mathcal{O}(\tfrac{N^{3/2}}{k(1-\sigma_R^2)}) + \mathcal{O}(\tfrac{N^{3/2}\theta^2}{k(1-\sigma_R^2)}) + \mathcal{O}(\tfrac{1}{1-\sigma_R^2}\sigma_\zeta^2)\Big).
$$

Therefore, for all $0 \leqslant k \leqslant \mathcal{K}_0 - 1$, we obtain

$$
\frac{\mathbb{E}\!\left[\|\bar{\boldsymbol{x}}^\tau - \bar{\boldsymbol{x}}^0\|^2\right]}{\mathcal{K}_0\alpha} \leqslant \mathcal{O}(1) + \mathcal{O}(\tfrac{6\alpha}{(1-\sigma_R^2)(1-\sigma_C^2)}) + \mathcal{O}(\tfrac{\alpha}{N^{1/2}(1-\sigma_R^2)})
$$
$$
+ \Big(\mathcal{O}(N) + \mathcal{O}(\tfrac{\alpha}{(1-\sigma_C^2)}) + \mathcal{O}(\tfrac{N^{1/2}\alpha}{(1-\sigma_R^2)})\Big)\theta^2 \tag{68}
$$
$$
+ \Big(\mathcal{O}\big(\sqrt{\tfrac{N}{(1-\sigma_R^2)(1-\sigma_C^2)^2}}\big) + \mathcal{O}(\tfrac{\alpha}{N(1-\sigma_C^2)}) + \mathcal{O}(\tfrac{\alpha}{N^2(1-\sigma_R^2)})\Big)\sigma_\zeta^2.
$$

$\square$

Next, we will establish that even when the optimization variables are initialized in the vicinity of a saddle point, Algorithm 1 can still guarantee a substantial decrease in the function value after $\mathcal{K}_0$ iterations when we add the noise at the first iteration. For the convenience of exposition, we denote $\boldsymbol{H} = \nabla^2 F(\bar{\boldsymbol{x}}^0)$, where $\bar{\boldsymbol{x}}^0$ is the saddle point. According to Definition 2 in the main text, we have $-\gamma = \lambda_{\min}(\boldsymbol{H}) < -\sqrt{\rho\epsilon}$. We denote the eigendirection corresponding to the eigenvalue $-\gamma$ as $\boldsymbol{e}_1$, which denotes the direction in which the global function has a negative curvature. We also let $\mathcal{P}_{-1}$ be the projection onto the subspace complement of $\boldsymbol{e}_1$.

**Definition C.4** (Coupling Sequences). *Consider two sequences $\{\boldsymbol{x}^k\}$ and $\{\boldsymbol{x}'^k\}$ that are obtained as separate runs of Algorithm 1, both starting from the same point at time $k = 0$. They are coupled if both sequences share the same randomness $\mathcal{P}_{-1}\bar{\boldsymbol{\xi}}^k$, while in $\boldsymbol{e}_1$ direction we have $\boldsymbol{e}_1^\top\bar{\boldsymbol{\xi}}^k = -\boldsymbol{e}_1^\top\bar{\boldsymbol{\xi}}'^k$.*

By utilizing the definition of "coupling sequences" and leveraging Lemma C.3, we can establish that if the function values of both sequences do not experience a significant decrease, they will stay around the saddle point within one noise period.

**Lemma C.5.** *Under the notation of Theorem B.2 and statements of Lemma C.3, we have the following result for the coupling sequences defined in Theorem C.4:*

$$
\min\left\{\mathbb{E}\!\left[F(\bar{\boldsymbol{x}}^{\mathcal{K}_0-1})\right] - F(\bar{\boldsymbol{x}}^0),\ \mathbb{E}\!\left[F(\bar{\boldsymbol{x}}'^{\mathcal{K}_0-1})\right] - F(\bar{\boldsymbol{x}}^0)\right\} \leqslant -Q,
$$
$$
\text{or}\quad \forall k < \mathcal{K}_0:\ \max\left\{\mathbb{E}\!\left[\|\bar{\boldsymbol{x}}^k - \bar{\boldsymbol{x}}^0\|^2\right],\ \mathbb{E}\!\left[\|\bar{\boldsymbol{x}}'^k - \bar{\boldsymbol{x}}^0\|^2\right]\right\} \leqslant \varphi^2. \tag{69}
$$

*Proof.* By Lemma C.3, the function value satisfies the following relationship with expectation:

$$
\frac{\mathbb{E}\!\left[\|\bar{\boldsymbol{x}}^\tau - \bar{\boldsymbol{x}}^0\|^2\right]}{\mathcal{K}_0\alpha} \leqslant \mathbb{E}\!\left[F(\bar{\boldsymbol{x}}^0)\right] - \mathbb{E}\!\left[F(\bar{\boldsymbol{x}}^{\mathcal{K}_0})\right] + \mathcal{R}_1,
$$

where

$$\mathcal{R}_1 = \left( \mathcal{O}\left( \frac{6\alpha}{\left(1-\sigma_R^2\right)\left(1-\sigma_C^2\right)} \right) + \mathcal{O}\left( \frac{\alpha}{N^{1/2}\left(1-\sigma_R^2\right)} \right) \right) \mathbb{E}\left[ \|\boldsymbol{e}_{y,0}\|^2 \right]$$

$$+ \left( \mathcal{O}(N) + \mathcal{O}\left( \frac{\alpha}{\left(1-\sigma_C^2\right)} \right) + \mathcal{O}\left( \frac{N^{1/2}\alpha}{\left(1-\sigma_R^2\right)} \right) \right) \theta^2$$

$$+ \left( \mathcal{O}\left( \sqrt{ \frac{N}{\left(1-\sigma_R^2\right)\left(1-\sigma_C^2\right)^2} } \right) + \mathcal{O}\left( \frac{\alpha}{N\left(1-\sigma_C^2\right)} \right) + \mathcal{O}\left( \frac{\alpha}{N^2\left(1-\sigma_R^2\right)} \right) \right) \sigma_\zeta^2.$$

If

$$\max\left\{ \mathbb{E}\left[ \|\bar{\boldsymbol{x}}^k - \bar{\boldsymbol{x}}^0\| \right], \ \mathbb{E}\left[ \|\bar{\boldsymbol{x}}'^k - \bar{\boldsymbol{x}}^0\| \right] \right\} \geqslant \varphi,$$

then at least one sequence satisfies

$$\mathbb{E}\left[ F(\bar{\boldsymbol{x}}^0) \right] - \mathbb{E}\left[ F(\bar{\boldsymbol{x}}^{\mathcal{K}_0}) \right] \geqslant \frac{\varphi^2}{\left( 24 + \frac{6\mathcal{C}_{y,1}\alpha}{N} + \frac{6L^2\tilde{\mathcal{C}}_{x,1}\alpha}{N^2} \right)\mathcal{K}_0\alpha} - \frac{\mathcal{R}_1}{24}$$

$$\geqslant \frac{c_\varphi^2}{c_{\mathcal{K}_0}}\epsilon^2 - \tfrac{1}{8}\left( \alpha\tilde{\mathcal{C}}_{\nabla,1} + \theta^2\tilde{\mathcal{C}}_{\nabla,2} \right)$$

$$- \frac{\tilde{\mathcal{C}}_{y,2}\alpha^2}{4N} - \frac{L^2\tilde{\mathcal{C}}_{x,2}\alpha^2}{4N^2} - \frac{\tilde{\mathcal{C}}_{y,3}\alpha\theta^2}{4N} - \frac{L^2\tilde{\mathcal{C}}_{x,3}\alpha\theta^2}{4N^2}$$

$$- \frac{2b_{23}}{N\left(1-\sigma_R^2\right)}\alpha^2\sigma_\zeta^2 \ \geqslant Q,$$

where the second inequality holds due to the results in Eq. (59) and Lemma C.3.

By selecting and rearranging the parameters in Eq. (61), we obtain the third inequality under constants $c_\alpha \leqslant \frac{c_\varphi\sqrt{N(1-\sigma_R^2)}}{4\sigma_\zeta\sqrt{6c_{\mathcal{K}_0}b_{23}}}$ and $c_Q \leqslant \frac{c_\varphi^2}{50c_{\mathcal{K}_0}}$,.

This finishes the proof. $\square$

To analyze the dynamics of all agents' optimization variables around the saddle point, we need to consider the cumulative effects of various algorithm components. These components include disagreements in local gradient estimations, local clipping results, and added noise. Because each agent has distinct local objective functions $f_i(\cdot)$, local decision variables $\boldsymbol{x}_i^k$, and noise $\boldsymbol{\xi}_i^k$, they undergo different update processes. All of these factors collectively influence the dynamics of the difference between coupling sequences.

**Lemma C.6.** *Let $\bar{\boldsymbol{x}}^k$ denote the average of $\{\boldsymbol{x}_i^k\}$ in Eq. (5), and let $\bar{\boldsymbol{x}}'^k$ denote the average of $\{\boldsymbol{x}_i'^k\}$. Define*

$$\hat{\bar{\boldsymbol{x}}}^k \triangleq \bar{\boldsymbol{x}}^k - \bar{\boldsymbol{x}}'^k.$$

*Then, the iterates satisfy*

$$\hat{\bar{\boldsymbol{x}}}^{k+1} = p_q(k) - p_h(k) + p_n(k), \tag{70}$$

*where*

$$p_q(k) \triangleq \theta \boldsymbol{M}_k^k \hat{\bar{\boldsymbol{\xi}}}^0, \ p_n(k) \triangleq \sum_{l=0}^k \boldsymbol{M}_l^k \left( \boldsymbol{p}^{k-l} + \hat{\boldsymbol{q}}^{k-l} \right), \ p_h(k) \triangleq \sum_{l=0}^k \bar{\alpha}_{k-l}\Delta_{k-l}\boldsymbol{M}_l^k \hat{\bar{\boldsymbol{x}}}^{k-l}. \tag{71}$$

$$\hat{\bar{\boldsymbol{\xi}}}^0 = \tfrac{1}{N}\boldsymbol{1}^\top\boldsymbol{\xi}^0 - \tfrac{1}{N}\boldsymbol{1}^\top\boldsymbol{\xi}'^0,$$

$$\boldsymbol{q}^k = \frac{1}{N}\sum_{i=1}^N \alpha_i^k\boldsymbol{y}_i^k - \bar{\alpha}_k v_i \nabla F(\bar{\boldsymbol{x}}^k, \bar{\boldsymbol{v}}^k),$$

$$\Delta_k = \int_0^1 \nabla^2 F\left( \psi\bar{\boldsymbol{x}}^k + (1-\psi)\bar{\boldsymbol{x}}'^k \right) d\psi - \boldsymbol{H},$$

$$\boldsymbol{M}_l^k = \begin{cases} \boldsymbol{I}, & l = 0, \\ \left( \boldsymbol{I} - \bar{\alpha}_k\boldsymbol{H}_0 \right)\cdots\left( \boldsymbol{I} - \bar{\alpha}_{k-l+1}\boldsymbol{H}_0 \right), & l \geq 1, \end{cases}$$

$$
\boldsymbol{p}^k = \begin{cases}
\alpha c_0 \left( \frac{1}{\|\boldsymbol{v}\| \|\nabla F(\bar{\boldsymbol{x}}^k, \bar{\boldsymbol{v}}^k)\|} - \frac{1}{\|\boldsymbol{v}\| \|\nabla F(\bar{\boldsymbol{x}}'^k, \bar{\boldsymbol{v}}'^k)\|} \right) \nabla F(\bar{\boldsymbol{x}}'^k, \bar{\boldsymbol{v}}'^k) \\
\qquad \text{if} \qquad \|\boldsymbol{v}\| \|\nabla F(\bar{\boldsymbol{x}}^k, \bar{\boldsymbol{v}}^k)\| \geq c_0 \quad \text{and} \quad \|\boldsymbol{v}\| \|\nabla F(\bar{\boldsymbol{x}}'^k, \bar{\boldsymbol{v}}'^k)\| \geq c_0, \\[2mm]
\alpha c_0 \left( \frac{1}{\|\boldsymbol{v}\| \|\nabla F(\bar{\boldsymbol{x}}^k, \bar{\boldsymbol{v}}^k)\|} - \frac{1}{c_0} \right) \nabla F(\bar{\boldsymbol{x}}'^k, \bar{\boldsymbol{v}}'^k) \\
\qquad \text{if} \qquad \|\boldsymbol{v}\| \|\nabla F(\bar{\boldsymbol{x}}^k, \bar{\boldsymbol{v}}^k)\| \geq c_0 \quad \text{and} \quad \|\boldsymbol{v}\| \|\nabla F(\bar{\boldsymbol{x}}'^k, \bar{\boldsymbol{v}}'^k)\| \leq c_0, \\[2mm]
\alpha c_0 \left( \frac{1}{c_0} - \frac{1}{\|\boldsymbol{v}\| \|\nabla F(\bar{\boldsymbol{x}}'^k, \bar{\boldsymbol{v}}'^k)\|} \right) \nabla F(\bar{\boldsymbol{x}}'^k, \bar{\boldsymbol{v}}'^k) \\
\qquad \text{if} \qquad \|\boldsymbol{v}\| \|\nabla F(\bar{\boldsymbol{x}}^k, \bar{\boldsymbol{v}}^k)\| \leq c_0 \quad \text{and} \quad \|\boldsymbol{v}\| \|\nabla F(\bar{\boldsymbol{x}}'^k, \bar{\boldsymbol{v}}'^k)\| \geq c_0, \\[2mm]
0 \\
\qquad \text{if} \qquad \|\boldsymbol{v}\| \|\nabla F(\bar{\boldsymbol{x}}^k, \bar{\boldsymbol{v}}^k)\| \leq c_0 \quad \text{and} \quad \|\boldsymbol{v}\| \|\nabla F(\bar{\boldsymbol{x}}'^k, \bar{\boldsymbol{v}}'^k)\| \leq c_0.
\end{cases}
$$

*Proof.* By Eq. (5), we can represent the dynamics as follows:

$$
\begin{aligned}
\bar{\boldsymbol{x}}^{k+1} &= \bar{\boldsymbol{x}}^k - \bar{\alpha}_k \nabla F(\bar{\boldsymbol{x}}^k, \bar{\boldsymbol{v}}^k) - \left( \frac{1}{N} \sum_{i=1}^N \alpha_i^k \boldsymbol{y}_i^k - \bar{\alpha}_k \nabla F(\bar{\boldsymbol{x}}^k, \bar{\boldsymbol{v}}^k) \right) \\
&= \bar{\boldsymbol{x}}^k - \bar{\alpha}_k \nabla F(\bar{\boldsymbol{x}}^k, \bar{\boldsymbol{v}}^k) - \left( \frac{1}{N} \sum_{i=1}^N \left( \alpha_i^k - \bar{\alpha}_k \right) \boldsymbol{y}_i^k + \bar{\alpha}_k \left( \boldsymbol{y}_i^k - \nabla F(\bar{\boldsymbol{x}}^k, \bar{\boldsymbol{v}}^k) \right) \right) \\
&\triangleq \bar{\boldsymbol{x}}^k - \bar{\alpha}_k \nabla F(\bar{\boldsymbol{x}}^k, \bar{\boldsymbol{v}}^k) - \boldsymbol{q}^k,
\end{aligned} \tag{72}
$$

Then, we can derive the following relation for the difference between coupling sequences:

$$
\begin{aligned}
\bar{\boldsymbol{x}}^{k+1} - \bar{\boldsymbol{x}}'^{k+1} &= \bar{\boldsymbol{x}}^k - \bar{\boldsymbol{x}}'^k - \left( \bar{\alpha}_k \nabla F(\bar{\boldsymbol{x}}^k, \bar{\boldsymbol{v}}^k) - \bar{\alpha}_k' \nabla F(\bar{\boldsymbol{x}}'^k, \bar{\boldsymbol{v}}'^k) \right) - \left( \boldsymbol{q}^k - \boldsymbol{q}'^k \right) \\
\hat{\bar{\boldsymbol{x}}}^{k+1} &= \hat{\bar{\boldsymbol{x}}}^k - \left( \bar{\alpha}_k \nabla F(\bar{\boldsymbol{x}}^k, \bar{\boldsymbol{v}}^k) - \bar{\alpha}_k' \nabla F(\bar{\boldsymbol{x}}'^k, \bar{\boldsymbol{v}}'^k) \right) - \hat{\boldsymbol{q}}^k \\
&= \hat{\bar{\boldsymbol{x}}}^k - \bar{\alpha}_k \left( \nabla F(\bar{\boldsymbol{x}}^k, \bar{\boldsymbol{v}}^k) - \nabla F(\bar{\boldsymbol{x}}'^k, \bar{\boldsymbol{v}}'^k) \right) - \boldsymbol{p}^k - \hat{\boldsymbol{q}}^k \\
&= \left( \boldsymbol{I} - \bar{\alpha}_k \boldsymbol{H}_0 \right) \hat{\bar{\boldsymbol{x}}}^k - \bar{\alpha}_k \Delta_t \hat{\bar{\boldsymbol{x}}}^k - \boldsymbol{p}^k - \hat{\boldsymbol{q}}^k,
\end{aligned} \tag{73}
$$

where

$$
\boldsymbol{p}^k = \begin{cases}
\alpha c_0 \left( \frac{1}{\|\boldsymbol{v}\| \|\nabla F(\bar{\boldsymbol{x}}^k, \bar{\boldsymbol{v}}^k)\|} - \frac{1}{\|\boldsymbol{v}\| \|\nabla F(\bar{\boldsymbol{x}}'^k, \bar{\boldsymbol{v}}'^k)\|} \right) \nabla F(\bar{\boldsymbol{x}}'^k, \bar{\boldsymbol{v}}'^k) \\
\qquad \text{if} \qquad \|\boldsymbol{v}\| \|\nabla F(\bar{\boldsymbol{x}}^k, \bar{\boldsymbol{v}}^k)\| \geq c_0 \quad \text{and} \quad \|\boldsymbol{v}\| \|\nabla F(\bar{\boldsymbol{x}}'^k, \bar{\boldsymbol{v}}'^k)\| \geq c_0, \\[2mm]
\alpha c_0 \left( \frac{1}{\|\boldsymbol{v}\| \|\nabla F(\bar{\boldsymbol{x}}^k, \bar{\boldsymbol{v}}^k)\|} - \frac{1}{c_0} \right) \nabla F(\bar{\boldsymbol{x}}'^k, \bar{\boldsymbol{v}}'^k) \\
\qquad \text{if} \qquad \|\boldsymbol{v}\| \|\nabla F(\bar{\boldsymbol{x}}^k, \bar{\boldsymbol{v}}^k)\| \geq c_0 \quad \text{and} \quad \|\boldsymbol{v}\| \|\nabla F(\bar{\boldsymbol{x}}'^k, \bar{\boldsymbol{v}}'^k)\| \leq c_0, \\[2mm]
\alpha c_0 \left( \frac{1}{c_0} - \frac{1}{\|\boldsymbol{v}\| \|\nabla F(\bar{\boldsymbol{x}}'^k, \bar{\boldsymbol{v}}'^k)\|} \right) \nabla F(\bar{\boldsymbol{x}}'^k, \bar{\boldsymbol{v}}'^k) \\
\qquad \text{if} \qquad \|\boldsymbol{v}\| \|\nabla F(\bar{\boldsymbol{x}}^k, \bar{\boldsymbol{v}}^k)\| \leq c_0 \quad \text{and} \quad \|\boldsymbol{v}\| \|\nabla F(\bar{\boldsymbol{x}}'^k, \bar{\boldsymbol{v}}'^k)\| \geq c_0, \\[2mm]
0 \\
\qquad \text{if} \qquad \|\boldsymbol{v}\| \|\nabla F(\bar{\boldsymbol{x}}^k, \bar{\boldsymbol{v}}^k)\| \leq c_0 \quad \text{and} \quad \|\boldsymbol{v}\| \|\nabla F(\bar{\boldsymbol{x}}'^k, \bar{\boldsymbol{v}}'^k)\| \leq c_0.
\end{cases}
$$

Unrolling the recursion yields

$$
\hat{\bar{\boldsymbol{x}}}^{k+1} = \theta \boldsymbol{M}_k^k \hat{\bar{\boldsymbol{\xi}}}^0 - \sum_{l=0}^k \bar{\alpha}_{k-l} \Delta_{k-l} \boldsymbol{M}_l^k \hat{\bar{\boldsymbol{x}}}^{k-l} - \sum_{l=0}^k \boldsymbol{M}_l^k \left( \boldsymbol{p}^{k-l} + \hat{\boldsymbol{q}}^{k-l} \right). \tag{74}
$$

This finishes the proof. $\qquad \square$

Next, we show that the term $p_q(k)$ in Eq. (70) dominates the dynamics of the difference $\bar{\boldsymbol{x}}^k - \bar{\boldsymbol{x}}'^k$. In particular, $p_q(k)$ is comparably larger than both $p_h(k)$ and $p_n(k)$.

**Lemma C.7.** *Under the parameter choices in Eq.* (61) *and the decomposition in Eq.* (70), *the localization condition in Lemma C.5, i.e.,*

$$\forall k < \mathcal{K}_0: \quad \max\left\{\mathbb{E}\left[\|\bar{\boldsymbol{x}}^k - \bar{\boldsymbol{x}}^0\|^2\right], \mathbb{E}\left[\|\bar{\boldsymbol{x}}'^k - \bar{\boldsymbol{x}}^0\|^2\right]\right\} \leqslant \varphi^2,$$

*implies that for all $k' \leqslant k$,*

$$\mathbb{E}[\|p_h(k') - p_n(k')\|] \leqslant \beta_1 a_k^k, \tag{75}$$

*where*

$$\beta_1 = \frac{\theta}{2\sqrt{\pi}} + 5\alpha\mathcal{K}_0 c_0 + \frac{5}{2}\alpha\mathcal{K}_0\sigma_\zeta, \qquad a_l^k = \begin{cases} 1, & l = 0, \\ (1 + \bar{\alpha}_k\gamma)\cdots(1 + \bar{\alpha}_{k-l+1}\gamma), & l > 0. \end{cases}$$

*Proof.* According to the derivation of Lemma C.6, by projecting onto the $\boldsymbol{e}_1$ direction and taking the norm and expectation on both sides, we obtain

$$\begin{aligned}
\mathbb{E}\left[\|\hat{\bar{\boldsymbol{x}}}^{k+1}\|\right] &\geqslant \mathbb{E}\left[\left\|\theta\prod_{l=0}^{k}(1 + \bar{\alpha}_{k-l+1}\gamma)\langle\boldsymbol{e}_1, \hat{\bar{\boldsymbol{\xi}}}^0\rangle\boldsymbol{e}_1\right\|\right] \\
&\quad - \mathbb{E}\left[\left\|\sum_{l=0}^{k}\bar{\alpha}_{k-l}\Delta_{k-l}\boldsymbol{M}_l^k\hat{\bar{\boldsymbol{x}}}^{k-l}\right\|\right] \\
&\quad - \mathbb{E}\left[\left\|\sum_{l=0}^{k}\boldsymbol{M}_l^k\left(\boldsymbol{p}^{k-l} + \hat{\boldsymbol{q}}^{k-l}\right)\right\|\right] \\
&\triangleq \mathbb{E}\left[\|p_q(k)\|\right] - \mathbb{E}\left[\|p_h(k)\|\right] - \mathbb{E}\left[\|p_n(k)\|\right].
\end{aligned} \tag{76}$$

For the first term $\mathbb{E}\left[\|p_q(k)\|\right]$, since the noise added to the coupling sequences follows the Gaussian distribution, we have $\frac{2}{\pi}\mathbb{E}\left[\|p_q(k)\|^2\right] = \mathbb{E}\left[\|p_q(k)\|\right]^2$ and further

$$\mathbb{E}\left[\|p_q(k)\|\right] = \frac{2}{\sqrt{\pi}}\theta\prod_{l=0}^{k}(1 + \bar{\alpha}_{k-l+1}\gamma). \tag{77}$$

For the third term $\mathbb{E}\left[\|p_n(k)\|\right]$, we first prove the following result about the $\boldsymbol{p}^k$:

$$\|\boldsymbol{p}^k\| \leqslant \bar{\alpha}_k L\|\hat{\bar{\boldsymbol{x}}}^k\| + \bar{\alpha}_k\|\bar{\zeta}^k - \bar{\zeta}'^k\|. \tag{78}$$

**Case 1:** $\left\|\nabla F(\bar{\boldsymbol{x}}^k, \bar{\boldsymbol{v}}^k)\right\| > c_0$ and $\left\|\nabla F(\bar{\boldsymbol{x}}'^k, \bar{\boldsymbol{v}}'^k)\right\| \leqslant c_0$:

In this case, $\|\boldsymbol{p}^k\|$ is expressed as follows:

$$\begin{aligned}
\|\boldsymbol{p}^k\| &= \alpha c_0\left(\frac{1}{\|\nabla F(\bar{\boldsymbol{x}}^k, \bar{\boldsymbol{v}}^k)\|} - \frac{1}{c_0}\right)\|\nabla F(\bar{\boldsymbol{x}}'^k, \bar{\boldsymbol{v}}'^k)\| \\
&= \alpha c_0\frac{c_0 - \|\nabla F(\bar{\boldsymbol{x}}^k, \bar{\boldsymbol{v}}^k)\|}{\|\nabla F(\bar{\boldsymbol{x}}^k, \bar{\boldsymbol{v}}^k)\|c_0}\|\nabla F(\bar{\boldsymbol{x}}'^k, \bar{\boldsymbol{v}}'^k)\| \\
&\leqslant \alpha c_0\frac{\|\nabla F(\bar{\boldsymbol{x}}^k, \bar{\boldsymbol{v}}^k)\| - \|\nabla F(\bar{\boldsymbol{x}}'^k, \bar{\boldsymbol{v}}'^k)\|}{\|\nabla F(\bar{\boldsymbol{x}}^k, \bar{\boldsymbol{v}}^k)\|},
\end{aligned} \tag{79}$$

where the third inequality used the condition $\left\|\nabla F(\bar{\boldsymbol{x}}'^k, \bar{\boldsymbol{v}}'^k)\right\| \leqslant c_0$.

According to the triangle inequality $\|\|a\| - \|b\|\| \leqslant \|a - b\|$ and the Lipschitz condition in Theorem 2.4, we have

$$\begin{aligned}
\|\boldsymbol{p}^k\| &\leqslant \bar{\alpha}_k\left\|\nabla F(\bar{\boldsymbol{x}}^k, \bar{\boldsymbol{v}}^k) - \nabla F(\bar{\boldsymbol{x}}'^k, \bar{\boldsymbol{v}}'^k)\right\| \\
&\leqslant \bar{\alpha}_k\left\|\nabla F(\bar{\boldsymbol{x}}^k) - \nabla F(\bar{\boldsymbol{x}}'^k) + \bar{\boldsymbol{v}}^k - \bar{\zeta}'^k\right\| \\
&\leqslant \bar{\alpha}_k L\|\hat{\bar{\boldsymbol{x}}}^k\| + \bar{\alpha}_k\|\bar{\zeta}^k - \bar{\zeta}'^k\|,
\end{aligned} \tag{80}$$

**Case 2:** $\left\|\nabla F(\bar{\boldsymbol{x}}^k, \bar{\boldsymbol{v}}^k)\right\| \leqslant c_0$ and $\left\|\nabla F(\bar{\boldsymbol{x}}'^k, \bar{\boldsymbol{v}}'^k)\right\| > c_0$:

In this case, with a derivation similar to that of the previous case, we obtain

$$
\begin{aligned}
\left\|\boldsymbol{p}^k\right\| &= \alpha c_0 \left(\frac{1}{c_0} - \frac{1}{\|\nabla F(\bar{\boldsymbol{x}}'^k, \bar{\boldsymbol{v}}'^k)\|}\right) \left\|\nabla F(\bar{\boldsymbol{x}}'^k, \bar{\boldsymbol{v}}'^k)\right\| \\
&= \alpha c_0 \frac{\left\|\nabla F(\bar{\boldsymbol{x}}'^k, \bar{\boldsymbol{v}}'^k)\right\| - c_0}{\|\nabla F(\bar{\boldsymbol{x}}'^k, \bar{\boldsymbol{v}}'^k)\| c_0} \left\|\nabla F(\bar{\boldsymbol{x}}'^k, \bar{\boldsymbol{v}}'^k)\right\| \\
&\leqslant \alpha \left(\left\|\nabla F(\bar{\boldsymbol{x}}'^k, \bar{\boldsymbol{v}}'^k)\right\| - \left\|\nabla F(\bar{\boldsymbol{x}}^k, \bar{\boldsymbol{v}}^k)\right\|\right) \leqslant \bar{\alpha}_k L \|\hat{\bar{\boldsymbol{x}}}^k\| + \bar{\alpha}_k \left\|\bar{\boldsymbol{\zeta}}^k - \bar{\boldsymbol{\zeta}}'^k\right\|,
\end{aligned}
\tag{81}
$$

**Case 3:** $\left\|\nabla F(\bar{\boldsymbol{x}}^k, \bar{\boldsymbol{v}}^k)\right\| > c_0$ and $\left\|\nabla F(\bar{\boldsymbol{x}}'^k, \bar{\boldsymbol{v}}'^k)\right\| > c_0$:

In this case, $\left\|\boldsymbol{p}^k\right\|$ can be expressed as follows, and using a derivation similar to that of the first case, we obtain

$$
\begin{aligned}
\left\|\boldsymbol{p}^k\right\| &= \alpha c_0 \left(\frac{1}{\|\nabla F(\bar{\boldsymbol{x}}^k, \bar{\boldsymbol{v}}^k)\|} - \frac{1}{\|\nabla F(\bar{\boldsymbol{x}}'^k, \bar{\boldsymbol{v}}'^k)\|}\right) \nabla F(\bar{\boldsymbol{x}}'^k, \bar{\boldsymbol{v}}'^k) \\
&= \alpha c_0 \frac{\left\|\|\nabla F(\bar{\boldsymbol{x}}'^k, \bar{\boldsymbol{v}}'^k)\| - \|\nabla F(\bar{\boldsymbol{x}}^k, \bar{\boldsymbol{v}}^k)\|\right\|}{\|\nabla F(\bar{\boldsymbol{x}}^k, \bar{\boldsymbol{v}}^k)\| \|\nabla F(\bar{\boldsymbol{x}}'^k, \bar{\boldsymbol{v}}'^k)\|} \left\|\nabla F(\bar{\boldsymbol{x}}'^k, \bar{\boldsymbol{v}}'^k)\right\| \\
&= \frac{\alpha c_0}{\|\nabla F(\bar{\boldsymbol{x}}^k, \bar{\boldsymbol{v}}^k)\|} \left|\left\|\nabla F(\bar{\boldsymbol{x}}'^k, \bar{\boldsymbol{v}}'^k)\right\| - \left\|\nabla F(\bar{\boldsymbol{x}}^k, \bar{\boldsymbol{v}}^k)\right\|\right| \\
&\leqslant \bar{\alpha}_k L \|\hat{\bar{\boldsymbol{x}}}^k\| + \bar{\alpha}_k \left\|\bar{\boldsymbol{\zeta}}^k - \bar{\boldsymbol{\zeta}}'^k\right\|,
\end{aligned}
\tag{82}
$$

**Case 4:** $\left\|\nabla F(\bar{\boldsymbol{x}}^k, \bar{\boldsymbol{v}}^k)\right\| \leqslant c_0$ and $\left\|\nabla F(\bar{\boldsymbol{x}}'^k, \bar{\boldsymbol{v}}'^k)\right\| \leqslant c_0$:

In this case, we already have

$$
\boldsymbol{p}^k = 0.
\tag{83}
$$

Next, using Eq. (72) and the definition of $\alpha_i^k$ and $\bar{\alpha}_k$, we obtain

$$
\left\|\boldsymbol{q}^k\right\| \leqslant \frac{1}{N} \sum_{i=1}^{N} \|\alpha_i^k \boldsymbol{y}_i^k\| + \|\bar{\alpha}_k \nabla F(\bar{\boldsymbol{x}}^k)\| \leqslant 2\alpha c_0.
\tag{84}
$$

Then, by taking expectation on both sides and combining Eq. (84) and Eq. (78), we have the following inequality about the third term $\mathbb{E}\left[\|p_n(k)\|\right]$:

$$
\begin{aligned}
\mathbb{E}\left[\left\|\sum_{l=0}^{k} \boldsymbol{M}_l^k \left(\boldsymbol{p}^{k-l} + \hat{\boldsymbol{q}}^{k-l}\right)\right\|\right] &\leqslant \sum_{l=0}^{k} a_l^k \left(\mathbb{E}\left[\|\boldsymbol{p}^{k-l}\|\right] + \mathbb{E}\left[\|\hat{\boldsymbol{q}}^{k-l}\|\right]\right) \\
&\leqslant \sum_{l=0}^{k} a_l^k \left(\bar{\alpha}_{k-l} L \mathbb{E}[\|\hat{\bar{\boldsymbol{x}}}^{k-l}\|] + 2\bar{\alpha}_{k-l}\sigma_\zeta + 4\alpha c_0\right) \\
&\leqslant L \sum_{l=0}^{k} \bar{\alpha}_{k-l} a_l^k \mathbb{E}[\|\hat{\bar{\boldsymbol{x}}}^{k-l}\|] + (4\alpha c_0 + 2\alpha\sigma_\zeta) \sum_{l=0}^{k} a_l^k,
\end{aligned}
\tag{85}
$$

where $a_l^k = \begin{cases} 1, & l = 0 \\ (1 + \bar{\alpha}_k \gamma) \cdots (1 + \bar{\alpha}_{k-l+1} \gamma), & l > 0 \end{cases}$.

Based on the results in (77) and (85), we employ induction.

Clearly, for the base case $k = 0$, the claim holds trivially, as $p_h(0) = p_n(0) = 0$.

Suppose that the claim holds for up to iteration $k$, then for any $\tau \leqslant k$, we have :

$$
\left\|\hat{\bar{\boldsymbol{x}}}^\tau\right\| \leqslant \|p_q(\tau)\| + \|p_h(\tau) - p_n(\tau)\| \leqslant \frac{2\theta}{\sqrt{\pi}} a_\tau^\tau + \beta_1 a_\tau^\tau.
\tag{86}
$$

Since $\{\forall k < \mathcal{K}_0 : \max\{\left\|\bar{\boldsymbol{x}}^k - \bar{\boldsymbol{x}}^0\right\|^2, \left\|\bar{\boldsymbol{x}}'^k - \bar{\boldsymbol{x}}^0\right\|^2\} \leqslant \varphi^2\}$ implies that the system is still in the vicinity of the saddle point, by the Hessian Lipschitz property, we have $\|\Delta_k\| = \left\|\int_0^1 \nabla^2 F\left(\psi \bar{\boldsymbol{x}}^k + (1 - \psi) \bar{\boldsymbol{x}}'^k\right) d\psi - \boldsymbol{H}\right\| \leqslant \rho \max\left\{\left\|\bar{\boldsymbol{x}}^k - \bar{\boldsymbol{x}}^0\right\|, \left\|\bar{\boldsymbol{x}}'^k - \bar{\boldsymbol{x}}^0\right\|\right\} \leqslant \rho\varphi$.

Furthermore, we can obtain the following inequality:

$$\|p_h(k) - p_n(k)\| \leqslant \|p_h(k)\| + \|p_n(k)\|$$

$$\leqslant (\rho\varphi + L) \sum_{l=0}^{k} \bar{\alpha}_{k-l} a_l^k \mathbb{E}[\|\hat{\bar{\boldsymbol{x}}}^{k-l}\|] + (4\alpha c_0 + 2\alpha\sigma_\zeta) \sum_{l=0}^{k} a_l^k$$

$$\leqslant (\rho\varphi + L) \, \alpha(\frac{2\theta}{\sqrt{\pi}} + \beta_1) \sum_{l=0}^{k} a_l^k a_l^l + (4\alpha c_0 + 2\alpha\sigma_\zeta) \sum_{l=0}^{k} a_l^k$$

$$\leqslant (\rho\varphi + L) \, (\frac{2\theta}{\sqrt{\pi}} + \beta_1)\alpha\mathcal{K}_0 a_k^k + (4\alpha c_0 + 2\alpha\sigma_\zeta)\mathcal{K}_0 a_k^k \qquad (87)$$

$$\leqslant \frac{(\rho\varphi + L)}{10L}(\frac{2\theta}{\sqrt{\pi}} + \beta_1)a_k^k + (4\alpha c_0 + 2\alpha\sigma_\zeta)\mathcal{K}_0 a_k^k$$

$$\leqslant \frac{1}{5}(\frac{2\theta}{\sqrt{\pi}} + \beta_1)a_k^k + (4\alpha c_0 + 2\alpha\sigma_\zeta)\mathcal{K}_0 a_k^k$$

$$\leqslant \beta_1 a_{\mathcal{K}_0}^{\mathcal{K}_0}.$$

where the fifth inequality used the condition $\alpha\mathcal{K}_0 L \leqslant \frac{1}{10}$ and the sixth inequality used the relation $L \geqslant \sqrt{\rho\epsilon}$. The relation $L \geqslant \sqrt{\rho\epsilon}$ holds because otherwise (i.e., under the condition $L < \sqrt{\rho\epsilon}$), $\epsilon-$second-order stationary points are equivalent to $\epsilon-$first-order stationary points since the function $F(\cdot)$ is $L$-gradient Lipschitz.

Hence, the claim will hold for iteration $k + 1$, which completes the proof. $\qquad\square$

### C.1.1 PROOF OF THEOREM 4.4 (THEOREM C.1)

*Proof.* We prove the result by employing a contradiction argument. Suppose to the contrary that both sequences are stuck in the saddle point, i.e., $\min\{F(\bar{\boldsymbol{x}}^{\mathcal{K}_0}) - F(\bar{\boldsymbol{x}}^0), F(\bar{\boldsymbol{x}}'^{\mathcal{K}_0}) - F(\bar{\boldsymbol{x}}^0)\} \geqslant -Q$ holds, then we have

$$\mathbb{E}\left[\|p_q(k+1)\|\right] = \frac{2}{\sqrt{\pi}}\theta a_{\mathcal{K}_0}^{\mathcal{K}_0},$$

$$\mathbb{E}\left[\|p_h(\mathcal{K}_0) - p_n(\mathcal{K}_0)\|\right] \leqslant (\frac{\theta}{2\sqrt{\pi}} + 5\alpha\mathcal{K}_0 c_0 + \frac{5}{2}\alpha c_0\mathcal{K}_0\sigma_\zeta)a_{\mathcal{K}_0}^{\mathcal{K}_0}. \qquad (88)$$

Thus, in the vicinity of the saddle point, the two sequences satisfy the following relationship in expectation

$$\mathbb{E}[\|\hat{\bar{\boldsymbol{x}}}^{\mathcal{K}_0}\|] \geqslant \mathbb{E}\left[\|p_q(\mathcal{K}_0)\|\right] - \mathbb{E}\left[\|p_h(\mathcal{K}_0)\|\right] - \mathbb{E}\left[\|p_n(\mathcal{K}_0)\|\right]$$

$$\geqslant \left[\frac{2}{\sqrt{\pi}}\theta - (\frac{\theta}{2\sqrt{\pi}} + 5\alpha\mathcal{K}_0 c_0)\right] a_{\mathcal{K}_0}^{\mathcal{K}_0} \qquad (89)$$

$$\geqslant \left(\frac{3}{2\sqrt{\pi}}c_\theta - \frac{5(1 + \frac{1}{2}\sigma_\zeta)c_{\mathcal{K}_0}}{c_\alpha}\right) a_{\mathcal{K}_0}^{\mathcal{K}_0}\epsilon \geqslant \varphi,$$

where the last inequality holds under $c_\theta c_\alpha \geqslant \frac{20\sqrt{\pi}(1 + \frac{1}{2}\sigma_\zeta)}{3}c_{\mathcal{K}_0}$ and $c_\theta \geqslant \frac{4\sqrt{\pi}}{3}c_\varphi$. This contradicts Lemma C.5, and hence, we have the second claim in Theorem 4.4. $\qquad\square$

### C.2 PROOF FOR CONVERGING TO A SECOND-ORDER STATIONARY POINT

To establish that the proposed Algorithm 1 converges to an $\epsilon-$second-order stationary point, we first analyze the function's descent behavior when the gradient magnitude is large.

**Lemma C.8** (Descent Lemma with Large Gradient). *Under Assumption 2.1, Assumption 2.3, Assumption 2.4, Assumption 3.1, and Assumption 4.2, in the proposed Algorithm 1, for any $\epsilon > 0$, after $\mathcal{K}_0 = \mathcal{O}\left(\frac{1}{\epsilon}\right)$ iterations, the following inequality holds when $\|\nabla F(\bar{\boldsymbol{x}}^l)\| > \epsilon$ after one noise injection:*

$$\mathbb{E}\left[F\left(\bar{\boldsymbol{x}}^{(m+1)\mathcal{K}_0-1}\right)\right] - \mathbb{E}\left[F\left(\bar{\boldsymbol{x}}^{m\mathcal{K}_0}\right)\right] \leqslant -\frac{c_{\mathcal{K}_0}}{8}\epsilon^2. \qquad (90)$$

*Proof.* According to the proof for Corollary 4.3 in Section B.4, we have the following relation:

$$\frac{\boldsymbol{u}^{\top}\boldsymbol{v}}{3N}\bar{\alpha}_k\mathbb{E}\big[\|\nabla F(\bar{\boldsymbol{x}}^k)\|^2\big] \leqslant \mathbb{E}\big[F(\bar{\boldsymbol{x}}^k)\big] - \mathbb{E}\big[F(\bar{\boldsymbol{x}}^{k+1})\big]$$

$$+ \left(\frac{6L\kappa_{uv}\|\boldsymbol{v}\|}{N} + \frac{4\kappa_{uv}^2\|\boldsymbol{v}\|^2}{N\boldsymbol{u}^{\top}\boldsymbol{v}}\right)\bar{\alpha}_k\mathbb{E}\big[\|\boldsymbol{e}_{y,k}\|^2\big]$$

$$+ \left(\frac{6L\kappa_{uv}\|\boldsymbol{v}\|^3}{N} + \frac{4\kappa_{uv}^2\|\boldsymbol{v}\|^4}{N\boldsymbol{u}^{\top}\boldsymbol{v}}\right)L^2\bar{\alpha}_k\mathbb{E}\big[\|\boldsymbol{e}_{x,k}\|^2\big] + \frac{L}{2}\theta^2 \qquad (91)$$

$$+ \left(\frac{6L\kappa_{uv}\|\boldsymbol{v}\|^3}{N} + \frac{4\kappa_{uv}^2\|\boldsymbol{v}\|^4}{N\boldsymbol{u}^{\top}\boldsymbol{v}}\right)\bar{\alpha}_k\sigma_\zeta^2.$$

Combining the results in Eq. (59) and rearranging like terms, we arrive at the results under the conditions $c_\alpha \leqslant \frac{Nc_{\mathcal{K}_0}}{6\Delta F} + \frac{Nc_{\mathcal{K}_0}^2}{2(9L^2+3)}$ and $c_\alpha \leqslant \frac{1}{16\sigma_\zeta^2}$.

$\square$

### C.2.1 PROOF FOR THEOREM 4.7 (THEOREM 4.6)

*Proof.* First, we split all iterations into each noise cycle ($\mathcal{K}_0$ iterations after noise injection) and assume that the total number of the noise injections is $I$. In this case, we have the total iterations $K$ to converge to second-order stationary point as follows:

$$K = 10I\mathcal{K}_0 = 10\mathcal{K}_0 \max\left\{\frac{f_0 - \bar{f}}{Q}, \frac{8(f_0 - \bar{f})}{c_{\mathcal{K}_0}\epsilon^2}\right\}. \qquad (92)$$

Next, we will show that we have the following two claims simultaneously:

1. In at most $I/4$ noise cycles, the optimization variables are close to saddle points;

2. In at most $I/4$ noise cycles, the gradients are large.

We first prove the first claim.

By Theorem C.1, we know that when all agents encounter the saddle point, they can escape from it within one noise cycle and have the objective function reduced by a substantial amount $Q$ in expectation. Define the stopping times as follows:

$$z_1 = \inf\left\{m \,|\, \|\nabla F(\bar{\boldsymbol{x}}^{m\mathcal{K}_0})\| \leqslant \epsilon \text{ and } \lambda_{\min}(\nabla^2 F(\bar{\boldsymbol{x}}^{m\mathcal{K}_0})) \leqslant -\sqrt{\rho\epsilon}\right\},$$

$$z_j = \inf\left\{m \,|\, m > z_{j-1} \text{ and } \|\nabla F(\bar{\boldsymbol{x}}^{m\mathcal{K}_0})\| \leqslant \epsilon \text{ and } \lambda_{\min}(\nabla^2 F(\bar{x}^{m\mathcal{K}_0})) \leqslant -\sqrt{\rho\epsilon}\right\}, \quad \forall j > 1. \qquad (93)$$

It is obvious that $z_j$ is the $j$-th time that Algorithm 1 encounters a saddle point since the beginning of a noise cycle. Let $M$ be the total number of saddle points that the algorithm meets. It is obvious that $M \leqslant I$ holds.

Here, we decompose the decrease of the global objective function values as follows:

$$F(\bar{\boldsymbol{x}}^K) - F(\bar{\boldsymbol{x}}^0) = \sum_{j=1}^{M}\left(F(\bar{\boldsymbol{x}}^{z_j(m+1)\mathcal{K}_0}) - F(\bar{\boldsymbol{x}}^{z_jm\mathcal{K}_0})\right)$$

$$+ \left(F(\bar{\boldsymbol{x}}^K) - F(\bar{\boldsymbol{x}}^{z_M(m+1)\mathcal{K}_0})\right)$$

$$+ \left(F(\bar{\boldsymbol{x}}^{z_jm\mathcal{K}_0}) - F(\bar{\boldsymbol{x}}^0)\right) \qquad (94)$$

$$+ \sum_{j=1}^{M}\left(F(\bar{\boldsymbol{x}}^{z_{j+1}m\mathcal{K}_0}) - F(\bar{\boldsymbol{x}}^{z_j(m+1)\mathcal{K}_0})\right).$$

By Theorem C.1, for $M$ encountered saddle points, the first term satisfies

$$\sum_{j=1}^{M} \left( \mathbb{E}\left[ F(\bar{\boldsymbol{x}}^{z_j(m+1)\mathcal{K}_0}) \right] - \mathbb{E}\left[ F(\bar{\boldsymbol{x}}^{z_j m\mathcal{K}_0}) \right] \right) \leqslant -MQ. \tag{95}$$

For the other items in Eq. (94), we use Eq. (90) and obtain

$$\begin{aligned}
\sum_{j=1}^{M} & \left( \mathbb{E}\left[ F(\bar{\boldsymbol{x}}^{z_{j+1} m\mathcal{K}_0}) \right] - \mathbb{E}\left[ F(\bar{\boldsymbol{x}}^{z_j(m+1)\mathcal{K}_0}) \right] \right) \\
& + \left( \mathbb{E}\left[ F(\bar{\boldsymbol{x}}^{K}) \right] - \mathbb{E}\left[ F(\bar{\boldsymbol{x}}^{z_M(m+1)\mathcal{K}_0}) \right] \right) \\
& + \left( \mathbb{E}\left[ F(\bar{\boldsymbol{x}}^{z_j m\mathcal{K}_0}) \right] - \mathbb{E}\left[ F(\bar{\boldsymbol{x}}^{0}) \right] \right) \\
& \leqslant (I - M) \frac{c\mathcal{K}_0}{8} \epsilon^2.
\end{aligned} \tag{96}$$

Therefore, if we assume the contrary of Claim 1, namely that the agents' optimization variables are close to saddle points in more than $I/4$ holding stages, i.e., $M \geqslant I/4$, we have the following inequalities under $c_Q \geqslant \frac{3}{4} c_{\mathcal{K}_0}$:

$$\begin{aligned}
\mathbb{E}\left[ F(\bar{\boldsymbol{x}}^{K}) \right] - \mathbb{E}\left[ F(\bar{\boldsymbol{x}}^{0}) \right] & \leqslant -MQ + (I - M) \frac{c\mathcal{K}_0}{8} \epsilon^2 \\
& \leqslant -\tfrac{1}{4} IQ + \tfrac{3}{4} I \cdot \frac{c\mathcal{K}_0}{8} \epsilon^2 \\
& \leqslant -\tfrac{1}{8} IQ.
\end{aligned} \tag{97}$$

This implies $\mathbb{E}\left[ F(\bar{\boldsymbol{x}}^{K}) \right] < \bar{f}$, contradicting the definition of the lower bound. Therefore, Claim 1 holds.

The second claim follows by a similar argument based on Lemma C.8. $\qquad\square$

## D EXPERIMENTS

### D.1 R AND C MATRICES USED IN SECTION 5.3

$$\mathbf{R} = \begin{bmatrix} 0.6 & 0.1 & 0 & 0.3 & 0 \\ 0.2 & 0.8 & 0 & 0 & 0 \\ 0 & 0 & 0.6 & 0.1 & 0.3 \\ 0.1 & 0 & 0.1 & 0.3 & 0.5 \\ 0 & 0 & 0 & 0.4 & 0.6 \end{bmatrix}$$

$$\mathbf{C} = \begin{bmatrix} 0.6 & 0.2 & 0 & 0.1 & 0 \\ 0.1 & 0.8 & 0 & 0 & 0 \\ 0 & 0 & 0.6 & 0.1 & 0 \\ 0.3 & 0 & 0.1 & 0.3 & 0.4 \\ 0 & 0 & 0.3 & 0.5 & 0.6 \end{bmatrix}$$

### D.2 BINARY CLASSIFICATION

In this experiment, we consider a simple $\{0, 1\}$–classification task using a neural network with a single linear hidden layer and a logistic activation function. We use the cross-entropy loss function to train the network.

We denote the feature vector as $\boldsymbol{h} \in \mathbb{R}^M$ and the binary class label as $y \in \{-1, 1\}$. For the fully connected hidden layer, we represent the weights as $\boldsymbol{W}_2 \in \mathbb{R}^{L \times M}$ and $\boldsymbol{W}_1 \in \mathbb{R}^L$. The output is of the form:

$$\hat{y} = \frac{1}{1 + e^{-\langle \boldsymbol{h}, \boldsymbol{W}_2^\top \boldsymbol{W}_1 \rangle}}. \tag{98}$$

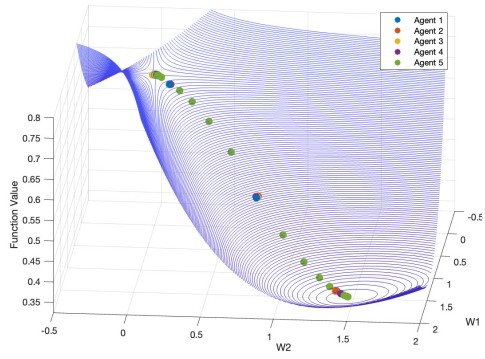
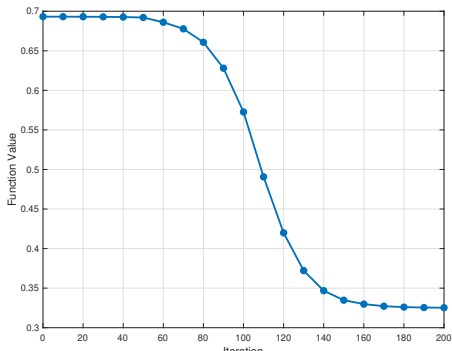

Figure 6: Trajectories of all five agents when initialized on the saddle point (0,0). Note that all trajectories overlap with each other, implying perfect consensus among the agents.

Figure 7: Global function values.

Under the commonly used cross-entry loss function, the objective function is of the following form: $L(\boldsymbol{W}_1, \boldsymbol{W}_2) = \log\left(1 + e^{-y\langle \boldsymbol{h}, \boldsymbol{W}_2^\top \boldsymbol{W}_1\rangle}\right)$. We considered the scalar case with $L = M = 1$. Synthetic data samples ($\boldsymbol{h}$ and $y$) are randomly generated under the constraint $\boldsymbol{h}y = 1$ and the regularization parameter is set to $\rho = 0.1$. One can obtain analytically that there is a saddle point at $(0,0)$. We can also verify that this saddle point is a strict saddle point since its Hessian has a negative eigenvalue of $-0.4$.

To visualize the evolution of optimization variables under our algorithm, we purposely initialized all the agents from the strict saddle point $(0,0)$ and ploted in Figure 6 the evolution of each agent and Figure 7 the global function value dynamics. The stepsize, noise amplitude, noise-injection interval, and clipping threshold were set to $\alpha = 0.1$, $\theta = 0.1$, $\mathcal{K}_0 = 50$, and $c_0 = 1.0$, respectively. It can be seen that due to the noise and gradient clipping effect, all five agents collectively move along the descending direction, clearly indicating that our algorithm can ensure efficient escape from saddle points.

### D.3 CONVOLUTIONAL NEURAL NETWORK

For this experiment, we consider the training of a convolutional neural network (CNN) for the classification of the CIFAR-10 dataset, which contains 50,000 training images across 10 different classes. We evenly spread the CIFAR-10 dataset to the five agents and set the batch size as 32. Our baseline CNN architecture is a deep network ResNet-18, the training of which is a highly nonconvex problem characterized by the presence of many saddle points. In the experiments, we train the CNN using both the proposed Algorithm 1 and the distributed gradient tracking method with noise. For our algorithm, the noise amplitude, noise-injection interval and clipping threshold were set to $\theta = 1 \times 10^{-4}$, $\mathcal{K}_0 = 100$ and $c_0 = 0.7$, respectively. The stepsize $\alpha$ used for algorithm 1 is 0.05 and 0.02 for distributed gradient tracking (0.02 is the largest stepsize that can still ensure convergence). In order to ensure fairness in comparison, both algorithms use the same noise amplitude. The evolution of loss-function values and the training accuracies averaged over 10 runs are illustrated in Figure 8 and Figure 9, respectively. It is evident that Algorithm 1 achieves lower loss function values more rapidly that compared to distributed gradient tracking with noise. This difference indicates that gradient clipping can indeed expedites saddle avoidance.

### D.4 SENSITIVITY ANALYSIS IN THE LOGISTIC REGRESSION EXPERIMENT.

In this subsection, we perform the sensitivity analysis for the noise amplitude parameter $\theta$ and clipping threshold parameter $c_0$ in the logistic regression experiment in Section 5.1. Figure 10 reveals that although noise can help the algorithm escape saddle points, it also hinders convergence. Both a too small noise amplitude and a too large noise amplitude reduce the efficiency of saddle-

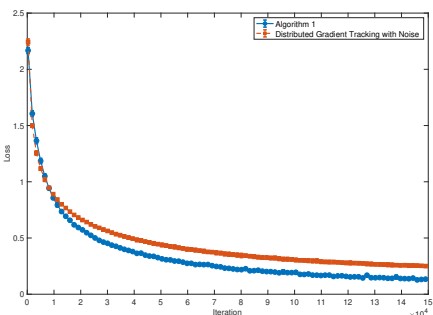

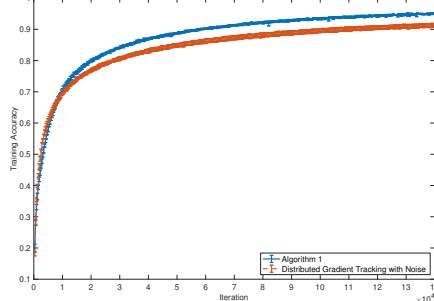

Figure 8: Loss function value comparison between the proposed algorithm and distributed gradient tracking with noise.

Figure 9: Training accuracy comparison between the proposed algorithm and distributed gradient tracking with noise

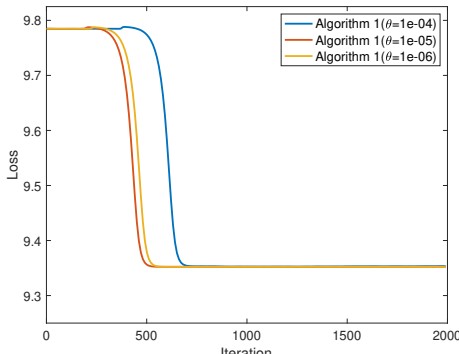

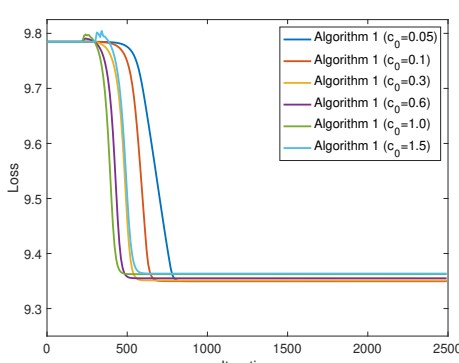

Figure 10: Influence of noise amplitude $\theta$.

Figure 11: Influence of clipping threshold $c_0$.

point avoidance. Figure 11 indicates that a larger threshold facilitates faster escape from saddle points by allowing a larger step of descent along the descent direction. However, the threshold cannot be too large, which will essentially diminish the influences of clipping and lead to reduced saddle-point escaping performance.

## D.5 EXPERIMENTAL RESULTS FOR REGULARIZED LOGISTIC REGRESSION ON THE GISETTE DATASET USING GRADIENT NORM METRICS.

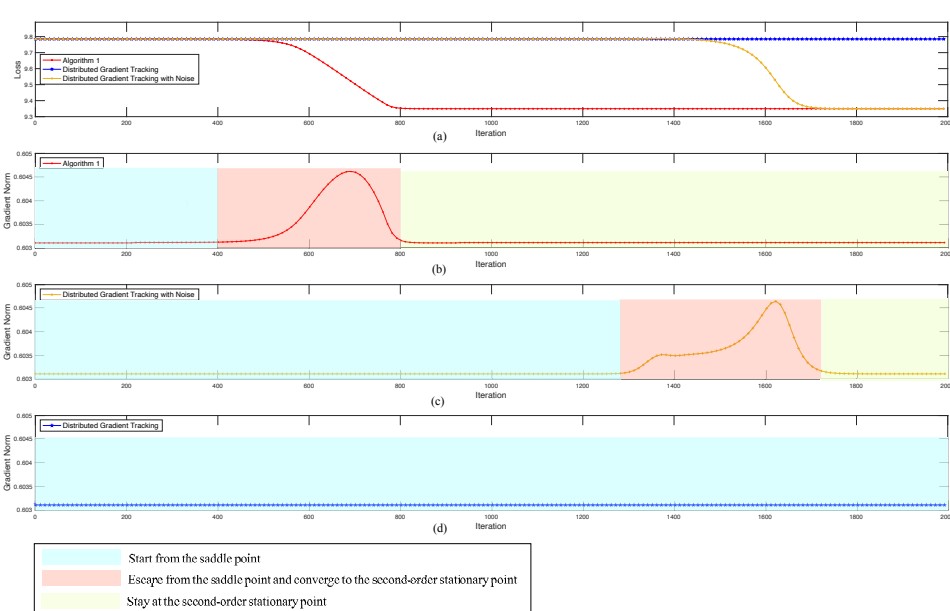

Figure 12: Influence of clipping threshold $c_0$.

Figure 12 presents the comparison results for Regularized Logistic Regression on the Gisette dataset. Specifically, Figure 12 (a) illustrates the loss function values. In Figure 12 (b), starting from a saddle point, our algorithm 1 enables saddle-point escaping from iteration around 400. (Note that the gradient norm at the saddle point is non-negotiable since the dimension of the state variable is as high as 5002.) After escaping from the saddle point, our algorithm enables convergence to a second-order stationary point starting from iteration around 685 (the peak of the curve). Starting from iteration around 800, the states reach the second-order stationary point. In Figure 12 (c), when starting from a saddle point, the distributed gradient tracking algorithm with noise enables saddle-point escaping from iteration around 1300. After escaping from the saddle point, the algorithm enables convergence to a second-order stationary point starting from iteration around 1620 (the peak of the curve). Starting from iteration around 1700, the states reach the second-order stationary point. In contrast, Figure 12 (d) shows that the state variable is trapped at the saddle point all the time under the conventional gradient tracking algorithm.

## D.6 CONVOLUTIONAL NEURAL NETWORK (CNN) ON IMAGENET

In this section, we supplement the CIFAR-100 experiments in Section 5.3 with large-scale experiments on the ImageNet ILSVRC-2012 dataset using the MobileNet-V3 architecture. ImageNet contains 1.2 million training images and 50,000 validation images across 1,000 classes. ImageNet represents one of the most widely used and computationally demanding benchmarks for deep learning, making it an ideal setting for evaluating decentralized optimization algorithms at scale. We used a pre-trained MobileNet-V3 architecture, as it is standard practice for decentralized ImageNet experiments, which significantly reduces computational cost and allows us to focus on the stability, convergence, and generalization of decentralized optimization algorithms under directed networks.

We compare the proposed Algorithm 1 with the decentralized stochastic gradient method CDSGD (Jiang et al., 2017) and the distributed gradient tracking method with noise.

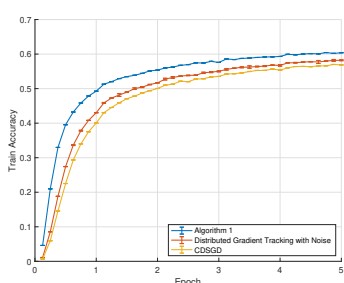 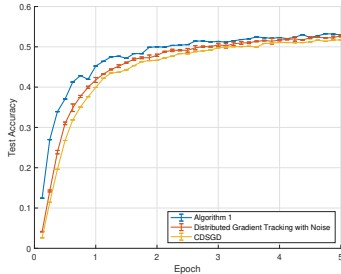

Figure 13: CNN experiment on the ImageNet dataset using the pretrained MobileNet architecture: Training accuracy.

Figure 14: CNN experiment on the ImageNet dataset using the pretrained MobileNet architecture: Test accuracy.

For Algorithm 1, the noise amplitude, noise-injection interval, and clipping threshold are set to $\theta = 1 \times 10^{-4}$, $\mathcal{K}_0 = 100$, and $c_0 = 10$, respectively. The stepsize $\alpha$ is chosen as $0.1$ for Algorithm 1. For CDSGD, the largest stepsizes that ensure convergence were adopted to provide a fair comparison. For distributed gradient tracking method with noise the stepsize is chosen as $\alpha = 0.05$, which is also the largest stepsize ensuring convergence. To ensure fairness in comparison, both algorithms use the same noise amplitude.

The training and test accuracies, averaged over 10 runs, are shown in Figure 13 and Figure 14, respectively.

From the training curves in Figure 13, we observe that Algorithm 1 achieves noticeably faster convergence in the first few epochs and consistently maintains a higher training accuracy than both CDSGD and the noisy gradient tracking method. This improved performance demonstrates that the proposed noise-injected and clipping-based mechanism provides more stable descent directions under directed communication topology.

Figure 14 further shows that Algorithm 1 also achieves superior generalization performance. It attains higher test accuracy throughout all five epochs, with particularly clear advantages during the early stages of training. Compared to CDSGD and the gradient tracking method with noise, Algorithm 1 continues to improve steadily. These results highlight that the proposed method is not only more stable and efficient during training but also yields better generalization when applied to large-scale decentralized learning tasks on ImageNet.

