# OpenReview forum: "Gradient Clipping Accelerates Saddle Avoidance in Distributed Optimization"
_ICLR.cc/2026/Conference — Submitted to ICLR 2026_

### Official Review · Reviewer_Xj1b · 2025-10-27

**Soundness:** 3
**Presentation:** 3
**Contribution:** 3
**Rating:** 4
**Confidence:** 4

**Summary:**

This paper proposes a novel distributed optimization algorithm that integrates gradient clipping with gradient tracking to accelerate saddle point avoidance in nonconvex settings. The authors demonstrate that gradient clipping, commonly used to mitigate gradient explosion in deep learning, can significantly enhance escape from saddle points and improve convergence to second-order stationary points. The algorithm employs a gradient-tracking framework with periodic noise injection and gradient clipping applied to local gradient estimates. Theoretical analysis proves consensual convergence to first-order stationary points and establishes an iteration complexity of O($\varepsilon^{-3}$) for reaching ε-second-order stationary points, outperforming existing distributed methods. Numerical experiments on benchmark datasets (e.g., Gisette, WallFlower) for logistic regression and robust PCA validate the algorithm's efficiency in escaping saddle points and achieving faster convergence compared to state-of-the-art approaches.

**Strengths:**

1. **Novel Integration:** The work is the first to successfully incorporate gradient clipping into gradient-tracking-based distributed optimization with provable convergence, addressing the challenge of nonlinearity introduced by clipping.
2. **Theoretical Rigor:** Comprehensive convergence analysis is provided, including guarantees for both first-order and second-order stationary points, with explicit complexity bounds (e.g., O($\varepsilon^{-3}$) iterations) that match centralized or semi-distributed state-of-the-art rates.
3. **Empirical Validation:** Extensive experiments on real-world datasets demonstrate the algorithm's superiority in saddle-point avoidance and convergence speed over existing methods, including scenarios with non-IID data.
4. **Practical Relevance:** Gradient clipping is already widely adopted in deep learning, so the additional benefit of accelerated saddle avoidance comes at no extra cost, enhancing its applicability.
5. **Comparative Advantage:** The algorithm achieves faster saddle evasion than all existing fully distributed counterparts, as shown in Table 1, and avoids the need for global information (e.g., counters in semi-distributed methods).

**Weaknesses:**

1. **Assumption Dependency:** The theoretical guarantees rely on strong assumptions, such as a symmetric doubly stochastic communication matrix and Lipschitz/Hessian-Lipschitz continuity of objective functions, which may not hold in all practical settings.
2. **Parameter Sensitivity:** Performance is sensitive to hyperparameters (e.g., stepsize $\alpha$, clipping threshold $c_0$, noise amplitude $\theta$), as evidenced by sensitivity analyses in experiments, requiring careful tuning for optimal results.
3. **Complexity of Analysis:** The theoretical derivations are highly technical and may be challenging to follow for practitioners, potentially limiting accessibility and implementation.
4. **Scope Limitations:** The focus on strict saddle points (excluding cases with zero minimal Hessian eigenvalues) may not cover all nonconvex problems, and the assumption of private local objectives could restrict use in scenarios requiring data sharing.
5. **Experimental Scope:** While experiments cover standard tasks, broader validation on large-scale or diverse datasets (e.g., in federated learning environments) is needed to generalize the findings.
6. **Lack of Generalization Performance Evaluation:** The experimental validation primarily focuses on training loss convergence without systematically assessing generalization capability. The absence of test set performance metrics (e.g., test accuracy for logistic regression, reconstruction error on unseen data for robust PCA) makes it difficult to evaluate overfitting risks and the algorithm's practical utility on unseen data.

**Questions:**

1. How does the algorithm perform under more realistic communication topologies (e.g., time-varying or directed graphs) that violate the symmetric doubly stochastic assumption?
2. Could the theoretical analysis be simplified to provide more intuitive insights into the role of gradient clipping in saddle avoidance?
3. What strategies are recommended for automatic tuning of hyperparameters (e.g., $c_0$, $\theta$) in practice, especially in non-stationary environments?
4. How does the algorithm scale with the number of agents and problem dimension, and are there any communication bottlenecks?
5. Have the authors considered comparing with recent adaptive gradient methods (e.g., Adam) in distributed settings to contextualize the benefits of gradient clipping?

---

> ### Author Response · Authors · 2025-11-21
> **Response to Reviewer Xj1b: Part I**
>
> **Response to Reviewer $\rm\color{green}{Xj1b}$**:
>
> We thank the reviewer for the detailed comments.  In the revised manuscript, we have made several substantial improvements. First, we have extended all theoretical results from the undirected communication setting to the more general and technically challenging directed graph setting, with explicit dependence on network spectral properties. Second, we have significantly enhanced the experimental section by adding a new CNN experiment on both CIFAR-100 (containing 100 classes) and ImageNet (with over 1,000,000 images across 1,000 classes) datasets, thereby demonstrating the applicability of our method to a larger and more complex realistic deep learning task. These revisions strengthen both the theoretical scope and the empirical validation of our work. Below, we respond to each concern in turn.
>
> ${\rm\color{green} Response \ to \ Weakness \ 1}$:
>
> Regarding the network assumption, our revised manuscript has significantly relaxed the structural requirements: the analysis is no longer limited to symmetric doubly stochastic matrices or undirected graphs. Instead, we have fully extended the approach and theoretical guarantees to directed communication networks. Moreover, the updated convergence results explicitly incorporate the number of agents $N$ and the spectral gaps of mixing matrices. As shown in Theorem 4.1 and Corollary 4.3 on page 6-page 7 of the revised manuscript, the convergence of both gradient norms and the consensus/tracking errors now depends directly on these spectral properties.
>
> We have revised the theorem statements and added clarifications in the analysis section to emphasize these topology-dependent factors (see page 6-page 7 of the revised manuscript).
>
>
>
> Regarding the smoothness assumptions, Lipschitz gradient continuity and Hessian-Lipschitz continuity are standard and widely adopted in nonconvex optimization and saddle-point escape analysis. These assumptions also appear in established works such as [1], [2], and [3], and are known to be the basic smoothness needed to guarantee stable descent and saddle-point analysis in both centralized and decentralized settings.
>
> [1] Chi Jin, Praneeth Netrapalli, Rong Ge, Sham M Kakade, and Michael I Jordan. On nonconvex optimization for machine learning: Gradients, stochasticity, and saddle points. Journal of the ACM (JACM), 68(2):1–29, 2021.
>
> [2] Rong Ge, Furong Huang, Chi Jin, and Yang Yuan. Escaping from saddle points—online stochastic gradient for tensor decomposition. In Conference on learning theory, pp. 797–842. PMLR, 2015.
>
> [3] Chi Jin, Praneeth Netrapalli, and Michael I Jordan. Accelerated gradient descent escapes saddle points faster than gradient descent. In Conference On Learning Theory, pp. 1042–1085. PMLR, 2018.
>
> ${\rm\color{green} Response \ to \ Weakness \ 2 }$:
>
> We respectfully disagree with the claim that performance is highly sensitive to hyperparameters in our experiments. In fact, although the scales of our experiments vary significantly—such as the robust PCA and the deep CNN for CIFAR-100 classification, which differ by orders of magnitude in both model size and problem complexity—the variation in the chosen parameters is relatively minor (e.g., $\theta = 0.01$ in robust PCA vs. $\theta = 0.0001$ in CIFAR-100, with both using the same noise-injection interval $\mathcal{K}_0 = 100$). Additionally, we conducted a sensitivity analysis on the logistic regression task, as detailed in Section D.4 of the Appendix, where we varied both the noise amplitude and clipping threshold across a range spanning two orders of magnitude. Even in this smaller-scale problem, the algorithm consistently avoided saddle points, demonstrating stable performance across a wide range of parameters.
>
> This insensitivity to hyperparameters is further supported by our theoretical analysis, which requires only the orders of $\alpha$, $c_0$, $\theta$, and $\mathcal{K}_0$ rather than their exact values to ensure convergence and saddle-point avoidance. Specifically, Remark 4.5 provides theoretical guidance on the orders of the key parameters: $\alpha = \mathcal{O}(\varepsilon)$, $\theta = \mathcal{O}(\varepsilon)$, $\mathcal{K}_0 = \mathcal{O}(\frac{1}{\alpha})$, and $c_0 = \mathcal{O}(\varepsilon)$. These conditions are sufficient to guarantee escape from $(\varepsilon,\sqrt{\varepsilon})$–strict saddle points. In practice, these order-of-magnitude guidelines offer a solid starting point, with the parameters then fine-tuned empirically for optimal performance if needed. Since clipping stabilizes the iterates, our algorithm typically supports larger step sizes than decentralized methods without clipping. Importantly, the theoretical constraints are more conservative than what is necessary in practice, as confirmed by our experimental results.
>
>
> (Continued on Part II)

---

> ### Author Response · Authors · 2025-11-21
> **Response to Reviewer Xj1b: Part II**
>
> ${\rm\color{green} Response \ to \ Weakness \ 3}$:
>
> This is largely due to several intrinsic difficulties of the setting we consider. First, gradient clipping, while helpful for accelerating escape from saddle regions, can amplify gradient disagreement across agents in a decentralized network: clipping distorts the relative magnitudes of local gradients, which can enlarge discrepancies between local updates and potentially destabilize the dynamics if not carefully controlled. Second, under the gradient-tracking framework there is an additional gradient-estimation variable besides the decision variable. This makes the system significantly more complex than standard decentralized gradient descent: the consensus errors (for both the decision and tracking variables) and the optimality gap become tightly coupled and cannot be analyzed independently, so our proof must control these interacting error terms simultaneously to ensure both consensus and convergence. Third, achieving second-order guarantees in a decentralized setting is intrinsically more difficult than in the centralized case, because no agent can observe the global saddle geometry; as a result, standard centralized saddle-detection and noise-scheduling mechanisms are not applicable, and we must design and analyze escape behavior without explicitly detecting saddle points.
>
> To improve accessibility for practitioners, we have added high-level explanations and proof roadmaps to summarize the main ideas behind the technical arguments, and we also introduced a clearer structural outline so that the flow of the derivations is easier to follow.
> For implementation, Remark 4.5 provides theoretical guidance on the order of the stepsize $\alpha$, noise amplitude $\theta$, and clipping threshold $c_{0}$, showing that they should satisfy $\alpha = \mathcal{O}(\varepsilon)$, $\theta = \mathcal{O}(\varepsilon)$, and $c_0 = \mathcal{O}(\varepsilon)$ to guarantee escape from $(\varepsilon,\sqrt{\varepsilon})$–strict saddle points. In practice, these conditions serve as a reasonable first choice for tuning and can be adjusted empirically based on stability. Because clipping stabilizes the iterates, our algorithm can typically adopt larger stepsizes than decentralized methods without clipping. Moreover, since we impose $\alpha c_0 = \varepsilon^2$, one can increase $\alpha$ while proportionally decreasing $c_0$; excessively large values for both may cause divergence, but moderate tuning around the theoretical scaling works well in our experiments. We believe these additions make the analysis and implementation more accessible while preserving the necessary technical depth.
>
>
> ${\rm\color{green} Response \ to \ Weakness \ 4}$:
>
> Regarding the assumption of strict saddle points, this condition is standard in the literature on second-order analysis and saddle-point escaping (e.g., [1], [2]). Strict saddles guarantee the existence of a well-defined negative-curvature direction, which enables bounded escape time and makes second-order analysis tractable. In contrast, non-strict saddle points may exhibit flat curvature, and distinguishing them from local minima is, in general, NP-hard. For this reason, both our work and prior literature focus on escaping strict saddle points, where at least one direction has strictly negative curvature and gradient-based algorithms can reliably escape.
>
> A large body of work in nonconvex optimization has shown that many practical problems satisfy the strict-saddle property. In fact, for a wide range of important applications—including principal component analysis, canonical correlation analysis, orthogonal tensor decomposition, phase retrieval, dictionary learning, matrix sensing, matrix completion, and other nonconvex low-rank models—it has been established that every saddle point is a strict saddle [2]. Moreover, in all of these problems, every local minimum is also a global minimum. Consequently, in these settings, any efficient algorithm capable of finding an $\epsilon-$second-order stationary point immediately becomes an efficient algorithm for globally solving the underlying nonconvex problem.
>
> [1] Chi Jin, Praneeth Netrapalli, Rong Ge, Sham M Kakade, and Michael I Jordan. On nonconvex optimization for machine learning: Gradients, stochasticity, and saddle points. Journal of the ACM (JACM), 68(2):1–29, 2021.
>
> [2] Rong Ge, Furong Huang, Chi Jin, and Yang Yuan. Escaping from saddle points—online stochastic gradient for tensor decomposition. In Conference on learning theory, pp. 797–842. PMLR, 2015.
>
> (Continued on Part III)

---

> ### Author Response · Authors · 2025-11-21
> **Response to Reviewer Xj1b: Part III**
>
> ${\rm\color{green} Response \ to \ Weakness \ 5,6}$:
>
> To address the concerns regarding experimental scope, generalization performance, and comparison with commonly used baselines, we have substantially expanded the experimental section in the revised manuscript.
>
> In particular, we added a new CNN experiment on the CIFAR-100 (containing 100 classes) and ImageNet (with over 1,000,000 images across 1,000 classes) datasets on pages 10 and 38, which is significantly more challenging and better reflects large-scale deep learning scenarios. In this experiment, we report both training accuracy and test accuracy, allowing us to directly assess the generalization capability of the proposed algorithm. The results show that our clipped-gradient tracking method not only converges faster but also achieves better generalization performance compared to decentralized SGD.
>
> It is worth noting that the comparison adaptive algorithms DAMSGrad and DAdaGrad exhibit poor generalization (test accuracy), which is understandable. In these methods, consensus among agents is insufficient because each agent follows its own gradient direction, with adaptive methods further amplifying this effect through local-geometry-based step sizes. This leads to overfitting to local data—reflected in their high local training accuracy—and is particularly problematic in the highly heterogeneous CIFAR-100 setting with 100 classes. As a result, these methods achieve high training accuracy but substantially lower test accuracy compared to our approach.
>
> ${\rm\color{green} Response \ to  \ Question \ 1}$:
>
> In the revised version, we have removed the symmetric doubly stochastic assumption and generalized our analysis from undirected graphs to directed communication networks. Specifically, we now work with separate row-stochastic and column-stochastic mixing matrices $R$ and $C$, which allows our results to cover a significantly broader and more realistic class of decentralized topologies. The updated convergence guarantees explicitly incorporate the number of agents $N$ and the spectral gaps $1-\sigma_R^{2}$ and $1-\sigma_C^{2}$. As shown in Theorem 4.1 and Corollary 4.3 (page 6 – page 7 of the revised manuscript), the convergence of both the global gradient norm and the consensus/tracking errors depends directly on these spectral properties.
>
> On the empirical side, we have added a new CNN experiment on the CIFAR-100 dataset over a directed communication graph, which is significantly more challenging and better reflects large-scale deep learning scenarios. In this experiment, we report both training and test accuracy to directly assess generalization. The results show that our clipped-gradient tracking method achieves better generalization performance than Decentralized SGD and the decentralized adaptive algorithms DAMSGrad and DAdaGrad. These additions strengthen both the theoretical and empirical support for the algorithm under more realistic (directed) communication topologies.
>
>
>
> ${\rm\color{green} Response \ to  \ Question \ 2}$:
>
> Thank you for the insightful question. Gradient clipping plays a critical role in escaping saddle points in the decentralized setting. Near a saddle point, the gradient norm is typically very small due to first-order stationarity. Most directions are nearly flat, and only the direction corresponding to the smallest (negative) eigenvalue of the Hessian provides a meaningful escape direction. In such cases, gradient clipping implicitly amplifies this escape direction: when the gradient norm $||y_i^k||$ is small but still above the clipping threshold $c_0$, the clipped update rescales the direction by $\frac{1}{||y_i^k||}$, producing a much larger effective step. This amplification works together with stochastic noise to push the iterates away from the saddle point more quickly, thereby accelerating saddle-point escape.
>
> (Continued on Part IV)

---

> ### Author Response · Authors · 2025-11-21
> **Response to Reviewer Xj1b: Part IV**
>
> ${\rm\color{green} Response \ to  \ Question \ 3}$:
>
> The current paper focuses on scenarios where the data distributions (or functions) remain stationary over time. For these problems, as discussed in the response to Weakness 2, determining the hyperparameters is relatively straightforward. Our theoretical guarantee requires only the orders of $\alpha$, $c_0$, $\theta$, and $\mathcal{K}_0$—not their exact values—to ensure convergence and avoid saddle points. This is further supported by our experiments, where, despite the problems differing by orders of magnitude in both model size and problem complexity, the variations in hyperparameters are relatively small. In fact, the sensitivity analysis experiment on logistic regression, detailed in Section D.4 of the Appendix, showed that the algorithm consistently avoided saddle points even when both the noise amplitude and clipping threshold were varied across a range spanning **two orders of magnitude**. Therefore, dedicated automatic tuning approaches are not necessary for these types of problems.
>
> In future work, we will explore non-stationary settings where automatic tuning will be more critical.
>
> ${\rm\color{green} Response \ to  \ Question \ 4}$:
>
> We have strengthened the paper in two directions to address this concern.
>
> (1) Extension from undirected to directed graphs.
> In the revised version, we generalize the analysis from an undirected graph with a symmetric mixing matrix to the more general directed setting. This extension allows our results to cover a significantly broader class of decentralized networks.
>
> (2) Explicit dependence on network size and spectral gaps.
> The updated convergence results explicitly incorporate the number of agents $N$ and the spectral gaps $1-\sigma_R^{2}$ and $1-\sigma_C^{2}$. As shown in Theorem 4.1 and Corollary 4.3 on page 6-page 7 in the revised version, the convergence of both gradient norms and the consensus/tracking errors now depends directly on these spectral properties. In particular, weaker connectivity or a larger network size increases the constants in the rate bounds, resulting in slower convergence and consensus. We have revised the theorem statements and added clarifications in the analysis section to emphasize agent number-dependent factors (see page 6-page 7 of the revised manuscript).
>
> Regarding scalability with respect to the problem dimension $d$, under the Lipschitz gradient and Hessian assumptions, our convergence guarantees are **independent** of $d$. Similar dimension-free results have been reported in prior work on centralized saddle-point avoidance methods [1,2]. This suggests that our algorithm **scales effectively** to high-dimensional problems, making it well-suited for modern large-scale machine learning applications.
>
>
> [1] Fang, C., Z. Lin, and T. Zhang. Sharp analysis for nonconvex SGD escaping from saddle points. COLT 2019.
>
> [2] Avdiukhin, D., and G. Yaroslavtsev. Escaping saddle points with compressed SGD. NeurIPS 2021.
>
> ${\rm\color{green} Response \ to  \ Question \ 5}$:
>
> Thank you for the suggestion. Although Adam performs efficiently in the centralized setting, directly applying it in the decentralized case often leads to divergence, as indicated in [3] and confirmed by our own results. Consequently, in the revised version, we have added comparisons with the decentralized adaptive methods DAMSGrad [3] and DAdaGrad [3], which adapt the well-known adaptive methods AMSGrad [4] and AdaGrad [5] to the decentralized setting. In this experiment, we report both training accuracy and test accuracy, allowing us to directly assess convergence speed and generalization performance. The results are summarized in the new Figures 4 and 5 on page 10.
>
> Both adaptive algorithms, DAMSGrad and DAdaGrad, exhibit poor generalization (test accuracy), which is understandable. In these methods, consensus among agents is insufficient because each agent follows its own gradient direction, with adaptive methods further amplifying this effect through local-geometry-based step sizes. This leads to overfitting to local data—reflected in their high local training accuracy—and is particularly problematic in the highly heterogeneous CIFAR-100 setting with 100 classes. As a result, these methods achieve high training accuracy but substantially lower test accuracy compared to our approach.
>
> These new large-scale results strengthen the empirical section and demonstrate that our method performs well on modern high-dimensional deep learning tasks under realistic distributed communication constraints.
>
> [3] Chen, X., Karimi, B., Zhao, W., & Li, P. On the convergence of decentralized adaptive gradient methods. In Asian Conference on Machine Learning 2023.
>
> [4] Reddi, S., Charles, Z., Zaheer, M., Garrett, Z., Rush, K., Konečný, J.,McMahan, H. B. Adaptive Federated Optimization. In ICLR 2021.
>
> [5] Duchi, J., Hazan, E., & Singer, Y. Adaptive subgradient methods for online learning and stochastic optimization. JMLR 2011.

---

### Official Review · Reviewer_XMz5 · 2025-10-31

**Soundness:** 2
**Presentation:** 3
**Contribution:** 2
**Rating:** 2
**Confidence:** 2

**Summary:**

The paper presents a gradient method for distributed machine learning that incorporates gradient tracking with gradient clipping. Theoretically, it analyzes the gradient complexity required to find a second-order stationary point and demonstrates that the method achieves state-of-the-art (SoTA) complexity bounds in some settings. Experimental results are also provided to validate the effectiveness of the proposed approach.

**Strengths:**

The paper establishes a clear problem setting, and the complexity bounds achieved under settings are demonstrated to be reasonable.

**Weaknesses:**

I find the current contribution of this paper to be incremental, and therefore recommend revision. My primary concerns are detailed below.

1.  Lack of Dependence on Network Topology: The theoretical analysis does not explicitly characterize how the convergence rates depend on the spectral properties of the connected matrix W. For a decentralized learning paper, explicitly showing this dependence is crucial.

2. Novelty of Convergence Rates: The claimed faster convergence rates do not appear to constitute a significant advancement. The same rates have already been established in the centralized setting. Furthermore, the rate provided for the deterministic setting is comparatively slow.

3. Technical Contribution and Motivation: The extension of non-convex optimization analysis to the decentralized setting is not hard. Because one can treat it as a  linear-constrained problem. The paper would greatly benefit from a more detailed and compelling explanation of its core technical novelty and motivation.

4. Insufficient Citations: The manuscript fails to properly cite the original sources of the convergence rates it obtains. For instance: The epsilon^{-1.75}  rate for the deterministic setting was first achieved by [COLT, 2017, "Accelerated Gradient Descent Escapes Saddle Points Faster than Gradient Descent"]. In the stochastic setting, techniques like variance reduction (or gradient sliding) combined with clipping have been extensively studied in centralized works like [NeurIPS, 2018, "SPIDER: Near-Optimal Non-Convex Optimization..."].

**Questions:**

See the weakness.

---

> ### Author Response · Authors · 2025-11-21
> **Response to Reviewer XMz5: Part I**
>
> **Response to Reviewer $\rm\color{purple}{XMz5}$**:
>
> We sincerely appreciate your detailed comments. In the revised manuscript, we have made several substantial improvements. First, we have extended all theoretical results from the undirected communication setting to the more general and technically challenging directed graph setting, with explicit dependence on network spectral properties. Second, we have significantly enhanced the experimental section by adding new CNN experiments on CIFAR-100 (containing 100 classes) and ImageNet (with over 1,000,000 images across 1000 classes), thereby demonstrating the applicability of our method to a larger and more complex realistic deep learning task. These revisions strengthen both the theoretical scope and the empirical validation of our work. We also revised the manuscript to address all of the comments. Please see below for the responses:
>
>  ${\rm\color{purple} Response \ to \ Weakness \ 1}$:
>
> We have strengthened the paper in two important directions to address this concern.
>
> 1. **Extension from undirected to directed graphs.**
> In the revised version, we generalize the analysis from an undirected graph with a symmetric mixing matrix to the more general directed setting by introducing separate row-stochastic and column-stochastic matrices $R$ and $C$. This extension allows our results to cover a significantly broader class of decentralized networks.
>
> 2.  **Adding explicit dependence on network size and spectral gaps.**
> The updated convergence results explicitly incorporate the number of agents $N$ and the spectral gaps $1-\sigma_R^{2}$ and $1-\sigma_C^{2}$. As shown in Theorem 4.1 and Corollary 4.3 on page 6-page 7, the convergence of both gradient norms and the consensus/tracking errors now depends directly on these spectral properties. In particular, weaker connectivity or a larger network size increases the constants in the rate bounds, resulting in slower convergence and consensus.
>
> We have revised the theorem statements and added clarifications in the analysis section to emphasize these topology-dependent factors (see page 6-page 7 of the revised manuscript).
>
> ${\rm\color{purple} Response \ to \ Weakness \ 2}$:
>
> We argue that saddle avoidance in a **fully distributed setting is intrinsically more challenging than in the centralized case**. In centralized settings, the global gradient is readily available, allowing direct access to the global saddle geometry and enabling the optimization procedure to move in a guided and informed manner to avoid saddle points. In contrast, in a fully distributed setting no information about the global gradient or global saddle geometry is available. As a result, each agent can rely only on local information, and we must design and analyze escape behavior without explicitly detecting saddle points.
>
> With this understanding—i.e., the centralized setting has full information to escape saddle points, whereas in decentralized settings individual agents lack knowledge of the global saddle geometry—it is generally **impossible to surpass** the convergence rate achieved in the centralized setting. In fact, most existing results for decentralized second-order methods exhibit convergence rates inferior to those in the centralized case. Therefore, achieving a convergence rate that matches the centralized setting in our paper is a significant result. Specifically, our results tackle the following technical challenges introduced by the decentralized structure:
>
> 1. In decentralized optimization, each agent has access only to its own local gradient rather than the full global gradient. As a result, agents must rely on communication with neighbors to approximate the global gradient, which introduces errors;
>
> 2. Gradient clipping amplifies gradient disagreement and may introduce instability. Gradient clipping is often employed to control the magnitude of updates, but in decentralized settings, it can distort the relative sizes of gradients across agents. This can amplify discrepancies between local updates among agents, and potentially destabilize the optimization dynamics;
>
> 3. Under the gradient-tracking framework, the presence of an additional gradient-estimation variable, in addition to the decision variable, significantly complicates the system dynamics compared to conventional decentralized gradient descent. In particular, the consensus errors and the optimality gap become tightly coupled and cannot be analyzed independently, requiring us to control these interacting error terms simultaneously to ensure both consensus and optimality;
>
> 4. Achieving second-order guarantees in decentralized systems is substantially more difficult because no agent can observe the global saddle geometry, rendering standard centralized saddle-detection mechanisms inapplicable.
>
> Showing that stable convergence and reliable saddle escape remain possible under all these challenges constitutes the main technical contribution of our work.
>
> (Continued on Part II)

---

> ### Author Response · Authors · 2025-11-21
> **Response to Reviewer XMz5: Part II**
>
> ${\rm\color{purple} Response \ to \ Weakness \ 3}$:
>
> We thank the reviewer for this comment and for prompting us to clarify our core technical novelty and motivation. Below, we summarize the motivations, key difficulties and our contributions.
>
> **Motivation for the algorithm design.**
> Introducing gradient clipping in the decentralized setting significantly complicates the dynamics. While clipping can accelerate escape from saddle regions, it also amplifies gradient disagreement, and naïvely clipping local gradients can easily lead to divergence. To address this, our method incorporates gradient tracking (GT), enabling agents to operate on global gradient estimators rather than raw local gradients. We clip the global gradient estimation variables instead of the local gradients, which is essential for maintaining stability while still benefiting from clipping.
>
> **Challenges in Combining Gradient Tracking and Clipping.**
> Combining clipping with gradient tracking is far from straightforward, as clipping is a nonlinear operator that significantly complicates the analysis. Our analysis must jointly control three tightly coupled quantities: the optimality gap, the consensus error of the optimization variables, and the consensus error of the gradient-tracking variables. These errors influence each other bidirectionally, and none of them can be analyzed in isolation. We therefore develop a new coupled system of recursion inequalities to show that all three quantities remain controlled under clipping, and that the algorithm converges to a neighborhood of a second-order stationary point over directed networks. In fact, pushing the iterates too aggressively toward global optimality can enlarge the consensus error, while forcing the consensus error to shrink too quickly can slow down progress toward optimality; this intrinsic tension further illustrates why a coupled analysis is necessary.
>
> **Difficulties of decentralized second-order guarantees.**
> Obtaining second-order guarantees introduces an additional layer of complexity. In centralized settings, detecting saddle points is straightforward because the global gradient and curvature information are directly available. In decentralized settings, however, the global saddle structure is not observable at any single agent, making classical saddle-point detection and noise-injection strategies inapplicable. Our analysis and algorithm design must therefore ensure saddle escape without explicitly detecting a saddle point, which significantly increases the technical complexity of the approach.
>
> Taken together, these aspects—the nontrivial integration of clipping with gradient tracking, the coupled analysis of optimality and consensus errors, the decentralized second-order guarantees, and the fully distributed directed-graph setting—form the core technical contributions of our work. We have added a concise explanation of these points on page 8 of the revised version to better convey the motivation and novelty of our analysis.
>
> ${\rm\color{purple} Response \ to \ Weakness \ 4}$:
>
> Thank you for pointing this out. In the revised manuscript, we have updated Table 1 to correctly reflect the original convergence-rate sources and to provide more complete citations.

---

### Official Review · Reviewer_3qpC · 2025-11-01

**Soundness:** 4
**Presentation:** 4
**Contribution:** 4
**Rating:** 8
**Confidence:** 3

**Summary:**

Distributed optimization is becoming increasingly more important in the era when machine learning models are becoming larger. Saddle points can significantly impede training efficiency, and escaping from saddle points can be even more complex in distributed training. This paper proposed a distributed optimization algorithm based on gradient clipping that is usually adopted to avoid gradient explosion. The authors proved the convergence of the proposed method and empirically demonstrated its effectiveness in tasks including non-convex regularized logistic regression, robust PCA, and neural network-based classification.

**Strengths:**

1. The paper is very clearly structured and well-written.

2. It is surprising to see that simple gradient clipping can help escape from saddle points, since gradient clipping is usually used at the other end of scenario where the gradient magnitude is too large. The effect is intuitive though, since it basically just modifies the effective stepsize. This usage is novel.

3. The theoretical complexity is better than existing saddle avoidance approaches.

4. Figure 2 clearly illustrates the benefits of gradient clipping.

**Weaknesses:**

1. Selecting appropriate values for stepsize $\alpha$, clipping threshold $c_0$, noise amplitude $\theta$, and noise-injection interval $\kappa_0$ may be difficult for various real-world tasks. Normally, these parameters should be set case by case.

2. The conducted experiments are not large-scale enough.

3. The algorithm only converges to a ball centered at a second-order stationary point. Since the noise is not injected at every iteration, it would be better if the authors could remove $\theta$ from the convergence rate bound.

**Questions:**

See above.

---

> ### Author Response · Authors · 2025-11-21
> **Response to Reviewer 3qpC: Part I**
>
> Response to Reviewer $\rm\color{blue}{3qpC}$:
>
> We sincerely thank the reviewer for the detailed, thoughtful, and positive assessment of our work. In the revised manuscript, we have made several substantial improvements. First, we have extended all theoretical results from the undirected communication setting to the more general and technically challenging directed graph setting, with explicit dependence on network spectral properties. Second, we have significantly enhanced the experimental section by adding a new CNN experiment on both CIFAR-100 (containing 100 classes) and ImageNet (with over 1,000,000 images across 1,000 classes) datasets, demonstrating the applicability of our method to a more complex and realistic deep learning task. These revisions strengthen both the theoretical scope and the empirical validation of our work. We also revised the manuscript to address all of the comments. Below, we address the reviewer’s concerns point by point.
>
> ${\rm\color{blue} Response \ to \ Weakness \ 1}$:
>
> We agree that the optimal values of these parameters depend on factors such as network size, topology, and function properties. However, in our theoretical results, exact parameter values are not required—only their orders of magnitude matter, as indicated in the revised Remark 4.5.
>
> Specifically, Remark 4.5 provides theoretical guidance on the orders of the key parameters: $\alpha = \mathcal{O}(\varepsilon)$, $\theta = \mathcal{O}(\varepsilon)$, $\mathcal{K}_0 = \mathcal{O}(\frac{1}{\alpha})$, and $c_0 = \mathcal{O}(\varepsilon)$, which are sufficient to guarantee escape from $(\varepsilon,\sqrt{\varepsilon})$–strict saddle points. In practice, these order-of-magnitude guidelines provide a reasonable starting point, and the parameters can be fine-tuned empirically to achieve optimal performance (if needed). Since clipping stabilizes the iterates, our algorithm typically supports larger stepsizes than decentralized methods without clipping. Notably, the theoretical constraints are more conservative than what is required in practice, as confirmed by our experiments.
>
> In fact, our experiments show that parameter choices are very flexible. Although the scales of the experiments differ substantially—for example, robust PCA and the CNN used in the CIFAR-100 experiment differ by orders of magnitude in model size and problem complexity—the variation in chosen parameters is comparatively small (e.g., $\theta = 0.01$ in robust PCA vs. $\theta = 0.0001$ in CIFAR-100, with both using the same noise-injection interval $\mathcal{K}_0=100$). We also conducted a sensitivity analysis on the logistic regression task in Section D.4 of the Appendix, varying both the noise amplitude and clipping threshold over a range spanning two orders of magnitude. The algorithm consistently escaped saddle points (despite the small scale of the problem), demonstrating stable performance of our algorithm across wide parameter ranges.
>
> These empirical observations reinforce the idea that the theoretical order-of-magnitude conditions offer considerable flexibility in practice. As a result, we believe that setting these parameters is practical for a broad range of real-world tasks.
>
> (Continued on Part II)

---

> ### Author Response · Authors · 2025-11-21
> **Response to Reviewer 3qpC: Part II**
>
> ${\rm\color{blue} Response \ to \ Weakness \ 2}$:
>
> We appreciate the reviewer’s feedback and have substantially expanded the experimental section in the revised manuscript to address this concern. In particular, we added new CNN experiments on the CIFAR-100 and ImageNet datasets (with over one million images across 1,000 classes), thereby demonstrating the applicability of our method to a larger and more complex, realistic deep learning task.
>
> We employ a deep CNN architecture and evaluate performance over a directed communication graph, matching the theoretical setting of our paper. In this experiment, we report both training accuracy and test accuracy, allowing us to directly assess convergence speed as well as generalization performance. The results show that our clipped-gradient tracking algorithm converges noticeably faster than the gradient tracking method without clipping. By comparing our algorithm with the CDSGD and the decentralized adaptive algorithms DAMSGrad and DAdaGrad, we achieve higher test accuracy, indicating a stronger generalization ability.  (Both CDSGD and the decentralized adaptive algorithms DAMSGrad and DAdaGrad exhibit poor generalization—test accuracy—which is understandable. In these methods, consensus among agents is insufficient because each agent follows its own gradient direction, with adaptive methods further amplifying this effect through local-geometry-based step sizes. This causes overfitting to local data—reflected in their high local training accuracy—and is particularly problematic in the highly heterogeneous CIFAR-100 setting with 100 classes. As a result, these methods achieve high training accuracy but substantially lower test accuracy compared to our approach.) These new large-scale results strengthen the empirical section and demonstrate that our method performs well on modern high-dimensional deep learning tasks under realistic distributed communication constraints.
>
> To further strengthen the empirical evaluation, we additionally include a new set of large-scale decentralized experiments on ImageNet using a pre-trained MobileNet-V3 architecture, which is a standard practice for decentralized ImageNet studies. ImageNet represents one of the most computationally demanding and widely adopted benchmarks in deep learning, making it an ideal setting for evaluating decentralized optimization algorithms at scale. Our results show that Algorithm 1 achieves faster convergence during training and consistently higher test accuracy compared to CDSGD and the decentralized gradient-tracking baseline with noise, across all five training epochs. These results highlight that our method remains stable and efficient even in this challenging large-scale setting, while also yielding better generalization than competing decentralized baselines.
>
> We have added the new ImageNet experiment and corresponding discussion in subsection D.6 "Convolutional Neural Network (CNN) on ImageNet" (see pages 38-39 of the revised manuscript), and expanded Section 5.3 to include the CIFAR-100 results.
>
>
> ${\rm\color{blue} Response \ to \ Weakness \ 3}$:
>
>
> We thank the reviewer for this insightful comment. In fact, our algorithm injects noise only for a finite number of cycles.  As shown in Theorem 4.6 (Corollary 4.7), Algorithm 1 reaches an $\epsilon$–second-order stationary point within  $\mathcal{O}(\epsilon^{-3})$ iterations. Therefore, noise is injected only during the first $\mathcal{O}(\epsilon^{-3})$ iterations, and after this point, all subsequent updates are purely noise-free.
>
> Consequently, over the infinite-time horizon, the convergence bound in Theorem 4.1 can be rewritten as
> $\frac{1}{k} \sum_{\mathrm{s}=1}^{k}{\bar{\alpha}_s\mathbb{E} \left[ \|\| \nabla F\left( \boldsymbol{\bar{x}}^s \right) \|\| ^2 \right]} \leqslant \mathcal{O}(\frac{N\theta^2}{k}) + \mathcal{O}(\frac{\sqrt{N}}{k(1-\sigma_C^2)})$, which shows that the influence of periodically injected noise decays as $\mathcal{O}(\frac{N\theta^2}{k})$ and thus vanishes as $k \to \infty$. This implies that Algorithm 1 eventually exhibits exact convergence.
>
> We have clarified this point in the revised manuscript (see Remark 4.8 on page 8 of the revised manuscript).

---

### Official Review · Reviewer_GGBi · 2025-11-01

**Soundness:** 2
**Presentation:** 1
**Contribution:** 1
**Rating:** 2
**Confidence:** 4

**Summary:**

The paper works on the problem of avoiding saddle points for distributed nonconvex optimization. They claim to show that gradient clipping can be used to find a $\epsilon-$second-order stationary point after $O(\epsilon^{-3})$ iterations. Some robust PCA experiments are performed.

**Strengths:**

The paper claims (Theorem 4.6) it finds an $\epsilon-$second-order stationary point after $O(\epsilon^{-3})$ iterations in a distributed (not semi-distributed) setting.

**Weaknesses:**

The main weakness of the paper is that it does not clarify its novelty compared to Xian and Huang (2023). The paper states that the work covers the semi-distributed setting, but it doesn't explain what is the difficulty in extending to the fully distributed setting. What is the main advantage offered here that was not possible for Xian and Huang (2023)? And to what degree do the authors claim the issue of "semi-distributed" is actually a concern? [See also questions.]

Relatedly, the paper claims that "results for distributed nonconvex optimization are relatively sparse". However, the paper does not cite [1-3], all of which have results for this setting, and even have "decentralized" and "nonconvex"/"non-convex" directly in their titles. (This is not an exhaustive list, just meant to provide evidence, from a very basic literature search, that rebuts the claim.)

Based on these limitations, the result appears incremental in comparison to the (inadequately reviewed) prior literature.

[1] Haoran Sun, Songtao Lu, and Mingyi Hong. "Improving the sample and communication complexity for decentralized non-convex optimization: Joint gradient estimation and tracking." In International Conference on Machine Learning, pp. 9217-9228. PMLR, 2020.

[2] Taoxing Pan, Jun Liu, and Jie Wang. "D-SPIDER-SFO: A decentralized optimization algorithm with faster convergence rate for nonconvex problems." In Proceedings of the AAAI Conference on Artificial Intelligence, vol. 34, no. 02, pp. 1619-1626. 2020.

[3] Ran Xin, Usman Khan, and Soummya Kar. "A hybrid variance-reduced method for decentralized stochastic non-convex optimization." In International Conference on Machine Learning, pp. 11459-11469. PMLR, 2021.

**Questions:**

The authors claim they achieve $\epsilon-$second-order stationary point after $O(\epsilon^{-3})$ iterations (not $\tilde{O}(\epsilon^{-3})$). Is this a typo? Or do the authors actually believe their claim holds without a $O(\mathrm{polylog}(1/\epsilon))$ factor? If the latter, can the authors explain how they manage to remove the $O(\mathrm{polylog}(1/\epsilon))$ term for achieving $\epsilon-$second-order stationary point?

Should it not be $\tilde{O}$ notation in Xian and Huang (2023) and Avdiukhin & Yaroslavtsev (2021) in "Semi-distributed" rows of Table 1?

Could you elaborate on the distinction between "distributed" and "semi-distributed"? Is there any other work that formalizes this distinction? If not, where in the current work is such a formal definition provided?

Precisely how does introducing gradient clipping significantly complicate convergence analysis? Assuming such complications exist, were they not already addressed in Xian and Huang (2023), which already gives a $\tilde{O}(\epsilon^{-3})$ iteration complexity to find an $O(\epsilon, \sqrt{\epsilon})$ stationary point?

Reference error: The work of Xian and Huang was part of the proceedings of NeurIPS 2023.

---

> ### Author Response · Authors · 2025-11-21
> **Response to Reviewer GGBi: Part I**
>
> **Response to Reviewer $\rm\color{orange}{GGBi}$**:
>
> We thank the reviewer for the detailed comments. In the revised manuscript, we have made several substantial improvements. First, we have generalized results from undirected (bidirectional) interaction graphs to directed graphs. Second, we enhanced the experimental section by adding new CNN experiments on CIFAR-100 and ImageNet datasets, demonstrating the applicability of our method to a more complex and realistic deep learning task. We also revised the manuscript to address all of the comments. Below, we address each concern in turn.
>
> ${\rm\color{orange} Response \ to \ Weakness \ 1}$:
>
> We now clarify more explicitly why our contribution goes beyond the semi-distributed setting in Xian and Huang (2023). The algorithm in Xian and Huang (2023) fundamentally relies on a semi-distributed architecture, in the sense that it is only partially decentralized and still requires local agents to access global information or network-wide aggregation at each step. For instance, its termination mechanism depends on knowing exactly how many worker nodes satisfy a certain stopping condition; to realize this, they introduce an additional global aggregation process that collects Boolean indicators from all agents. Although they argue that transmitting Booleans is cheaper than broadcasting full model parameters, this mechanism still presupposes the existence of a global counter or coordinator-like functionality, which is incompatible with a truly fully distributed system where each node can only communicate with its nearest neighbors and no global variable is ever formed. This reliance on global information not only limits scalability to very large networks but also makes their approach less suitable in privacy-sensitive or infrastructure-constrained environments, where maintaining any form of global aggregation channel is undesirable or infeasible. In contrast, our algorithm is fully distributed by design: all operations are based solely on local states and neighbor-to-neighbor communication, without any global statistics, global counters, or centralized processes.
>
> It is worth noting that saddle avoidance in a **fully distributed setting is intrinsically more challenging** than in the centralized or semi-distributed cases. With access to global information—such as the global gradient in centralized settings or global state statistics in semi-distributed settings—individual agents can observe (at least partially) the global saddle geometry. This makes it possible for agents to move in a coordinated manner and to use coordinated noise-scheduling mechanisms to avoid saddles.
>
> In contrast, no information about the global gradient or global saddle geometry is available in the fully distributed case. As a result, each agent can rely only on local information, and we must design and analyze escape behavior without explicitly detecting saddle points.
>
> We have clarified these fundamental differences and contributions with respect to Xian and Huang (2023) in the new footnote 3 on page 8.
>
> Moreover, our analysis in the revised version supports directed graphs, whereas Xian and Huang (2023) does not. This is significant as real-world communication networks are often asymmetric or unbalanced.
>
>
> ${\rm\color{orange} Response \ to \ Question \ 1}$:
>
> We obtain $\mathcal{O}(1/\epsilon^{3})$ complexity rather than $\tilde{\mathcal{O}}(1/\epsilon^{3})$ due to the use of gradient clipping. In prior work, each noise injection must be followed by a sufficiently long sequence of iterations so that the injected noise accumulates enough effect along the negative curvature direction to guarantee saddle-point escape [1]. This accumulation process is precisely what leads to the extra $\mathcal{O}(\mathrm{polylog}(1/\epsilon))$ factor in prior work.
>
> In contrast, with clipping, the escape from saddle points becomes faster. In the neighborhood of a saddle point, the gradient norm is small. Most directions are relatively flat, and only the direction associated with the smallest eigenvalue of the Hessian—the direction of negative curvature—offers a weak but meaningful escape direction. In such scenarios, gradient clipping implicitly amplifies the escape direction because in our update rule, when the gradient $||\boldsymbol{y}_i^k||$
> is small but still larger than the threshold $c_0$, the clipped value is scaled by $\frac{1}{||\boldsymbol{y}_i^k||}$. This amplification works together with stochastic noise to drive the iterates away from the saddle point more quickly.  As a result, we obtain a sufficiently large decrease in the saddle direction more quickly, eliminating the $\mathcal{O}(\mathrm{polylog}(1/\epsilon))$ overhead.
>
> [1] C. Jin et al. On nonconvex optimization for machine learning: Gradients, stochasticity, and saddle points. J. ACM, 68(2):1–29, 2021.
>
> ${\rm\color{orange} Response \ to \ Question \ 2}$:
>
> Thanks for pointing out the typo in Table 1, and have corrected it.
>
> (Continued on Part II)

---

> ### Author Response · Authors · 2025-11-21
> **Response to Reviewer GGBi: Part II**
>
> ${\rm\color{orange} Response \ to \ Question \ 3}$:
>
> Regarding the terminology, we clarify that “semi-distributed” is used in the literature to describe algorithms that are partially distributed but still require some form of global information, a coordinator, or network-wide aggregation at certain steps. Such requirements prevent these methods from operating in a fully decentralized manner. For example, in Xian & Huang (2023), the termination strategy relies on knowing how many worker nodes satisfy the stopping condition. To obtain this global count, their method introduces an additional mechanism that aggregates Boolean indicators across the entire network. While the authors argue that transmitting Boolean values is cheaper than broadcasting full model parameters, this procedure still fundamentally depends on global information, which is unavailable in a truly fully distributed system where each node communicates only with its local neighbors. In contrast, our algorithm is fully distributed: all decisions are made solely based on local information and neighbor communication, without requiring any global aggregation or central coordination. In the revised version, we have added our definition of ``semi-distributed" in footnote 3 on page 8. We note that this reflects our interpretation, and we recognize that others may define the term differently.
>
> ${\rm\color{orange} Response \ to \ Question \ 4}$:
>
> Gradient clipping introduces significantly more difficulty in the fully distributed setting. In centralized analysis, clipping only affects the descent direction, and its bias can be directly tracked using the global gradient. In contrast, in decentralized optimization, clipping interacts with the network disagreement dynamics, creating new challenges. Different agents have different local gradients, and the clipping operator can amplify small discrepancies among agents into large deviations after mixing, potentially leading to instability or divergence. Furthermore, clipping destroys linearity, which complicates both the gradient-tracking recursion and the convergence analysis.
>
> In fact, Section 3.2.2 of Xian & Huang (2023) also acknowledges the analytical difficulty of gradient normalization—a special case of our clipping with $c_0=0$. To avoid this complication, their algorithm employs an alternative mechanism that is substantially more complex and relies on global information, making it only semi-distributed.
>
> By contrast, our method remains fully distributed. Through a careful analysis of the clipped stepsizes (as shown in Lemma A.1 and A.2), we bound the deviation between the local clipped stepsize and its global counterpart ($ | \alpha_ i^k - \bar{\alpha}_ i^k|  || \boldsymbol{y}_ i^k||$) by using the optimization consensus errors ($|| \boldsymbol{e}_ {x,k} ||$) and gradient tracking errors ($|| \boldsymbol{e}_{y,k}||$). This allows us to establish stable convergence behavior even under fully distributed, directed communication without requiring any global information.
>
> ${\rm\color{orange} Response \ to \ Question \ 5}$:
>
> Thank you for pointing out the reference error. We have corrected it in the revised manuscript.
>
>
> ${\rm\color{orange} Response \ to \ Weakness \ 2 \ on \  Missing \ Some \ References}$:
>
> We thank the reviewer for pointing out this issue. Our intention was not to claim that results for decentralized nonconvex optimization are sparse, but rather that distributed methods with second-order convergence guarantees remain relatively limited. To avoid confusion, we have updated the main text to clearly reflect this distinction (see page 1 of the revised manuscript).

---

### Author Response · Authors · 2025-11-27

Dear Reviewer,

I hope everything is going smoothly. I am writing to kindly ask whether it might be possible for you to provide your follow-up comments on our rebuttal. We believe we have addressed the concerns you raised, and we would greatly appreciate hearing your updated perspectives.

Thank you very much for your time and effort.

Sincerely,

The Authors

---

### Meta-Review · Area_Chair_ZhTM · 2025-12-15

**Summary:**

This paper proposed a decentralized optimization method for finding the $\epsilon$-second-order stationary point of the nonconvex function with the complexity of $O(\epsilon^{-3})$. The main contribution is showing that the gradient clipping benefits to escaping from saddle point in decentralized setting. After rebuttal, I think the authors have clarified the difference between this submission and  Xian & Huang (2023), i.e., the implementation of Xian & Huang (2023) relies on global aggregation, which is not fully decentralized.
However, I think the other point have not been well addressed. I agree the comments of  Reviewer XMz5 that the achieved complexity is not satisfied. Based on assumptions of Lipschitz gradient and Hessian, the deterministic case should desire the computational complexity of $\tilde O(\epsilon^{-2})$, rather than $O(\epsilon^{-3})$.

**Reviewer Concerns:**

The reviewers have the following main concerns:
1. The comparison with Xian & Huang (2023) is unclear.
2. The complexity of $O(\epsilon^{-3})$ is worse than that of centralized setting.

**Reviewer Scores:**

I think Reviewer GGBi may raise the score because the authors have explained the differences between their method and that of Xian and Huang (2023).

---

### Decision · Program_Chairs · 2026-01-26

Reject